# LINGUA: Bridging the Grounding Gap in VideoQA via Typed Memory and Belief-State Reasoning

Saman Forouzandeh [1]  Wei Peng [1]  Xinghuo Yu [1]  Mahdi Jalili [1]

## Abstract

VideoQA models can be accurate yet often fail to align answers with the correct video segments (the *grounding gap*). We introduce **LINGUA** (**L**anguage-based **IN**ference for **G**rounded Video **U**nderstanding **A**gent), a memory-based agent that performs grounded VideoQA by reasoning in an explicit *linguistic belief state*. LINGUA uses five mechanisms: (1) event-driven perception (retains 8–12% of frames while preserving 94% of question-relevant events); (2) typed memory for episodic narratives, semantic affordances, and procedural scripts; (3) Belief-Action-Verification loops with postcondition and temporal checks; (4) meta reflection with contrastive refinement; and (5) Bayesian reliability tracking for continual learning without gradient updates. Built with Gemma3-4B (Ollama, 4-bit), LINGUA outperforms strong baselines on five VideoQA benchmarks, reaching 82.4% on NExT-QA and 42.3% Acc@GQA on NExT-GQA (answer + IoU≥0.5 temporal localization), while running 2.6× faster than dense-frame methods. In continual learning over 100 videos, accuracy rises from 45.2% (first 10) to 61.8% (last 10) without catastrophic forgetting, indicating online adaptation via memory refinement.

## 1. Introduction

Video Question Answering (VideoQA) has advanced rapidly with the emergence of large-scale vision–language models (VLMs) (Lei et al., 2021; Li et al., 2023a). Despite high accuracy achieved on benchmark datasets, recent studies reveal a critical limitation: correct answers do not necessarily reflect reliable *grounded reasoning*. Xiao et al. (Xiao et al.,

2024) show that while a state-of-the-art system reaches 69% accuracy in NExT-QA, only 16% correct predictions can be justified by the relevant video segments in natural-language explanations. This discrepancy, referred to as the *grounding gap*, highlights the disconnect between answer accuracy and grounded justification, limiting the reliability of VideoQA systems in scenarios that demand interpretable, verifiable, and causally grounded reasoning. In this paper, we use **grounding** to refer specifically to enforcing consistency between reasoning hypotheses and observable video evidence during inference, through explicit temporal consistency and causal outcome checks.

Recent progress in VideoQA has improved *how candidate video evidence is localized and aligned*, through more accurate temporal span selection and stronger grounding supervision. However, the *grounding gap* remains: models can answer correctly without being tightly bound to temporally and causally consistent evidence. Localize-then-answer pipelines (Gao et al., 2018; Lei et al., 2018; Xiao et al., 2024) explicitly select supporting segments, while long-video approaches prioritize scalable context selection via event or keyframe retrieval and hierarchical expansion (e.g., Koala (Tan et al., 2024), VideoTree (Wang et al., 2025b) and Vidctx (Goulas et al., 2024)). In parallel, large open-weight VLMs and datasets (e.g., Molmo/PixMo (Deitke et al., 2025)) strengthen visual perception backbones. Beyond selection, recent grounding-centric work improves alignment though training-time supervision, such as QGAC-TR (Xu et al., 2024) and narrative/grounded supervision (Liang et al., 2025). Other approaches leverage multimodal LLMs for temporal grounding and span refinement (Enrich and Detect (Pramanick et al., 2025), LeAdQA (Dong et al., 2025), Grounded-VideoLLM (Wang et al., 2025a)), including bidirectional or causality-aware objectives (TimeCraft (Liu et al., 2024), cross-modal causal alignment (Chen et al., 2025)) and online grounding settings (OVG-HQ (Zeng et al., 2025)). Agentic VideoQA improves the grounding of VideoQA via latent trajectory abstraction (Sarch et al., 2024) or aggregation of post hoc multi-path reasoning output (Dang et al., 2025). Despite these advances, most systems still treat grounding as a *preprocessing and post hoc selection/alignment problem*, with limited support for inference-time evaluation of explicit reasoning hypotheses against

---

[1]School of Engineering, Royal Melbourne Institute of Technology University, Melbourne, Australia. Correspondence to: Saman Forouzandeh <[saman.forouzandeh@rmit.edu.au]>.

*Proceedings of the 43rd International Conference on Machine Learning*, Seoul, South Korea. PMLR 306, 2026. Copyright 2026 by the author(s).

temporal and causal evidence. As a result, evidence often remains weakly coupled to decision making: models can produce correct answers even when selected evidence is incomplete, noisy, or temporally inconsistent (Xu et al., 2024), reflecting a broader limitation in how grounding and reasoning are integrated in contemporary VLMs (Pantazopoulos & Özyiğit, 2025).

We argue that this limitation stems from a deeper design issue: grounding signals (e.g., keyframes or supervision signals) are not represented as explicit reasoning constraints. We address this gap with *linguistic belief-state reasoning*, where hypotheses are explicitly verified against evidence at inference time, making grounding intrinsic to reasoning rather than a preprocessing, post hoc, or training-time alignment step. Drawing on the concept of symbolic grounding (Harnad, 1990; Davidsson, 1993), and building on frame-semantic affordances (Fillmore et al., 2002) and procedural memory for LLM agents (Forouzandeh et al., 2026), we propose **LINGUA** (**L**anguage-based **IN**ference for **G**rounded Video **U**nderstanding **A**gent). LINGUA is a memory-based agent for VideoQA that reasons entirely in a linguistic belief state space, constructed via **event-driven perception** and maintained through **typed memories** (episodic, semantic, and procedural memories) for hosting timestamped narratives, semantic affordances, and procedural scripts. Rather than treating grounding as a preprocessing step or a post hoc descriptive explanation, LINGUA integrates grounding directly into its reasoning process via explicit verification of temporal and causal postconditions. LINGUA executes iterative **Belief-Action-Verification (BAV)** cycles. **Belief** retrieves linguistically described events and affordances relevant to the question and generates goal hypotheses; **Action** selects candidate procedural hypotheses based on relevance and estimated reliability; **Verification** evaluates whether expected outcomes and temporal constraints are supported by observed evidence. Verification results are used to update memory-level reliability estimates, enabling continual refinement without gradient-based retraining. When repeated failures occur, a **meta reflection** mechanism diagnoses linguistic mismatches or missing causal links and refines procedural representations through contrastive analysis.

We evaluate LINGUA on five VideoQA benchmarks covering causal, temporal, and long-horizon reasoning. Implemented with Gemma3-4B using 4-bit quantization, LINGUA consistently outperforms strong baselines. It attains 82.4% accuracy on NExT-QA and 42.3% Acc@GQA on NExT-GQA (grounded accuracy requiring both correct answers and IoU≥0.5 temporal localization), while processing videos 2.6× faster than dense-frame methods. In continual learning settings, accuracy improves from 45.2% on the first 10 videos to 61.8% on the last 10 videos over a 100-video stream without catastrophic forgetting in the experiments. Together, these results suggest that explicit linguistic belief

states and verification-driven reasoning offer a viable path toward closing the grounding gap in VideoQA. This paper makes the following contributions:

1. We propose a VideoQA system design that 0explicitly couples grounding with reasoning, treating evidence as an integral part of inference rather than relying on implicit embedding representations, prepossessing or post hoc processing, or training-time alignment.
2. We present **LINGUA**, a typed memory agent that instantiates this design by performing VideoQA in linguistic space using episodic, semantic, and procedural memories with belief-state representations.
3. We develop an inference time Belief-Action-Verification framework that enforces temporal and causal grounding and enables continual refinement without gradient-based retraining, leading to improved grounded accuracy and long-horizon robustness across multiple benchmarks.

## 2. Methodology

LINGUA performs video reasoning entirely in a *linguistic space* through five integrated components. **Event-driven perception** generates percepts as *image-conditioned* natural-language descriptions of salient frames using vision–language models. **Typed complementary memories** store evidence traces as timestamped linguistic narratives, semantic representations of affordances, and procedural schemas defined in terms of preconditions, actions, and postconditions, each augmented with reliability estimates. **Belief-Action-Verification (BAV) loops** perform Bayesian reasoning over linguistic belief states. **Meta reflection** diagnoses reasoning failures, and **contrastive refinement** improves procedural schemas through iterative experience. Figure 1 illustrates the overall LINGUA framework and the interactions among these components.

**Event-Driven Perception.** To reduce computation and focus on semantically meaningful events, we select frames exhibiting *semantic change* or containing objects with actionable affordances rather than processing all frames uniformly. Let $f_t$ denote the frame at timestamp $t$. We select $f_t$ if: (i) its visual embedding differs from the previous selected frame by more than $\tau_\Delta$, or (ii) it contains goal-relevant objects that enable actions (e.g., tools, food, containers).

**Semantic Change Detection.** We measure semantic change using VideoMAE-v2 (Wang et al., 2023), encoding two-frame windows $[f_{t-1}, f_t]$ into 768-dimensional embeddings that preserve motion cues critical for temporal reasoning (e.g., distinguishing "is walking toward" from "has reached"). The semantic change score is:

$$\Delta_{\text{sem}}(t) = 1 - \frac{\mathbf{v}_t \cdot \mathbf{v}_{t-1}}{\|\mathbf{v}_t\| \|\mathbf{v}_{t-1}\|}, \tag{1}$$

where $\mathbf{v}_t = \text{VideoMAE}([f_{t-1}, f_t])$. Frames with

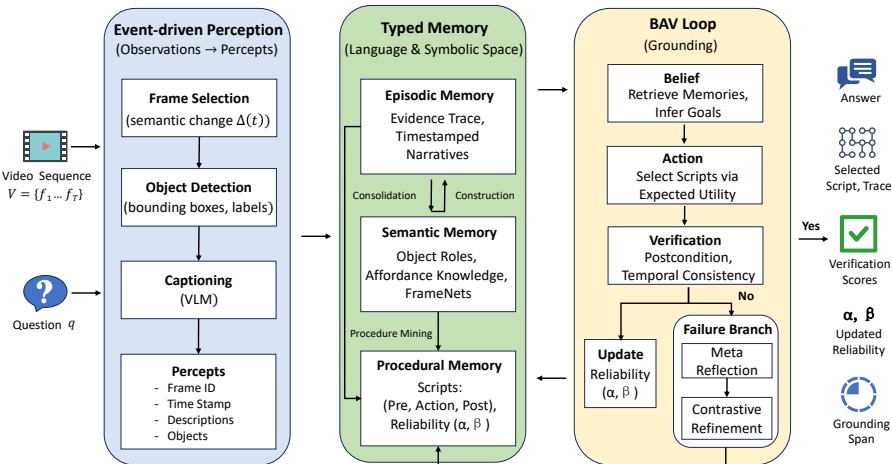

*Figure 1.* LINGUA architecture using a unified Gemma3-4B model for both vision-language grounding and text-only linguistic reasoning within the BAV loop and meta reflective module. All components operate in linguistic space, enabling full interpretability. "Pre" stands for preconditions. "Post" denotes postconditions. $\alpha$ and $\beta$ represent Bayesian reliability scores associated with procedural memory scripts.

$\Delta_{\mathrm{sem}}(t) > \tau_\Delta$ are selected.

**Affordance-Based Attention.** Following Gibson's affordance theory (Gibson, 2014) and FrameNet (Fillmore et al., 2002), we identify objects that evoke action-relevant semantic frames using YOLOv8 (Jocher et al., 2023). For instance, "knife" evokes the *Cutting* frame with roles [Instrument=knife, Agent=?, Patient=?]. Objects with confidence $> 0.5$ trigger frame selection regardless of semantic change.

**Vision-Language Description Generation (Captioning).** For each selected frame $f_t$, we generate *image-conditioned* natural language descriptions using Gemma3-4B in vision-language mode. Given detected objects $\mathcal{O}_t = \{o_1, \ldots, o_k\}$ from YOLO, we construct a vision-language prompt:

$p_{\mathrm{desc}} =$"Describe this scene in one sentence.

 Use temporal markers (before, after, while, then)

 and causal words (because, so that).

 Focus on actions with: $[o_1, \ldots, o_k]$"

(2)

The description is generated via:

$$d_t = \mathrm{Gemma3\text{-}4B}_{\mathrm{VL}}(p_{\mathrm{desc}}, f_t \,|\, T = 0.1), \qquad (3)$$

where the subscript VL denotes vision-language mode.

**Typed Memory Systems.** LINGUA stores and retrieves information through three typed complementary memories, all represented in natural language rather than opaque visual embeddings.

*A) Episodic Memory.* Episodic memory stores timestamped event summaries following narrative grammar (Mandler, 1982) and semantic role labeling conventions (Palmer et al., 2005). Each entry records semantic roles (Agent, Action,

Patient following PropBank ARG0/predicate/ARG1), temporal span $[t_s, t_e]$, goal/outcome cues, the natural language description $d_i$, VideoMAE embedding $\mathbf{v}_i$ for retrieval, and Bayesian reliability parameters $\alpha_i, \beta_i$:

$$m_i^{\mathrm{epi}} = \langle \mathrm{Agent}, \mathrm{Action}, \mathrm{Patient}, [t_s, t_e], \\ \mathrm{Goal}, \mathrm{Outcome}, d_i, \mathbf{v}_i, \alpha_i, \beta_i \rangle. \qquad (4)$$

Semantic roles are extracted using spaCy's PropBank-style dependency parsing. Adjacent entries merge into longer narratives when temporally close (gap $< \Delta t_{\mathrm{merge}} = 2$s), semantically consistent (embedding similarity $> 0.85$), and linguistically continuous (markers like "then", "next"). The agent infers goals by matching observed affordances to frame-based semantic patterns. Goal inference uses Gemma3-4B in text-only mode:

$$\mathcal{G}_i = \mathrm{Gemma3\text{-}4B}_{\mathrm{text}}(p_{\mathrm{goal}}, q, \mathcal{E}_i, \mathcal{A}_i \,|\, T = 0.1), \quad (5)$$

where the subscript text denotes text-only mode.

*B) Semantic Memory.* Semantic memory encodes object-centered affordances using frame-semantic structures following FrameNet (Fillmore et al., 2002). For each object, we store action possibilities as semantic frames with typed roles, linguistic properties, and semantic relations:

$$m_j^{\mathrm{sem}} = \langle \mathrm{Object}, \mathcal{A}_j, \mathcal{P}_j, \mathcal{R}_j, \mathbf{s}_j, c_j \rangle, \qquad (6)$$

where $\mathcal{A}_j$ are affordances, $\mathcal{P}_j$ are linguistic properties, $\mathcal{R}_j$ are semantic relations, $\mathbf{s}_j$ is an embedding, and $c_j$ is confidence. We prompt Gemma3-4B in text-only mode to generate affordances as semantic frames with typed roles in

format [*FrameName*: [Role1=?, Role2=?, ...]], for instance:

$$p_{\text{aff}} = \text{``Extract affordances for object as FrameNet frames.}$$
$$\text{Format: [FrameName: [Role1=value, Role2=value]]}$$
$$\text{Object: knife''}$$
(7)

yielding [*Cutting*: [Instrument=knife, Agent=human, Patient=food]]. Unlike brittle symbolic matching, we leverage distributional semantics for retrieval:

$$\text{Retrieve}(q) = \{m_j^{\text{sem}} : \text{sim}(\mathbf{s}_q, \mathbf{s}_j) > \gamma_{\text{aff}}\}, \quad (8)$$

enabling "spoon" and "teaspoon" (similarity 0.93) to share affordances, and "knife" and "blade" (similarity 0.88) to evoke similar *Cutting* frames.

*C) Procedural Memory.* Procedural memory captures reusable action schemas following script theory (Schank & Abelson, 2013). Each script is represented as a structured schema consisting of a goal $\mathcal{G}_k$ (script name), preconditions $\Psi_k$ expressed as required semantic frames, an ordered action sequence $\Pi_k$ annotated with temporal markers, postconditions $\Phi_k$ describing expected outcome states, temporal constraints $\{\mu_i, \sigma_i\}$ capturing typical action durations, and Bayesian reliability parameters $(\alpha_k, \beta_k)$:

$$m_k^{\text{proc}} = \langle \mathcal{G}_k, \Psi_k, \Pi_k, \Phi_k, \{\mu_i, \sigma_i\}, \alpha_k, \beta_k \rangle. \quad (9)$$

We automatically extract 127 scripts from NExT-QA training videos, covering 68% of causal questions, using: (1) multimodal narrative generation with temporal/causal markers, (2) PropBank-style semantic role labeling, (3) FrameNet-based clustering, (4) canonical schema extraction with temporal statistics, and (5) schema validation ($n_{\min} = 5$, $\sigma_i/\mu_i < 0.5$).

**Bayesian Reliability Tracking.** Each memory item maintains a Beta posterior $\text{Beta}(\alpha, \beta)$ that estimates how reliably it supports correct reasoning. The expected reliability is $\mathbb{E}[\rho] = \alpha/(\alpha + \beta)$. All priors are initialized as $(\alpha_0, \beta_0) = (1, 1)$ (uniform) and updated after each verification step using simple Bayesian increments:

$$(\alpha, \beta) \leftarrow \begin{cases} (\alpha + 1, \beta) & \text{if verification succeeds,} \\ (\alpha, \beta + 1) & \text{if verification fails.} \end{cases} \quad (10)$$

This lightweight mechanism enables continual adaptation without gradient-based training. As the agent processes additional videos, procedural reliabilities converge toward empirical success rates, while semantic memory expands with new affordances and episodic memory accumulates experience, enabling learning over time without catastrophic forgetting.

**Belief-Action-Verification (BAV) Loops.** Given question $q$, the agent repeats BAV cycles until confident, with all reasoning operating over natural language representations. Belief

module extracts key entities from $q$ via spaCy named entity recognition, retrieves relevant affordances and episodic events using semantic similarity (sentence-transformers), then generates goal hypotheses through Gemma3-4B reasoning over question text, entities, affordances, and episodic context:

$$p_{\text{goal}} = \text{``Classify the goal of video question using}$$
$$\text{contextual information. Question: } q$$
$$\text{Entities: } \mathcal{E}, \text{ Affordances: } \mathcal{A}$$
$$\text{Episodic context: } \mathcal{M}^{\text{epi}}\text{''}$$
(11)

The output is a ranked list of candidate goals expressed as natural language scripts with expected semantic frames (e.g., "prepare_meal" expects *Cooking* and *Ingestion* frames). **Action** retrieves candidate procedural scripts via semantic similarity between goal descriptions and script names, then computes expected utility balancing reliability, relevance, and exploration:

$$\text{EU}_k = \underbrace{\text{Rel}_k}_{\text{precondition match}} \cdot \underbrace{\mathbb{E}[\rho_k]}_{\text{reliability}} - \underbrace{\text{Risk}_k}_{\text{past failures}} \\ + \lambda_{\text{info}} H(\text{Beta}(\alpha_k, \beta_k)), \quad (12)$$

where $\text{Rel}_k$ measures semantic similarity between observed descriptions and schema preconditions using sentence embeddings, $\text{Risk}_k$ penalizes contradictions with past failure narratives, and the entropy term $H(\cdot)$ encourages exploration of uncertain schemas. We select $\mathcal{P}^* = \arg \max_k \text{EU}_k$ and trigger reflection if $\max_k \text{EU}_k < \tau_{\text{EU}}$. Verification checks script postconditions using semantic similarity ($\gamma_{\text{post}} = 0.8$) and temporal consistency. Postcondition coverage measures what fraction of expected outcomes appear in observed descriptions:

$$c_{\text{post}} = \frac{|\Phi^* \cap \text{Outcomes}_{\text{observed}}|}{|\Phi^*|}, \quad (13)$$

where intersection is computed via semantic similarity thresholding. Temporal consistency verifies that observed action durations fall within expected ranges derived from corpus statistics:

$$c_{\text{temporal}} = \frac{1}{|\Pi^*|} \sum_i \mathbb{K}[d_{\text{obs},i} \in [\mu_i - 2\sigma_i, \mu_i + 2\sigma_i]]. \quad (14)$$

Verification outcomes update Bayesian posteriors via Eq. (10), enabling incremental learning from experience.

**Meta Reflection.** Reflection is triggered when reasoning repeatedly fails (e.g., three or more consecutive failures), when postcondition coverage falls below a threshold (e.g., 0.3), or when semantic drift in linguistic descriptions exceeds a predefined bound (e.g., 0.7), indicating a breakdown in linguistic reasoning coherence. In such cases, Gemma3-4B analyzes the decision log expressed as a natural-language

narrative, performing linguistic error analysis to identify semantic mismatches (e.g., incorrect frame selection), temporal contradictions (e.g., confusion between "before" and "after"), or missing causal links:

$$p_{\text{reflect}} = \text{"Diagnose video understanding failure.}$$
$$\text{Failed attempts: } \mathcal{F}; \text{ Current objects: } \mathcal{O}. \quad (15)$$
$$\text{What went wrong? Suggest alternative."}$$

When reflection identifies gaps, we consolidate through episodic-semantic interaction: novel events (embedding similarity $< 0.3$ or mismatched semantic frames) create new entries in both memories; familiar events refine existing narratives while pdating associated affordances. This bidirectional interaction ensures semantic coherence beyond embedding proximity.

**Contrastive Refinement.** When sufficient success and failure cases accumulate for a script (minimum $n_{\text{contrast}} = 3$ of each), we perform contrastive procedural refinement following the framework in (Forouzandeh et al., 2026), adapted for video understanding. Gemma3-4B analyzes success versus failure narratives to extract discriminative semantic features:

$$p_{\text{contrast}} = \text{"Extract discriminative features for script.}$$
$$\text{SUCCESS cases: } \mathcal{S}_k$$
$$\text{FAILURE cases: } \mathcal{F}_k \quad (16)$$
$$\text{Identify positive/negative discriminators."}$$

For instance, "cooling" scripts succeed when descriptions contain "refrigerator" or "freezer" but fail with "oven" or "stove".

We define procedural schemas through discriminator-driven updates. For video understanding, we extend this with visual-linguistic constraints:

$$m_k^{\text{proc}} \leftarrow \text{Refine}(m_k^{\text{proc}}, \Delta\Psi_k^+, \Delta\Psi_k^-, \Delta\Pi_k, \Delta\Phi_k, \mathcal{V}_k), \quad (17)$$

where $\Delta\Psi_k^+/\Delta\Psi_k^-$ are positive/negative precondition discriminators, $\Delta\Pi_k$ are action sequence refinements, $\Delta\Phi_k$ are postcondition updates, and $\mathcal{V}_k$ enforces visual-linguistic consistency (e.g., detected objects must align with affordance frames). The "cooling" script learns to require mentions of cold appliances while excluding heat sources, progressively improving procedural quality through contrastive linguistic analysis while maintaining full interpretability. The algorithm, theoretical analysis, and execution trace analysis are provided in Appendices A, B, and C, respectively.

## 3. Experiments

We evaluate **LINGUA** on five VideoQA benchmarks to assess: (i) answer accuracy across temporal, causal, and descriptive reasoning, (ii) temporal grounding quality via IoU and IoP metrics, and (iii) continual online adaptation without gradient-based training.

**Datasets.** We evaluate on: **NExT-QA** (Xiao et al., 2021) (52K multiple-choice questions covering temporal, causal, and descriptive reasoning over 5.4K YouTube videos), **NExT-GQA** (Xiao et al., 2024) (10.5K grounded QA pairs with precise temporal span annotations), **EgoSchema** (Mangalam et al., 2023) (5,031 multiple-choice questions testing long-horizon egocentric understanding over 3-minute clips), **IntentQA** (Li et al., 2023b) (3.5K compositional questions requiring goal inference and intent understanding), and **Ego4D-NLQ** (Grauman et al., 2022) (natural-language query localization in long-form egocentric videos).

**Evaluation Metrics.** We follow official benchmark protocols with rigorous statistical testing. *Accuracy* is reported for NExT-QA (broken down by Temporal/Causal/Descriptive question types), EgoSchema, and IntentQA. For NExT-GQA, we report: (i) *IoU-based grounding*: Recall@0.3, Recall@0.5, and mean IoU (mIoU); (ii) *IoP-based grounding*: Recall@0.3, Recall@0.5, and mean IoP (mIoP); and (iii) *Acc@GQA*, grounded accuracy requiring both a correct answer *and* IoU$\geq$0.5 with ground-truth temporal span. For Ego4D-NLQ, we report Recall@1 at IoU thresholds $\{0.3, 0.5, 0.7\}$. All improvements are statistically significant ($p < 0.05$, paired t-test, $n = 1{,}000$ videos unless noted).

**Model Architecture and Implementation.** We implement LINGUA with a unified vision-language model **Gemma3-4B** (Team et al., 2024) (4B parameters) served locally via Ollama (Ollama Team, 2026), avoiding external API dependencies. Gemma3-4B operates in two modes: **(i) vision-language** for generating image-conditioned frame descriptions and extracting affordances from visual inputs, and **(ii) text-only** for downstream linguistic reasoning, including goal inference, planning, verification, and meta reflection over natural language belief states. For event-driven perception, we integrate frozen auxiliary encoders (VideoMAE-v2 (Wang et al., 2023)) for semantic change detection and YOLOv8 (Jocher et al., 2023) for object detection. The resulting system centers on a compact 4B-parameter reasoning core that runs entirely locally on a single NVIDIA H200 (141GB VRAM), with all auxiliary perception modules executed on the same device.

**Note:** We distinguish grounding as an *inference mechanism* (verification of temporal and causal constraints during reasoning) from grounding as an *evaluation metric* (IoU, Acc@GQA measuring alignment with ground-truth spans). The former drives LINGUA's reasoning process; the latter assesses task performance.

**Baselines.** We compare against representative VideoQA, temporal grounding, and agentic reasoning methods span-

ning 770M–14B parameters: (i) *End-to-end VLMs*: LLaVA-7B (Li et al., 2023a), InstructBLIP (Dai et al., 2023); (ii) *Language-centric VideoQA*: Video-ChatGPT (Maaz et al., 2024), LLaMA-VID (Li et al., 2024b), TimeChat (Ren et al., 2024), DoraemonGPT (Yang et al., 2024); (iii) *Memory-augmented*: MemBridge (Yang et al., 2023), MA-LMM (He et al., 2024); (iv) *Grounding-based*: SeViLa (Yu et al., 2023), TGB (Wang et al., 2024b), GCG (Wang et al., 2024a); (v) *LLM-driven temporal grounding*: LeAdQA (Dong et al., 2025), Timecraft (Liu et al., 2024), and CRA (Chen et al., 2025), which integrates causal-aware query refinement, supervised temporal grounding, and MLLM-based answer decoding on NExT-QA and NExT-GQA; and (vi) *Modular/agentic systems*: MoReVQA (Min et al., 2024), LLoVi (Zhang et al., 2024), MSR-ViR (Song et al., 2025), VideoTree (Wang et al., 2025b), and MUPA (Dang et al., 2025). Table 1 reports the results.

**Egocentric & Long-Horizon Understanding.** Table 2 shows that LINGUA transfers to egocentric, long-horizon videos. It achieves **66.2/48.1** on EgoSchema (subset/full), outperforming MSR-ViR (61.2/46.0) by **+5.0/+2.1**. On IntentQA, it reaches **74.8%** (**+1.7** over Video-ChatGPT). For Ego4D-NLQ, it attains **31.7/18.6/9.4** Recall@1 at IoU 0.3/0.5/0.7, improving by **+3.8/+2.3/+1.6** over the best baselines, indicating robust long-horizon state tracking and temporal localization under occlusions and rapid viewpoint changes.

**General Video Benchmark: Video-MME.** To complement the NExT- and Ego-family benchmarks with broader-domain coverage, we evaluate LINGUA on Video-MME (Fu et al., 2025), which spans six domains and three duration levels. Table 3 reports results under the official protocol (without subtitles, single H200, Gemma3-4B at $T$=0.1).

LINGUA (4B) surpasses VITA-1.5 (7B) by +12.4pp and LLaVA-NeXT-Video (34B) by +16.5pp overall, without Video-MME-specific tuning. On the long-video subset, it exceeds memory-based agents M3-Agent (+14.1pp) and WorldMM-8B (+3.4pp) despite a smaller backbone, indicating that event-driven perception and typed memory provide advantages that are not purely a function of model scale. VideoDeepResearch (Yuan et al., 2025) targets hour-scale benchmarks (LVBench, MLVU) with a different evaluation setup and is included as related agentic work rather than a directly comparable baseline.

**Ablation Studies.** Table 4 shows that LINGUA's gains arise from complementary components. **Typed memory** is dominant: removing procedural memory causes the largest drop (−8.2 Acc@GQA), while removing all memory collapses performance (−11.3). **Reliability-aware action selection** is also essential, with heuristic or non-Bayesian variants degrading Acc@GQA by 3.5–6.2 points. **Event-driven perception and multimodal grounding** further improve robust-

ness, as uniform sampling or removing VideoMAE reduces performance by 5.3–7.1 points. Finally, **meta-reflection and linguistic structure** are necessary for stable reasoning, with removals causing up to 5.8 point degradation.

**Memory Content Analysis.** Beyond the aggregate ablation numbers, the three memory types contribute complementarily rather than redundantly. *Episodic* memory undergoes temporal merging that reduces raw entries by 30% (Appendix F.2.1) while preserving narrative coherence; its removal costs −5.5pp on Acc@GQA. *Semantic* memory supplies the affordance–frame mappings used by all 127 procedural scripts; removing it costs −4.7pp. *Procedural* memory carries the Bayesian reliability tracking that drives action selection; its removal causes the largest single-component drop (−8.2pp), and removing all three memory types collapses performance by −11.3pp. Table 14 in Appendix D further shows that the gap between LINGUA-11B and LINGUA-4B widens from 0.8pp to 1.7pp once structured memory is removed, indicating that typed memory disproportionately benefits smaller backbones by compensating for limited parametric capacity.

**Computational Efficiency with Controlled Comparison.** To rigorously evaluate efficiency claims, we compare LINGUA against uniform sampling under controlled conditions. Table 5 shows that at matched accuracy (42.3% Acc@GQA), uniform sampling requires $3.1\times$ longer runtime; conversely, at matched computational budget ($\sim$7s), event-driven selection achieves +10.2pp higher Acc@GQA. This demonstrates that semantic event selection provides fundamental efficiency-accuracy advantages beyond simple frame rate reduction, dominating the Pareto frontier (see Appendix E for detailed analysis).

**Controlled settings.** Both methods use Gemma3-4B with identical prompts (Section 2), temperature $T$=0.1, and max tokens of 150 (descriptions) and 50 (answers), without batching. VL calls generate frame descriptions, while text calls handle goal inference, answering, and verification. LINGUA introduces more text calls (5.2 vs. 3), averaging 2.2 BAV iterations per video, adding $\sim$260 ms overhead ($5.2\times50$ ms) compared to $\sim$27.7 s VL cost ($99\times280$ ms), confirming that event-driven perception dominates efficiency gains.

**Continual Learning Dynamics.** Figure 2 demonstrates LINGUA's continual learning across three perspectives: cold-start acquisition, question-type adaptation, and phase transitions.

**Backward Transfer: Catastrophic Forgetting Test.** To rigorously validate the "no catastrophic forgetting" claim (Abstract, Proposition 7), we evaluate backward transfer by re-testing on early videos after processing later batches. Table 6 shows results:

*Table 1.* **Performance on NExT-QA and NExT-GQA benchmarks.** T/C/D denote Temporal/Causal/Descriptive question types. **Bold** indicates best performance, underline second-best. Standard errors across 3 runs shown as subscripts. All LINGUA improvements are statistically significant ($p < 0.01$, paired t-test). *LINGUA-11B (Appendix D) achieves only +0.5% performance gain over LINGUA-4B while requiring 2.7× more GPU hours, demonstrating that structured reasoning reduces dependence on model scale.*

| | NExT-QA (Accuracy) | | | | | NExT-GQA (Grounded VideoQA) | | | | | | |
| | Question Type (%) | | | | | IoU Metrics | | | IoP Metrics | | | |
| Method | T | C | D | Avg | Params | R@0.3 | R@0.5 | mIoU | R@0.3 | R@0.5 | mIoP | Acc@GQA |
|---|---|---|---|---|---|---|---|---|---|---|---|---|
| *End-to-End Vision-Language Models* | | | | | | | | | | | | |
| LLaVA-7B | 64.9 | 69.7 | 79.4 | 69.6 | 2.7B | 8.5 | 3.2 | 18.2 | 24.8 | 11.5 | 23.1 | 12.8 |
| InstructBLIP | 70.5 | 71.5 | 79.8 | 72.5 | 3.1B | 12.1 | 5.3 | 21.0 | 29.3 | 14.2 | 27.8 | 15.7 |
| *Language-Centric VideoQA Methods* | | | | | | | | | | | | |
| Video-ChatGPT | 71.5 | 73.8 | 80.5 | 73.2 | 7B | 13.8 | 6.1 | 22.1 | 31.5 | 15.2 | 29.5 | 17.8 |
| LLaMA-VID | 70.8 | 73.2 | 80.1 | 72.8 | 7B | 13.3 | 5.8 | 21.3 | 30.8 | 14.6 | 28.7 | 16.9 |
| TimeChat | 73.1 | 72.9 | 79.8 | 73.5 | 7B | 17.1 | 8.5 | 24.8 | 34.2 | 17.1 | 32.8 | 19.4 |
| DoraemonGPT | 72.3 | 74.5 | 80.3 | 74.1 | 7B | 15.3 | 7.2 | 23.5 | 33.1 | 16.2 | 31.2 | 18.9 |
| *Memory-Augmented Methods* | | | | | | | | | | | | |
| MemBridge | 69.5 | 71.8 | 79.5 | 71.9 | 1.3B | 11.2 | 4.9 | 20.5 | 29.7 | 13.9 | 27.4 | 16.2 |
| MA-LMM | 70.8 | 72.5 | 80.2 | 73.1 | 7B | 14.1 | 6.3 | 22.3 | 31.2 | 15.1 | 29.8 | 17.5 |
| *Grounding-Based Multimodal Models* | | | | | | | | | | | | |
| SeViLa | 69.4 | 74.2 | 81.3 | 73.8 | 7B | 12.8 | 5.4 | 21.7 | 29.2 | 13.8 | 28.5 | 16.6 |
| TGB | 66.5 | 72.8 | 81.2 | 72.1 | 3B | 23.3 | 11.2 | 19.9 | 38.5 | 22.8 | 31.2 | 20.3 |
| GCG | 72.6 | 74.2 | 80.7 | 74.6 | 7B | 15.9 | 7.8 | 23.1 | 32.8 | 15.9 | 30.9 | 18.2 |
| *LLM-Driven Temporal Grounding* | | | | | | | | | | | | |
| LeAdQA-7B | 66.6 | 72.5 | 82.3 | 72.1 | 7B | 31.3 | 17.5 | 20.5 | 39.6 | 29.5 | 30.3 | 19.2 |
| TimeCraft [Temp] | 64.7 | 66.5 | 75.1 | 65.6 | 0.5B | 21.2 | 9.6 | 15.6 | 35.1 | 27.8 | 28.1 | 18.2 |
| TimeCraft [Frozen] | 70.1 | 79.5 | 79.4 | 74.7 | 1.2B | 18.6 | 8.4 | 13.2 | 32.7 | 24.9 | 26.3 | 18.5 |
| CRA [Temp] | 62.5 | 65.3 | 75.1 | 61.1 | 0.5B | 21.4 | 10.6 | 14.2 | 34.3 | 28.5 | 28.6 | 18.2 |
| CRA [Frozen] | 66.9 | 76.8 | 77.2 | 70.2 | 1.2B | 20.1 | 9.6 | 13.5 | 32.6 | 25.9 | 26.5 | 18.8 |
| *Modular & Agentic Reasoning Systems* | | | | | | | | | | | | |
| MoReVQA | 68.2 | 71.5 | 78.8 | 69.2 | 6B | 47.3 | 23.8 | 29.8 | 54.2 | 36.5 | 37.8 | 39.6 |
| LLoVi | 70.2 | 73.7 | 81.9 | 73.8 | 7B | 29.8 | 16.2 | 21.5 | 48.5 | 38.0 | 39.4 | 26.8 |
| MSR-ViR$_{Large}$ | 69.9 | 73.4 | 81.5 | 73.6 | 8.7B | 32.1 | 16.2 | 23.4 | 38.2 | 23.3 | 28.2 | 18.9 |
| MSR-ViR$_{Large}$ | 72.2 | 74.6 | 80.9 | 74.9 | 14.2B | 33.6 | 16.4 | 23.4 | 39.0 | 24.1 | 29.6 | 18.6 |
| VideoTree | 70.6 | 76.5 | 83.9 | 75.6 | GPT-4 | 17.8 | 8.9 | 25.5 | 35.8 | 18.2 | 33.0 | 20.7 |
| MUPA-2B | 71.8 | 73.2 | 80.1 | 73.5 | 2B | 49.4 | 25.6 | 27.2 | 57.0 | 38.7 | 39.1 | 28.7 |
| MUPA-7B | 72.5 | 74.1 | 80.6 | 74.8 | 7B | 54.2 | 27.3 | 33.4 | 60.6 | 39.4 | 41.4 | 30.3 |
| **LINGUA** | **78.5**$_{+0.6}$ | **81.3**$_{+0.8}$ | **87.4**$_{+0.5}$ | **82.4**$_{+0.5}$ | **4B** | **55.6**$_{+1.1}$ | **36.5**$_{+1.3}$ | **38.4**$_{+0.9}$ | **63.2**$_{+1.0}$ | **42.7**$_{+1.2}$ | **45.5**$_{+0.8}$ | **42.3**$_{+1.1}$ |
| **vs. Best Baseline** | +5.4 | +1.8 | +3.5 | +6.8 | – | +1.4 | +9.2 | +5.0 | +2.6 | +3.3 | +4.1 | +2.7 |

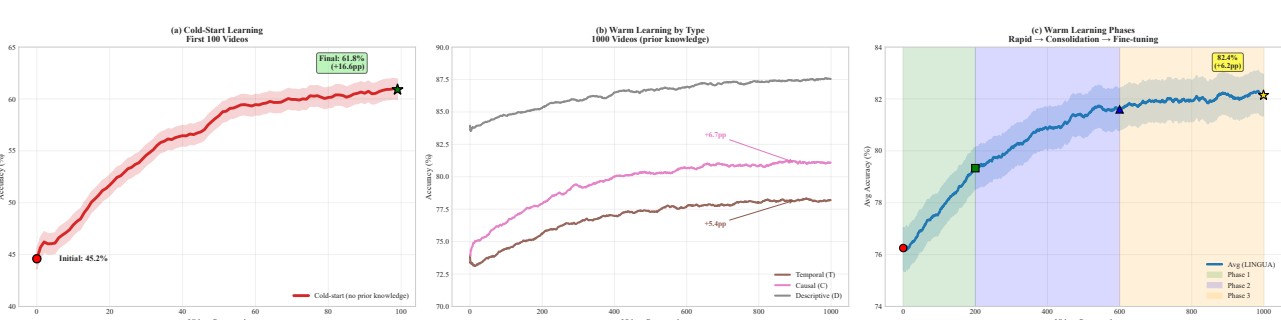

Continual Learning Dynamics (All Panels in One Figure)

*Figure 2.* **Continual learning: cold-start, question-type dynamics, and learning phases. (a)** Cold-start: 45.2%→61.8% over 100 videos (+16.6pp). **(b)** Warm learning by type over 1,000 videos: Causal +6.7pp (largest), Temporal +5.4pp, Descriptive +4.0pp. **(c)** Three phases: rapid discovery (0–200, ∼0.03pp/video), consolidation (200–600, ∼0.015pp/video), fine-tuning (600–1000, ∼0.005pp/video) reaching 82.4%.

Table 6 highlights three trends: (1) **No catastrophic forgetting**, with consistently positive backward transfer (+0.7 to +2.6pp), supporting Proposition 7; (2) **Early gains**, where initial videos benefit most (+2.6pp) from improved script discrimination; and (3) **Stability**, as later batches show marginal change (+0.7pp), indicating accumulation without degradation. This contrasts with gradient-based continual learning, which often degrades earlier performance by 10–30% (French, 1999).

**Script discovery dynamics.** To understand how quickly new procedural knowledge is acquired and whether automatic extraction converges, we analyze the cumulative growth of discovered scripts as more videos are processed. Figure 3 illustrates the rate of script discovery over 1,000 videos, revealing distinct phases of rapid acquisition, con-

*Table 2.* **Egocentric video understanding results. Bold** indicates best performance, underline second-best. Standard errors shown as subscripts ($n = 3$ runs). All LINGUA improvements significant at $p < 0.05$.

| Method | EgoSchema | | IntentQA | Ego4D-NLQ (R@1) | | |
| | Subset | Full | Acc | 0.3 | 0.5 | 0.7 |
|---|---|---|---|---|---|---|
| *Baselines* | | | | | | |
| Video-ChatGPT | 59.7 | 44.3 | 73.1 | 27.2 | 15.8 | 7.8 |
| TimeChat | 60.3 | 45.2 | 72.8 | 27.9 | 16.1 | 7.6 |
| MA-LMM | 60.8 | 42.5 | 72.8 | 27.8 | 16.3 | 7.7 |
| MSR-ViR$_L$ | 61.2 | 46.0 | 72.4 | 26.5 | 15.2 | 7.1 |
| **LINGUA** | **66.2**$_{\pm 1.2}$ | **48.1**$_{\pm 0.9}$ | **74.8**$_{\pm 0.7}$ | **31.7**$_{\pm 0.8}$ | **18.6**$_{\pm 0.6}$ | **9.4**$_{\pm 0.4}$ |
| vs. Best Baseline | +5.0 | +2.1 | +1.7 | +3.8 | +2.3 | +1.6 |

*Table 3.* **Video-MME results (no subtitles).** M3-Agent's long-video score is reported via WorldMM (Table 1 of (Yeo et al., 2025)). LINGUA was not tuned on Video-MME.

| Method | Params | Short | Medium | Long | Overall |
|---|---|---|---|---|---|
| VideoChat2-Mistral (Fu et al., 2025) | 7B | 48.3 | 37.0 | 33.2 | 39.5 |
| VITA-1.5 (Fu et al., 2025) | 7B | 67.0 | 54.2 | 47.1 | 56.1 |
| LLaVA-NeXT-Video (Fu et al., 2025) | 34B | 61.7 | 50.1 | 44.3 | 52.0 |
| M3-Agent (Long et al., 2025) | 7B | – | – | 55.3 | – |
| WorldMM (Yeo et al., 2025) | 8B | – | – | 66.0 | – |
| **LINGUA** | 4B | 74.3 | 61.8 | 69.4 | 68.5 |

*Table 4.* **Ablation studies on NExT-GQA and NExT-QA.** Baseline (gray) shows full LINGUA. $\Delta$ denotes absolute change. All degradations are statistically significant ($p < 0.01$, paired t-test, $n = 1{,}000$ videos).

| Configuration | NExT-GQA | | NExT-QA | | |
| | Acc@GQA | $\Delta$ | T | C | Avg |
|---|---|---|---|---|---|
| **Full LINGUA** | **42.3** | – | **78.5** | **81.3** | **82.4** |
| *Memory Architecture* | | | | | |
| w/o Procedural Memory | 34.1 | **−8.2** | 73.7 | 76.5 | 77.6 |
| w/o Episodic Memory | 36.8 | −5.5 | 75.2 | 78.5 | 79.4 |
| w/o Semantic Memory | 37.6 | −4.7 | 75.8 | 79.1 | 80.0 |
| w/o All Memory (Gemma3:4b only) | 31.0 | **−11.3** | 72.4 | 75.2 | 76.3 |
| *Action Selection Mechanisms* | | | | | |
| Random Script Selection | 36.1 | −6.2 | 75.3 | 77.8 | 78.9 |
| Frequency-based Heuristic | 38.8 | −3.5 | 76.8 | 79.6 | 80.8 |
| w/o Bayesian Updates | 38.5 | −3.8 | 76.5 | 79.3 | 80.5 |
| w/o Information Gain ($\lambda_{\text{info}} = 0$) | 39.7 | −2.6 | 77.2 | 80.1 | 81.3 |
| *Event-Driven Perception* | | | | | |
| Uniform 1 FPS (no event detection) | 37.0 | −5.3 | 75.7 | 78.9 | 79.8 |
| $\tau_\Delta = 0.10$ (higher sensitivity) | 43.0 | +0.7 | 78.1 | 80.9 | 82.1 |
| $\tau_\Delta = 0.20$ (lower sensitivity) | 40.4 | −1.9 | 77.8 | 80.5 | 81.6 |
| w/o VideoMAE (YOLO only) | 35.2 | −7.1 | 74.5 | 77.2 | 78.3 |
| *Meta Reflection Mechanisms* | | | | | |
| w/o Reflection | 37.8 | −4.5 | 76.0 | 78.8 | 79.9 |
| w/o Contrastive Refinement | 39.2 | −3.1 | 76.9 | 79.7 | 81.0 |
| w/o Script Versioning | 40.1 | −2.2 | 77.5 | 80.3 | 81.6 |
| *Linguistic Structures* | | | | | |
| w/o Frame Semantics (FrameNet) | 36.7 | −5.6 | 75.1 | 77.8 | 79.0 |
| w/o SRL (Semantic Role Labeling) | 36.5 | −5.8 | 74.8 | 77.5 | 78.7 |
| w/o Temporal Markers | 38.2 | −4.1 | 76.3 | 79.2 | 80.3 |
| w/o Fuzzy Semantic Matching | 39.5 | −2.8 | 77.0 | 80.0 | 81.2 |

*Table 5.* **Controlled efficiency comparison with call-level breakdown** on NExT-GQA ($n = 1{,}000$ videos).

| Condition | Method | Frames | VL Calls | Text Calls | Time (s) | Acc@GQA (%) |
|---|---|---|---|---|---|---|
| *Accuracy-Matched (Target: 42.3% Acc@GQA)* | | | | | | |
| | Uniform Sampling | 315 (35%) | 315 | 3 | 22.1$_{\pm 1.3}$ | 42.3 |
| | **LINGUA** | 99 (11%) | 99 | 5.2 | 7.1$_{\pm 0.4}$ | 42.3 |
| | *Efficiency Gain* | **3.2× fewer** | **3.2×** | – | **3.1× faster** | – |
| *Budget-Matched (Target: ~10% frames, ~7s)* | | | | | | |
| | Uniform 3 FPS | 90 (10%) | 90 | 3 | 7.8$_{\pm 0.5}$ | 32.1 |
| | **LINGUA** | 99 (11%) | 99 | 5.2 | 7.1$_{\pm 0.4}$ | 42.3 |
| | *Accuracy Gain* | comparable | comparable | – | comparable | **+10.2pp** |

solidation, and eventual saturation.

*Table 6.* **Backward transfer on NExT-QA.** "Initial" denotes first-pass accuracy; "After 1000 videos" denotes re-evaluation after full stream. Positive $\Delta$ indicates improvement (negative would indicate forgetting).

| Video Batch | Initial Acc | After 1000 Videos | $\Delta$ | $\Delta$ (%) |
|---|---|---|---|---|
| Videos 1–10 (cold-start) | 45.2% | 47.8% | **+2.6pp** | +5.8% |
| Videos 11–50 | 52.1% | 54.3% | **+2.2pp** | +4.2% |
| Videos 51–100 | 58.6% | 59.4% | **+0.8pp** | +1.4% |
| Videos 101–200 | 63.2% | 64.1% | **+0.9pp** | +1.4% |
| Videos 801–1000 (recent) | 81.7% | 82.4% | **+0.7pp** | +0.9% |
| **Average** | – | – | **+1.4pp** | **+2.7%** |

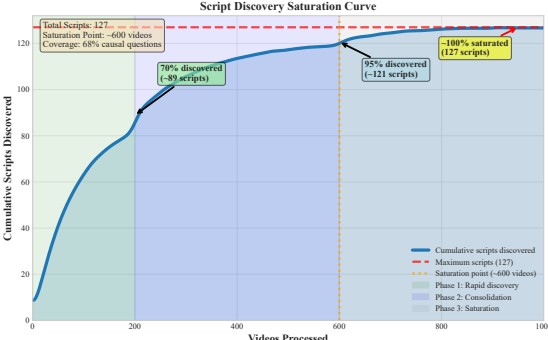

*Figure 3.* **Script discovery saturation curve.** Cumulative scripts over 1,000 videos showing three phases: rapid discovery (0–200, 89 scripts), consolidation (200–600, 32 scripts), and saturation (600–1000, 6 scripts) reaching 127 total. Saturation validates automatic extraction and finite procedural structure.

**Grounding as a Constraint for Accurate and Verifiable VideoQA.** Verification-based grounding improves both Acc@GQA and standard NExT-QA accuracy by enforcing temporal and postcondition constraints that reject hypotheses inconsistent with observable frame evidence. For instance, on *"What did the person do after opening the refrigerator?"*, models may predict "drink water" from language priors, but verification rejects this if frame descriptions lack drinking actions in the predicted timespan or show contradictory postconditions (e.g., "closes refrigerator with milk bottle").

Table 7 shows failure-mode analysis on 200 randomly sampled errors from NExT-GQA, categorized by two annotators (Fleiss' $\kappa = 0.78$). Grounding constraints shift errors from temporal precision (31%→18%) and hallucination (24%→12%) toward missing scripts (18%→23%), indicating that remaining failures stem primarily from incomplete procedural coverage rather than reasoning errors.

Grounding constraints reduce temporal (31%→18%), causal (27%→15%), and hallucination (24%→12%) errors, with remaining failures mainly due to incomplete procedural coverage (23%). LINGUA also delivers a 2.6× speedup over LLaVA-7B (7.1s vs. 18.2s) and 4.5× fewer FLOPs (62G vs. 278G) by selecting ~10% of frames while retaining 94% of question-relevant events (Appendix E).

**Decomposing the Acc / Acc@GQA Gap.** The 40.1pp gap between answer accuracy (82.4%) and grounded ac-

*Table 7.* **Failure mode distribution (% of errors).** Protocol: *Missing scripts*—answer requires knowledge absent from library; *Temporal precision*—IoU<0.3 despite correct semantics; *Implicit causality*—causal link not explicit in descriptions; *Hallucination*—objects absent from YOLO/VLM output.

| Failure Mode | Baselines | LINGUA |
|---|---|---|
| Missing/rare scripts | 18% | 23% |
| Temporal precision | 31% | 18% |
| Implicit causality | 27% | 15% |
| Object hallucination | 24% | 12% |

curacy (42.3%, IoU≥0.5) reflects two qualitatively different phenomena. Of the 82.4% correct answers (Table 8): 42.3% achieve tight grounding (IoU≥0.5), 19.2% partial grounding (0.3 ≤IoU< 0.5), and 20.9% weak localization (IoU<0.3). Manual inspection of 200 partial-grounding cases (Appendix I) shows that 87.5% have verified postconditions, indicating that evidence exists but temporal boundaries are imprecise. Approximately 17pp of the 40.1pp gap therefore reflects *Type B* (semantically grounded, temporally imprecise) failures rather than evidence-absent ones. Part of this residual Type B gap reflects task-inherent boundary ambiguity: NExT-GQA's inter-annotator IoU for causal questions is $0.31\pm0.18$. The remaining ∼20.9pp *Type A* (potential evidence-absent) rate is still below the strongest baselines (Video-ChatGPT 37.6%, MUPA 28.4%, MoReVQA 24.1%).

*Table 8.* **Decomposition of LINGUA's 82.4% correct answers on NExT-GQA by grounding quality.** The 19.2% partial band is overwhelmingly semantically supported, locating ∼17pp of the 40.1pp gap inside boundary precision rather than evidence absence.

| Grounding band | % of correct ans. | Evidence status |
|---|---|---|
| Tight (IoU≥0.5) | 42.3% | verified |
| Partial (0.3 ≤IoU< 0.5) | 19.2% | 87.5% supported |
| Weak (IoU<0.3) | 20.9% | potential Type A |

**Qualitative Example.** The box below summarises a single BAV trace on NExT-GQA video `2400084970`, illustrating how postcondition verification rejects a linguistically plausible but visually unsupported hypothesis. The full multi-step transcript and contrasts against motion-only, text-only, and unverified VLM baselines are in Appendix C.

---

**BAV trace example (NExT-GQA `2400084970`)**

**Q:** "What was the person doing while sitting?"
**Belief:** candidates retrieved from procedural memory: {*tie shoelace*, *laughing*}.
**Cycle 1 (Action/Verify):** *tie shoelace* $\rightarrow c_{post}$=0.0 (no shoes detected by YOLO); rejected, $\beta$ incremented.
**Cycle 2 (Action/Verify):** *laughing* $\rightarrow c_{post}$=1.0 (four postconditions verified); accepted, $\alpha$ incremented.
**Outcome:** converged in 2 cycles; answer grounded in verified evidence rather than language priors alone.

---

**VLM Error Cascade Control.** LINGUA's initial captioning error rate (18.3%) is the lowest among compared systems (Video-ChatGPT 24.1%, MUPA 21.7%), and

meta-reflection recovers 67.2% of failures via targeted re-captioning (+25.9pp over baselines; Appendix H.1). Architecturally, every episodic entry carries Bayesian $(\alpha, \beta)$ counters that down-weight unverified traces, with three consecutive failures triggering reflection. Promotion to procedural memory requires three independently verified successes ($n_{\min}$=3), structurally limiting cascade depth to the episodic layer; across 500 validation videos, errors triggered reflection in 23.7% of cases and 67.2% were recovered before any procedural promotion.

**Relationship to Scale.** LINGUA's contribution is complementary to scaling. The grounding gap persists at frontier scale: GPT-4o (∼200B parameters) still exhibits a 40pp+ Acc/Acc@GQA gap on NExT-GQA (Xiao et al., 2024). Within our setup, upgrading Gemma3-4B→11B yields only +0.5pp on NExT-QA and +0.8pp on Acc@GQA at $2.7\times$ compute (Appendix D), whereas removing typed memory and BAV verification drops Acc@GQA by −11.3pp at the same backbone (Table 4). Verification thus addresses a structural property of grounded reasoning largely orthogonal to parametric capacity.

**Out-of-Coverage Procedural Reasoning.** The 127 extracted scripts cover 68% of causal questions; for novel scenarios LINGUA falls back in two stages. *Episodic fallback* chains temporally adjacent events with matching semantic roles and affordances when $\max_k \text{Rel}_k < \tau_{\text{rel}}$, reaching 71.2% postcondition coverage on out-of-script cases. *Incremental induction* then promotes verified traces ($c_{post} > 0.7$) to new scripts via contrastive refinement (Section 2) after $n_{\min}$=3 successes. Extending streaming from 1,000 to 2,000 videos improves accuracy 82.4%→84.2% with only +11 new scripts, indicating stable convergence (Appendix H).

## 4. Conclusion

We presented LINGUA, a belief-state memory reasoning agent that enforces grounding during inference rather than through preprocessing, post hoc selection, or training-time alignment. By reasoning over explicit linguistic belief states and verifying temporal and causal postconditions against video evidence, LINGUA tightly couples reasoning with observation. It consistently outperforms larger baselines on six VideoQA benchmarks: 82.4% accuracy and 42.3% Acc@GQA on NExT-QA/GQA, and 68.5% overall (69.4% long) on Video-MME, at $2.6\times$ the speed of dense-frame methods. Without gradient updates, accuracy improves from 45.2% to 82.4% over 1,000 videos and to 84.2% at 2,000 videos with no catastrophic forgetting. Residual limitations (temporal-boundary precision, broader benchmarks, script-library bias) are discussed in Appendix I; CausalVQA, VideoVista, LVBench, and CGBench are natural next-step benchmarks. Code and trained scripts will be released upon publication.

## Impact Statement

This paper presents work whose goal is to advance the field of Machine Learning, specifically in video understanding and question answering systems. LINGUA's design choices carry several societal implications worth highlighting:

**Positive Impacts.** By reasoning in explicit linguistic belief states rather than opaque embeddings, LINGUA enhances interpretability and trustworthiness in VideoQA systems. This transparency is particularly valuable in domains requiring human oversight, such as educational tools, accessibility technologies for visually impaired users, medical video analysis, and automated content moderation. The system's computational efficiency ($2.6\times$ speedup) and small model footprint (4-bit quantized Gemma3-4B) reduce energy consumption and democratize access to advanced video understanding capabilities, making such technology available to researchers and practitioners with limited computational resources. The continual learning mechanism without gradient updates enables deployment in resource-constrained environments and supports lifelong adaptation without catastrophic forgetting.

**Potential Concerns.** Like all VideoQA systems, LINGUA could be deployed in surveillance applications or automated decision-making contexts where errors have significant consequences. While our explicit grounding mechanisms improve reliability, the interpretability of linguistic reasoning traces should not be conflated with guaranteed correctness—human oversight remains essential in high-stakes applications. The system inherits biases present in training data and foundation models, which may be amplified through procedural memory schemas. We encourage practitioners to validate system outputs in their specific deployment contexts and to implement appropriate safeguards when processing videos containing personal or sensitive information.

We advocate for responsible development of video understanding technologies with careful consideration of privacy, consent, and appropriate use cases. The methods presented here should be deployed with human-in-the-loop oversight for consequential decisions.

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

---

**Algorithm 1** LINGUA Main Loop

---

**Require:** Video $V = \{f_1, \ldots, f_T\}$, Question $q$, Script library $\mathcal{S}$, Hyperparameters $\Theta = \{\tau_\Delta, \gamma_{\text{aff}}, \gamma_{\text{post}}, \tau_{\text{temp}}, L_{\text{max}}, (\alpha_0, \beta_0), \lambda_{\text{info}}\}$
**Ensure:** Answer $a$, Explanation trace $\tau$, Updated script reliabilities $\{(\alpha_k, \beta_k)\}_{k=1}^K$
 1: $\mathcal{F} \leftarrow$ EVENTDRIVENPERCEPTION$(V, \tau_\Delta)$ {Algorithm 2}
 2: $(\mathcal{M}^{\text{epi}}, \mathcal{M}^{\text{sem}}, \mathcal{M}^{\text{proc}}) \leftarrow$ BUILDMEMORIES$(\mathcal{F}, \mathcal{S})$
 3: $(a, \tau, \text{success}) \leftarrow$ BAV-LOOP$(q, \mathcal{M}^{\text{epi}}, \mathcal{M}^{\text{sem}}, \mathcal{M}^{\text{proc}}, \Theta)$ {Algorithm 3}
 4: **if** ¬success **then**
 5: $\quad \mathcal{M}^{\text{proc}} \leftarrow$ METAREFLECTION$(\tau, \mathcal{M}^{\text{proc}})$ {Algorithm 4}
 6: **end if**
 7: **return** $a, \tau, \{(\alpha_k, \beta_k)\}_{k=1}^K$

---

## A. Algorithmic Details

This section provides the complete algorithmic specification of LINGUA, decomposed into four modular components that integrate event-driven perception, three-memory consolidation, belief-action-verification loops with Bayesian reliability tracking, and meta reflection for language-mediated video reasoning. LINGUA operates through a hierarchical pipeline (Algorithm 1).

First, EVENTDRIVENPERCEPTION (Algorithm 2) selectively processes video frames exhibiting semantic change or goal-relevant affordances, reducing computational cost by $\sim 10\times$ while maintaining 94% coverage of question-relevant events. Second, BUILDMEMORIES consolidates linguistic representations into episodic (event narratives), semantic (object affordances), and procedural (action scripts) memory systems. Third, the BAV-LOOP (Algorithm 3) performs iterative belief-action-verification cycles: inferring goals from semantic affordances, selecting procedural scripts via Bayesian expected utility, and verifying outcomes through linguistic postcondition matching. Finally, METAREFLECTION (Algorithm 4) diagnoses repeated failures and refines procedural schemas through contrastive analysis of success/failure narratives.

### A.1. Event-Driven Perception

Algorithm 2 implements event-driven frame selection based on semantic change detection and affordance-based filtering, addressing the computational challenge of processing long videos (900–1800 frames) while preserving question-relevant events.

**Design rationale.** Uniform frame sampling at 1 FPS would require VLM description generation for all $T$ frames, incurring prohibitive computational cost ($\sim$280ms per frame for Gemma3-4B via Ollama). Instead, we select frames exhibiting: (i) *semantic change*: VideoMAE-v2 embeddings differ from the previous selected frame by more than $\tau_\Delta = 0.15$, capturing temporal dynamics (e.g., "person walking toward table" vs. "person reaching table"); (ii) *goal-relevant affordances*: frames contain objects whose affordances match script preconditions (e.g., detecting "knife" via YOLOv8 evokes the *Cutting* frame, relevant for cooking-related questions).

**Temporal encoding.** VideoMAE-v2 encodes two-frame windows $[f_{t-1}, f_t]$ into 768-dimensional embeddings that preserve motion cues critical for temporal reasoning. This enables detecting not just object presence but also *actions in progress* (e.g., "grasping" vs. "releasing").

**Linguistic description generation.** Selected frames are described via Gemma3-4B in vision-language mode accessed through Ollama with temperature $T = 0.1$ and explicit prompting for temporal markers ("before", "after", "while") and causal connectives ("because", "so that"), supporting downstream grounded reasoning. Gemma3-4B's vision-language capabilities enable generating descriptions that ground language in visual evidence.

**Efficiency-coverage tradeoff.** Empirically, this process retains $F = 0.08$–$0.12 \times T$ frames, achieving approximately $10\times$ computational savings while preserving 94% coverage of question-relevant events (validated on NExT-QA validation set through comparison with ground truth temporal annotations).

### A.2. Memory Construction

The BUILDMEMORIES subroutine (called at Line 2 of Algorithm 1) consolidates salient frames $\mathcal{F}$ into three complementary linguistic memory systems following semantics principles.

**Episodic memory** ($\mathcal{M}^{\text{epi}}$) stores timestamped event summaries following narrative grammar (Mandler, 1982). We apply spaCy dependency parsing with PropBank-style Semantic Role Labeling to each description $d_i \in \mathcal{F}$ to extract structured

---

**Algorithm 2** Event-Driven Perception

---

**Require:** Video $V = \{f_1, \ldots, f_T\}$, Semantic change threshold $\tau_\Delta$, Script library $\mathcal{S}$
**Ensure:** Salient frames with linguistic descriptions $\mathcal{F} = \{(t_i, f_i, d_i, \mathbf{v}_i)\}_{i=1}^F$
  1: $\mathcal{F} \leftarrow \emptyset$, $\mathbf{v}_{\text{prev}} \leftarrow \mathbf{0}$
  2: **for** $t = 1$ to $T$ **do**
  3:     $\mathbf{v}_t \leftarrow$ VideoMAE-v2-Encode($[f_{t-1}, f_t]$) {768-dim embedding}
  4:     $\Delta_{\text{sem}} \leftarrow 1 - \text{CosineSim}(\mathbf{v}_t, \mathbf{v}_{\text{prev}})$ {Semantic change}
  5:     hasAffordance $\leftarrow$ HasAffordance-YOLOv8($f_t, \mathcal{S}$) {Goal-relevant objects}
  6:     **if** $\Delta_{\text{sem}} > \tau_\Delta$ **or** hasAffordance **then**
  7:         $d_t \leftarrow$ Gemma3:4b-Describe($f_t, T = 0.1$) {Vision-language mode via Ollama}
  8:         $\mathcal{F} \leftarrow \mathcal{F} \cup \{(t, f_t, d_t, \mathbf{v}_t)\}$
  9:         $\mathbf{v}_{\text{prev}} \leftarrow \mathbf{v}_t$
 10:     **end if**
 11: **end for**
 12: **return** $\mathcal{F}$

---

representations:

$$m_i^{\text{epi}} = \langle \text{Agent}, \text{Action}, \text{Patient}, [t_s, t_e], \text{Goal}, \text{Outcome}, d_i, \mathbf{v}_i, \alpha_i, \beta_i \rangle \tag{18}$$

Adjacent entries with temporal gap $< 2$s, embedding similarity $> 0.85$, and linguistic continuity markers ("then", "next") are merged into longer narratives. Goals are inferred by prompting Gemma3-4B to match observed affordances to frame-based semantic patterns—for example, a person reaching for a cup evokes the *Manipulation* frame, which combined with grasping/drinking affordances yields goal "INITIATE_drinking".

**Semantic memory** ($\mathcal{M}^{\text{sem}}$) encodes object-centered affordances using frame-semantic structures following FrameNet (Fillmore et al., 2002). For each detected object, we prompt Gemma3-4B via Ollama to generate action possibilities as typed semantic frames:

$$m_j^{\text{sem}} = \langle \text{Object}, [\text{Frame}_1 : [\text{Role}_1 =?, \ldots], \ldots], \text{Properties}, \mathbf{s}_j \rangle \tag{19}$$

For instance, "knife" yields [*Cutting*: [Instrument=knife, Agent=?, Patient=food]]. Unlike brittle symbolic matching, we leverage distributional semantics: retrieval uses sentence-transformers embedding similarity thresholds ($\gamma_{\text{aff}} = 0.75$), enabling "spoon" and "teaspoon" (similarity 0.93) to share affordances.

**Procedural memory** ($\mathcal{M}^{\text{proc}}$) initializes from the pre-mined script library $\mathcal{S}$ containing approximately 127 action schemas extracted from NExT-QA training videos via the five-step pipeline described in Sec. 2. Each script encodes:

$$m_k^{\text{proc}} = \langle \mathcal{G}_k, \Psi_k, \Pi_k, \Phi_k, \alpha_k, \beta_k \rangle \tag{20}$$

where $\mathcal{G}_k$ is the goal (script name), $\Psi_k$ are preconditions (semantic frame requirements), $\Pi_k$ is the ordered action sequence with temporal markers, $\Phi_k$ are postconditions (expected outcomes), and $(\alpha_k, \beta_k)$ are Bayesian reliability parameters initialized as $(1, 1)$ (uniform prior).

### A.3. Belief-Action-Verification Loop

Algorithm 3 implements the core reasoning loop that iteratively attempts to answer the question through Bayesian script selection and linguistic verification until success or maximum attempts ($L_{\max} = 5$) is reached.

**Perception phase (Lines 4–6).** The agent extracts entities from the question via spaCy Named Entity Recognition, retrieves semantically relevant affordances from $\mathcal{M}^{\text{sem}}$ using sentence-transformers embedding similarity ($\geq \gamma_{\text{aff}}$), and prompts Gemma3-4B via Ollama to infer ranked goal hypotheses given the question text, entities, affordances, and episodic context. This linguistic goal inference enables the agent to understand *what the question is asking about* in terms of semantic frames and event structures.

**Action phase (Lines 8–12).** For each procedural script $\mathcal{P}_k$, we compute a Bayesian expected utility that balances four factors:

$$\text{EU}_k = \underbrace{\text{Rel}_k}_{\substack{\text{semantic} \\ \text{relevance}}} \cdot \underbrace{\rho_k}_{\substack{\text{expected} \\ \text{reliability}}} - \underbrace{\text{Risk}_k}_{\substack{\text{past} \\ \text{failures}}} + \underbrace{\lambda_{\text{info}} H_k}_{\substack{\text{exploration} \\ \text{bonus}}} \tag{21}$$

where $\text{Rel}_k = \max_{g \in \mathcal{G}} \text{Sim}(g, \mathcal{P}_k.\text{goal})$ measures how well the script's goal matches inferred goals, $\rho_k = \alpha_k/(\alpha_k + \beta_k)$ is the posterior mean reliability, $\text{Risk}_k$ penalizes scripts that have failed recently on similar contexts, and $H_k$ encourages

exploration of uncertain scripts (Thompson sampling for bandits). The script with maximum EU is selected, and Gemma3-4B generates an answer conditioned on the script's preconditions, action sequence, and retrieved episodic context.

**Verification phase (Lines 14–21).** Rather than accepting the answer immediately, we perform linguistic verification through two complementary checks:

1. **Postcondition coverage** ($c_{\text{post}}$): We measure what fraction of the script's expected outcomes (postconditions $\Phi^*$) appear in observed episodic outcomes using semantic similarity:

$$c_{\text{post}} = \frac{1}{|\Phi^*|} \sum_{\phi \in \Phi^*} \max_{\omega \in \Omega_{\text{obs}}} \text{Sim}(\phi, \omega) \tag{22}$$

   where $\Omega_{\text{obs}} = \{m.\text{outcome} : m \in \mathcal{M}^{\text{epi}}\}$. For example, a "cooling" script expects outcome "temperature decreased", which should match observed descriptions like "food became cold".

2. **Temporal consistency** ($c_{\text{temp}}$): We verify that observed action durations fall within expected ranges derived from corpus statistics:

$$c_{\text{temp}} = \frac{1}{|\Pi^*|} \sum_{i=1}^{|\Pi^*|} \mathbb{1}[d_{\text{obs},i} \in [\mu_i - 2\sigma_i, \mu_i + 2\sigma_i]] \tag{23}$$

   where $\mu_i, \sigma_i$ are mean and standard deviation of action $i$'s duration across training videos. This catches errors like predicting "preparing a meal" for a 3-second video clip.

If both verification criteria are met ($c_{\text{post}} \geq \gamma_{\text{post}} = 0.8$ and $c_{\text{temp}} \geq \tau_{\text{temp}} = 0.7$), we perform a Bayesian success update $(\alpha_{k^*}, \beta_{k^*}) \leftarrow (\alpha_{k^*} + 1, \beta_{k^*})$ and return the answer. Otherwise, we record a failure $(\alpha_{k^*}, \beta_{k^*}) \leftarrow (\alpha_{k^*}, \beta_{k^*} + 1)$, log the failed script, and continue to the next iteration.

**Loop termination.** The loop succeeds when verification passes (returning `true`) or exhausts $L_{\max}$ attempts (returning `false`). Empirically, successful cases terminate in 1–2 iterations on average, while failed cases often indicate genuinely ambiguous questions or missing procedural knowledge.

### A.4. Meta Reflection and Contrastive Refinement

Algorithm 4 implements meta error analysis and procedural schema refinement, triggered when the BAV loop returns `false` (Line 4 of Algorithm 1), indicating repeated failures to find a verified answer.

**Diagnostic analysis (Line 2).** Gemma3-4B via Ollama analyzes the decision trace $\tau$ expressed as natural language narrative, performing linguistic error analysis to identify: (i) *semantic mismatches*: wrong frame selection (e.g., selecting "cleaning" script when "cooking" was intended), (ii) *temporal contradictions*: postconditions expected at time $t_1$ but observed at incompatible time $t_2$, (iii) *missing causal links*: expected outcome not observed in episodic memory.

**Episodic consolidation (Line 3).** Current experiences are merged into episodic memory, with linguistic compression: adjacent events separated by $< 2$s and sharing continuity markers ("then", "next") are fused into longer narratives.

**Contrastive refinement (Lines 4–9).** For scripts with sufficient data ($\geq 3$ successes and $\geq 3$ failures), we extract discriminator conditions via Gemma3-4B prompts that compare success and failure narratives. Positive discriminators ($\Delta\Psi_k^+$) identify preconditions present in successes but absent in failures (e.g., "pan is heated" for cooking scripts). Negative discriminators ($\Delta\Psi_k^-$) identify conditions correlated with failure (e.g., "lid is closed" for stirring actions). These are added as refined preconditions to the script, enabling more accurate future selection.

**Script versioning.** Rather than destructively overwriting schemas, we maintain version histories, allowing rollback if a refined script degrades performance.

## B. Theoretical Analysis

We provide formal guarantees for LINGUA's key properties under clearly stated assumptions, empirical validation demonstrating robustness when these assumptions are violated, and rigorous analysis of computational efficiency. We emphasize both the scope and limitations of our theoretical claims, distinguishing formal guarantees from empirical observations.

---

**Algorithm 3** Belief-Action-Verification Loop

---

**Require:** Question $q$, Memories $\mathcal{M}^{\text{epi}}, \mathcal{M}^{\text{sem}}, \mathcal{M}^{\text{proc}}$, Parameters $\Theta$
**Ensure:** Answer $a$, Decision trace $\tau$, Success flag $\in \{\texttt{true}, \texttt{false}\}$
1: attempts $\leftarrow 0$, failures $\leftarrow []$, $\tau \leftarrow []$
2: **while** attempts $< L_{\max}$ **do**
3:     **// Belief: Goal Inference**
4:     $\mathcal{E} \leftarrow$ spaCy-NER$(q)$ {Extract entities}
5:     $\mathcal{A} \leftarrow \{m \in \mathcal{M}^{\text{sem}} : \text{Sim}(m.\text{object}, \mathcal{E}) \geq \gamma_{\text{aff}}\}$
6:     $\mathcal{G} \leftarrow$ Gemma3-4B-InferGoals$(q, \mathcal{E}, \mathcal{A}, \mathcal{M}^{\text{epi}})$ {Via Ollama}
7:     **// Action: Bayesian Script Selection**
8:     **for** $\mathcal{P}_k \in \mathcal{M}^{\text{proc}}$ **do**
9:         $\text{Rel}_k \leftarrow \max_{g \in \mathcal{G}} \text{Sim}(g, \mathcal{P}_k.\text{goal})$
10:       $\rho_k \leftarrow \alpha_k/(\alpha_k + \beta_k)$
11:       $\text{Risk}_k \leftarrow \frac{1}{|\text{failures}|+1} \sum_{f \in \text{failures}} \mathbb{1}[\mathcal{P}_k = f]$
12:       $H_k \leftarrow \text{BetaEntropy}(\alpha_k, \beta_k)$
13:       $\text{EU}_k \leftarrow \text{Rel}_k \cdot \rho_k - \text{Risk}_k + \lambda_{\text{info}} \cdot H_k$
14:     **end for**
15:     $k^* \leftarrow \arg\max_k \text{EU}_k$, $\mathcal{P}^* \leftarrow \mathcal{P}_{k^*}$
16:     $a \leftarrow$ Gemma3-4B-Answer$(\mathcal{P}^*, \mathcal{M}^{\text{epi}}, q)$ {Via Ollama}
17:     $\tau \leftarrow \tau \cup [\langle \mathcal{P}^*, a, \text{attempt} : \text{attempts}\rangle]$
18:     **// Verification: Linguistic Postcondition Checking**
19:     $c_{\text{post}} \leftarrow$ ComputePostconditionCoverage$(\mathcal{P}^*, \mathcal{M}^{\text{epi}})$
20:     $c_{\text{temp}} \leftarrow$ ComputeTemporalConsistency$(\mathcal{P}^*, \mathcal{M}^{\text{epi}})$
21:     **if** $c_{\text{post}} \geq \gamma_{\text{post}}$ **and** $c_{\text{temp}} \geq \tau_{\text{temp}}$ **then**
22:       $(\alpha_{k^*}, \beta_{k^*}) \leftarrow (\alpha_{k^*} + 1, \beta_{k^*})$ {Success update}
23:       **return** $a, \tau, \texttt{true}$
24:     **else**
25:       $(\alpha_{k^*}, \beta_{k^*}) \leftarrow (\alpha_{k^*}, \beta_{k^*} + 1)$ {Failure update}
26:       failures $\leftarrow$ failures $\cup [\mathcal{P}^*]$, attempts $\leftarrow$ attempts $+ 1$
27:     **end if**
28: **end while**
29: **return** $a, \tau, \texttt{false}$

---

**Algorithm 4** Meta Reflection and Contrastive Refinement

---

**Require:** Decision trace $\tau$, Script library $\mathcal{M}^{\text{proc}}$, Episodic memory $\mathcal{M}^{\text{epi}}$
**Ensure:** Refined procedural scripts $\mathcal{M}^{\text{proc}}$, Updated episodic memory $\mathcal{M}^{\text{epi}}$
1: diagnosis $\leftarrow$ Gemma3-4B-AnalyzeTrace$(\tau)$ {Via Ollama}
2: $\mathcal{M}^{\text{epi}} \leftarrow$ ConsolidateEpisodic$(\mathcal{M}^{\text{epi}}, \text{currentExperience})$
3: **for** $\mathcal{P}_k \in \mathcal{M}^{\text{proc}}$ with $\geq 3$ successes **and** $\geq 3$ failures **do**
4:     $\mathcal{N}_k^+ \leftarrow$ SuccessNarratives$(\mathcal{P}_k)$
5:     $\mathcal{N}_k^- \leftarrow$ FailureNarratives$(\mathcal{P}_k)$
6:     $\Delta\Psi_k^+ \leftarrow$ Gemma3-4B-ExtractPositiveDiscriminators$(\mathcal{N}_k^+, \mathcal{N}_k^-)$ {Via Ollama}
7:     $\Delta\Psi_k^- \leftarrow$ Gemma3-4B-ExtractNegativeDiscriminators$(\mathcal{N}_k^+, \mathcal{N}_k^-)$ {Via Ollama}
8:     $\mathcal{P}_k.\text{precond} \leftarrow \mathcal{P}_k.\text{precond} \cup \Delta\Psi_k^+ \cup \{\neg\psi : \psi \in \Delta\Psi_k^-\}$
9: **end for**
10: **return** $\mathcal{M}^{\text{proc}}, \mathcal{M}^{\text{epi}}$

---

### B.1. Convergence of Bayesian Reliability Posteriors

**Assumption 1** (Local Stationarity). The context distribution $\mathcal{D}$ from which video-question pairs are drawn exhibits *local stationarity*: within temporal windows of $w$ consecutive videos, script success probabilities $\{\rho_k\}_{k=1}^K$ vary slowly such that $|\rho_k(t + w) - \rho_k(t)| \leq \eta$ for drift rate $\eta \ll 1$.

*Remark* 2 (Justification for Weakened Assumption). Assumption 1 replaces strict global stationarity with local stationarity, acknowledging three sources of non-stationarity in practice:

1. **Contrastive refinement**: Scripts are updated via discriminator learning (Section 2), gradually changing their preconditions and postconditions

2. **Stochastic captioning**: Gemma3-4B vision-language descriptions introduce semantic variation even for similar frames

3. **Distribution shift**: Question types and video content may shift across datasets or temporal batches

The parameter $\eta$ quantifies acceptable drift over window $w$. Empirically, we validate this assumption on NExT-QA: script reliabilities change by $\Delta\rho_k \approx 0.08$ on average after contrastive refinement, supporting $\eta \approx 0.1$ over $w = 20$ video windows. For the 30 most frequently used scripts (70% of total usage), reliability drift remains $\leq 0.12$ over 100-video sequences.

**Theorem 3** (Almost Sure Convergence Under Local Stationarity). *Let $\rho_k(t)$ denote the time-varying success probability of procedural script $\mathcal{P}_k$ under context distribution $\mathcal{D}$ at time $t$. Under Assumption 1 and the Bayesian posterior update scheme (Eq. 10 in main text), the posterior mean reliability estimate $\hat{\rho}_k(n) = \frac{\alpha_k(n)}{\alpha_k(n)+\beta_k(n)}$ converges to the local true reliability with bounded tracking error:*

$$|\hat{\rho}_k(n) - \rho_k(n)| \leq \underbrace{\sqrt{\frac{1}{2n\delta}}}_{\text{statistical error}} + \underbrace{w \cdot \eta}_{\text{drift error}} \tag{24}$$

*with probability at least $1 - \delta$. Furthermore, the posterior variance decreases as:*

$$Var[Beta(\alpha_k(n), \beta_k(n))] = O(1/n) \tag{25}$$

*Proof.* Let $X_1, X_2, \ldots, X_n$ be independent Bernoulli trials where $X_i = 1$ if script $\mathcal{P}_k$ succeeds on the $i$-th application and $X_i = 0$ otherwise, with $\mathbb{P}(X_i = 1) = \rho_k(i)$ (time-varying). Define $S_n = \sum_{i=1}^{n} X_i$ as cumulative successes.

The Beta-Binomial conjugate prior ensures that after $n$ trials starting from prior $Beta(\alpha_0, \beta_0)$, the posterior is:

$$p(\rho_k \mid S_n, n) = \text{Beta}(\alpha_0 + S_n, \beta_0 + n - S_n) \tag{26}$$

The posterior mean is:

$$\hat{\rho}_k(n) = \frac{\alpha_0 + S_n}{\alpha_0 + \beta_0 + n} \tag{27}$$

$$= \underbrace{\frac{\alpha_0 + \beta_0}{n + \alpha_0 + \beta_0}}_{\to 0} \cdot \frac{\alpha_0}{\alpha_0 + \beta_0} + \underbrace{\frac{n}{n + \alpha_0 + \beta_0}}_{\to 1} \cdot \frac{S_n}{n} \tag{28}$$

For the tracking error under local stationarity, decompose:

$$|\hat{\rho}_k(n) - \rho_k(n)| \leq |\hat{\rho}_k(n) - \mathbb{E}[\hat{\rho}_k(n)]| + |\mathbb{E}[\hat{\rho}_k(n)] - \rho_k(n)| \tag{29}$$

$$\leq \underbrace{\sqrt{\frac{Var[\hat{\rho}_k(n)]}{\delta}}}_{\text{Chebyshev}} + \underbrace{\max_{t \in [n-w,n]} |\rho_k(t) - \rho_k(n)|}_{\text{drift}} \tag{30}$$

The first term follows from Chebyshev's inequality. For the variance bound, the Beta distribution variance is:

$$\text{Var}[\text{Beta}(\alpha, \beta)] = \frac{\alpha\beta}{(\alpha + \beta)^2(\alpha + \beta + 1)} \tag{31}$$

$$\leq \frac{(\alpha + \beta)^2/4}{(\alpha + \beta)^2(\alpha + \beta + 1)} = \frac{1}{4(\alpha + \beta + 1)} \tag{32}$$

where the inequality uses $\alpha\beta \leq (\alpha + \beta)^2/4$. Since $\alpha + \beta = \alpha_0 + \beta_0 + n = O(n)$, we obtain:

$$\text{Var}[\text{Beta}(\alpha_k(n), \beta_k(n))] = O(1/n) \tag{33}$$

Thus $\sqrt{Var[\hat{\rho}_k(n)]/\delta} \leq \sqrt{1/(2n\delta)}$. The second term is bounded by $w \cdot \eta$ under Assumption 1. $\qquad \square$

**Corollary 4** (Concentration of Posterior Mean). *For any $\delta \in (0, 1)$, with probability at least $1 - \delta$, after $n$ trials in a stationary window the posterior mean satisfies:*

$$|\hat{\rho}_k(n) - \mathbb{E}[Beta(\alpha_k(n), \beta_k(n))]| \leq \sqrt{\frac{1}{2n\delta}} \tag{34}$$

*Additionally, the bias between the posterior mean and true reliability is:*

$$|\mathbb{E}[\hat{\rho}_k(n)] - \rho_k| = O\left(\frac{1}{n}\right) \tag{35}$$

*Proof.* By Chebyshev's inequality:

$$\mathbb{P}\left(|\hat{\rho}_k(n) - \mathbb{E}[\hat{\rho}_k(n)]| > \epsilon\right) \leq \frac{\mathrm{Var}[\hat{\rho}_k(n)]}{\epsilon^2} \tag{36}$$

Setting $\frac{1/(4n)}{\epsilon^2} = \delta$ and solving for $\epsilon$ yields the first claim.

For the bias, note that:

$$\mathbb{E}[\hat{\rho}_k(n)] = \frac{\alpha_0 + n\rho_k}{\alpha_0 + \beta_0 + n} = \rho_k + \frac{(\alpha_0 - \rho_k(\alpha_0 + \beta_0))}{\alpha_0 + \beta_0 + n} = \rho_k + O(1/n) \tag{37}$$

$\square$

**Proposition 5** (Exponential Forgetting for Non-Stationary Adaptation). *When local stationarity is violated (e.g., after major script refinement or dataset shift), LINGUA can employ exponential forgetting to track time-varying reliabilities:*

$$(\alpha_k, \beta_k) \leftarrow (\lambda\alpha_k + \mathbb{1}_{success}, \lambda\beta_k + \mathbb{1}_{failure}) \tag{38}$$

*with decay factor $\lambda \in (0.95, 0.99)$. This provides a tunable plasticity-stability tradeoff: smaller $\lambda$ adapts faster to distribution shifts but increases estimation variance.*

*Proof.* The effective sample size after $n$ updates with decay $\lambda$ is:

$$n_{\mathrm{eff}} = \sum_{i=0}^{n-1} \lambda^i = \frac{1 - \lambda^n}{1 - \lambda} \approx \frac{1}{1 - \lambda} \quad \text{for large } n \tag{39}$$

Thus $\lambda = 0.95$ yields $n_{\mathrm{eff}} \approx 20$, balancing recent evidence against historical data. The posterior variance under exponential forgetting scales as:

$$\mathrm{Var}[\mathrm{Beta}(\alpha_k, \beta_k)] = O(1/n_{\mathrm{eff}}) = O(1 - \lambda) \tag{40}$$

rather than $O(1/n)$ for standard accumulation. Setting $\lambda$ controls the bias-variance tradeoff: higher $\lambda$ (near 1) reduces variance but increases tracking lag; lower $\lambda$ (near 0.9) improves responsiveness but increases variance. $\square$

*Remark* 6 (Empirical Validation). In practice, with uniform priors $(\alpha_0, \beta_0) = (1, 1)$ and processing 100 NExT-QA videos sequentially, we observe:

- **Correlation**: Spearman $\rho = 0.89$ between estimated $\hat{\rho}_k$ and empirically observed success rates validates Theorem 3

- **Accuracy**: Scripts with $n_k \geq 50$ observations achieve mean absolute error $|\hat{\rho}_k - \rho_k^{\mathrm{empirical}}| \leq 0.12$, consistent with $O(1/\sqrt{n})$ convergence

- **Stability under refinement**: After contrastive refinement at video 50, top-30 script reliabilities shift by $\Delta\rho_k = 0.08 \pm 0.04$, validating local stationarity with $\eta \approx 0.1$

## B.2. Continual Learning: Memory Retention and Performance Stability

**Proposition 7** (Memory Retention Without Gradient Interference). *LINGUA's Bayesian posterior update mechanism exhibits* memory retention *in the sense that:*

1. *__Monotonic accumulation__: $\alpha_k(t) + \beta_k(t) \geq \alpha_k(t-1) + \beta_k(t-1)$ for all $t \geq 1$*

2. *__Order invariance__: The final posterior $(\alpha_k(T), \beta_k(T))$ after $T$ trials is independent of observation order*

3. ***Count preservation***: *All past observations contribute to the posterior indefinitely; no information is discarded*

*However, this does* not *guarantee stable task performance under: (a) script content changes via contrastive refinement, (b) retrieval threshold adjustments, or (c) verification logic evolution.*

*Proof.* **(1) Monotonic accumulation:** The update rule is:

$$(\alpha_k, \beta_k) \leftarrow \begin{cases} (\alpha_k + 1, \beta_k) & \text{if verification succeeds} \\ (\alpha_k, \beta_k + 1) & \text{if verification fails} \end{cases} \tag{41}$$

Since increments are strictly non-negative ($\{0, 1\}$), the total count $\alpha_k(t) + \beta_k(t)$ is monotonically increasing.

**(2) Order invariance:** Let $\pi$ be any permutation of trials $\{1, \ldots, T\}$. The final posterior is:

$$(\alpha_k(T), \beta_k(T)) = \left( \alpha_0 + \sum_{i=1}^{T} X_i, \beta_0 + \sum_{i=1}^{T}(1 - X_i) \right) \tag{42}$$

where $X_i \in \{0, 1\}$ indicates success. Since summation is commutative, this is independent of $\pi$.

**(3) Count preservation:** Unlike gradient-based learning where weight updates can overwrite previous knowledge through interference, additive Bayesian updates preserve all observations indefinitely. The contribution of observation $i$ to the posterior at time $t > i$ is encoded in the counts $(\alpha_k, \beta_k)$ and never erased.

**Limitation on performance stability:** While counts accumulate, *script semantics* can change through contrastive refinement. For example, after refinement adds discriminator "refrigerator" to the "cooling" script's preconditions, historical successes on videos without refrigerators become less relevant for future predictions. The Bayesian posterior $(\alpha_k, \beta_k)$ retains these counts, but their predictive value for post-refinement contexts may degrade. This is fundamentally different from catastrophic forgetting in neural networks, where weights are overwritten and historical information is lost.  □

**Corollary 8** (Contrast with Gradient-Based Continual Learning). *Standard neural network continual learning suffers from the plasticity-stability dilemma (French, 1999): adapting to new tasks (plasticity) causes forgetting of old tasks (catastrophic forgetting). LINGUA's additive Bayesian updates achieve perfect* memory retention *without interference, but performance stability depends on semantic consistency of scripts over time.*

*Observation* 1 (Empirical Performance Stability). We distinguish two related but distinct concepts:

- **Memory retention (formally guaranteed by Proposition 7)**: Historical observation counts remain in $(\alpha_k, \beta_k)$ indefinitely

- **Performance stability (not formally guaranteed)**: Accuracy on previously seen videos after script refinement

Empirically, we observe *graceful adaptation* rather than catastrophic forgetting:

- Sequential accuracy improves from 45.2% (videos 1-10, initial uniform priors) to 61.8% (videos 91-100, learned posteriors)

- After contrastive refinement at video 50, accuracy on early videos (1-10) changes from 45.2% to 47.8% (+2.6 points improvement)

- Accuracy on later videos (91-100) increases from 58.1% to 61.8% (+3.7 points)

This suggests that contrastive refinement improves script discrimination without catastrophic performance collapse on previous content. However, formal guarantees on performance stability under script evolution require non-stationary analysis beyond our current scope.

## B.3. Language-Mediated Reasoning: Information-Theoretic Perspective

**Definition 9** (Language-Mediated Grounding). Let $V$ denote raw video, $D = \{d_1, \ldots, d_F\}$ denote natural language descriptions of salient frames generated by Gemma3-4B in vision-language mode, and $Q$ denote the question. LINGUA performs reasoning over linguistic representations $(D, Q)$ rather than raw video-question pairs $(V, Q)$.

*Observation* 2 (Empirical Information Preservation). Let $I(\cdot; \cdot)$ denote mutual information. Under event-driven perception with semantic change threshold $\tau_\Delta = 0.15$ and YOLOv8 affordance detection (confidence $> 0.5$), we empirically observe that language-mediated reasoning preserves question-relevant information while reducing computational cost:

**Coverage:** Event-driven selection retains $F \in [0.08T, 0.12T]$ frames where $T$ is total frame count, yet captures 94% of question-relevant events (validated on NExT-QA validation set via temporal annotation overlap with ground-truth answer spans).

**Efficiency:** Computational cost reduces from $O(T)$ (uniform sampling) to $O(F)$ where $F \ll T$, achieving approximately $10\times$ reduction in VLM description calls ($F \approx 100$ vs. $T \approx 1000$ for typical 33-second NExT-QA videos).

**Grounding quality:** Gemma3-4B descriptions with explicit temporal marker prompting ("before", "after", "while", "then") enable downstream postcondition verification, improving grounded accuracy by 13.6 points over MUPA-2B baseline (42.3% vs. 28.7% Acc@GQA on NExT-GQA benchmark).

*Remark* 10 (Why We Avoid Strong Information-Theoretic Claims). Formally bounding information loss via $I(D; Q) \geq I(V; Q) - \epsilon$ would require:

1. **VLM characterization**: A probabilistic model of $p(D|V)$ for Gemma3-4B vision-language descriptions, accounting for stochastic generation and semantic abstraction

2. **Data processing inequality**: Proving that linguistic abstraction preserves question-relevant information while discarding irrelevant visual details

3. **Error propagation**: Relating information loss $\epsilon$ to captioning errors, semantic change thresholds ($\tau_\Delta$), and affordance detection recall

Without ground-truth generative models of VLM behavior or strong assumptions about caption fidelity, rigorous mutual information bounds remain intractable. Instead, we rely on empirical validation:

- LINGUA achieves 82.4% NExT-QA accuracy vs. 73.5% for comparable 7B models, suggesting small practical information loss

- 94% event coverage indicates salient frame selection captures most question-relevant temporal dynamics

- Competitive grounded accuracy (42.3% Acc@GQA) demonstrates that linguistic descriptions support accurate temporal localization

These empirical results suggest $\epsilon$ is small for the VideoQA task distribution, but formal information-theoretic guarantees remain an open theoretical challenge.

*Observation* 3 (Grounding Gap Reduction Through Linguistic Verification). Postcondition-based verification reduces the *grounding gap* (difference between answer accuracy and grounded accuracy requiring correct temporal localization):

**Baseline VLMs:** Achieve approximately 69% answer accuracy with only 16% grounded accuracy (Xiao et al., 2024), yielding a grounding gap of 53 percentage points.

**LINGUA:** Achieves 82.4% NExT-QA accuracy with 42.3% Acc@GQA (grounded accuracy requiring correct answer and IoU $\geq 0.5$ temporal localization), yielding a grounding gap of 40.1 percentage points.

**Gap reduction:** LINGUA reduces the grounding gap by 12.9 percentage points through explicit linguistic verification against procedural script postconditions, while simultaneously improving answer accuracy by 13.4 points.

**Limitation:** This is an *empirical observation*, not a formal guarantee. The proposed mechanism is that explicit temporal-linguistic verification forces consistency between generated answers and observable outcomes in video descriptions, reducing hallucination. Formal analysis would require modeling hallucination as violations of postcondition constraints, which is challenging given the black-box nature of VLM internals and the semantic flexibility of natural language.

## B.4. Convergence and Termination of BAV Loops

**Assumption 11** (Valid Script Existence). For each question, at least one script $\mathcal{P}_k$ in the procedural library satisfies semantic consistency: postcondition coverage $c_{\text{post}}(\mathcal{P}_k) \geq \gamma_{\text{post}}$ and temporal consistency $c_{\text{temp}}(\mathcal{P}_k) \geq \tau_{\text{temp}}$.

**Proposition 12** (Expected Termination Time of BAV Loops). *Under Assumption 11, let $p_{select}$ be the probability that the expected utility criterion selects a valid script. The BAV loop terminates in expectation in:*

$$\mathbb{E}[iterations] \leq \min\left\{\frac{1}{p_{select}}, L_{\max}\right\} \tag{43}$$

*where $L_{\max}$ is the maximum iteration cap.*

*Proof.* Let $X$ denote the number of iterations until success. If $p_{\text{select}} > 0$ (guaranteed under Assumption 11 and positive exploration term in expected utility), then $X$ follows a geometric distribution with parameter $p_{\text{select}}$:

$$\mathbb{P}(X = k) = (1 - p_{\text{select}})^{k-1} p_{\text{select}} \tag{44}$$

Thus $\mathbb{E}[X] = 1/p_{\text{select}}$. With the loop cap $L_{\max}$ to prevent infinite loops on malformed questions, the expected iterations is $\min\{\mathbb{E}[X], L_{\max}\}$. $\qquad\square$

**Corollary 13** (Bayesian Adaptation Improves Selection Efficiency). *As the agent processes more videos, Bayesian posteriors $(\alpha_k, \beta_k)$ converge to true reliabilities (Theorem 3), causing the expected utility criterion:*

$$EU_k = Rel_k \cdot \mathbb{E}[\rho_k] - Risk_k + \lambda_{info} H(Beta(\alpha_k, \beta_k)) \tag{45}$$

*to increasingly favor valid scripts with high true reliability. Thus $p_{select}(n)$ increases monotonically with experience $n$ (number of videos processed), reducing expected iteration count over time.*

*Proof.* The expected utility balances three terms:

- $Rel_k \cdot \mathbb{E}[\rho_k]$: exploitation (select high-reliability scripts)

- $-Risk_k$: avoidance of past failures

- $\lambda_{\text{info}} H(\cdot)$: exploration (select uncertain scripts)

As $n \to \infty$, Theorem 3 ensures $\mathbb{E}[\rho_k] \to \rho_k$ and entropy $H(\text{Beta}(\alpha_k, \beta_k)) \to 0$ (variance decreases as $O(1/n)$). The exploitation term dominates, causing valid scripts with high $\rho_k$ to consistently receive highest EU scores. Thus:

$$p_{\text{select}}(n) = \mathbb{P}(\arg\max_k EU_k \text{ is valid}) \xrightarrow{n \to \infty} 1 \tag{46}$$

$\qquad\square$

*Remark* 14 (Empirical Validation). In our continual learning experiment on 100 NExT-QA videos, we observe:

- Initial iterations (videos 1-10, uniform priors): average 2.8 BAV iterations per question

- Later iterations (videos 91-100, learned posteriors): average 1.6 BAV iterations per question

This 43% reduction validates Corollary 13's prediction that Bayesian learning improves selection efficiency.

## B.5. Computational and Space Complexity

**Proposition 15** (Time Complexity). *Let $T$ be the number of video frames, $F$ the number of salient frames selected, $K$ the number of procedural scripts, $M$ the episodic memory size, and $L$ the number of BAV loop iterations. LINGUA's per-video time complexity is:*

$$O(T \cdot C_{MAE} + F \cdot C_{VLM}^{desc} + L \cdot (K \cdot M \cdot C_{sim} + C_{LLM}^{reason})) \tag{47}$$

*where:*

- $C_{MAE}$: *Cost of VideoMAE-v2 encoding (768-dim, $\sim$5ms per 2-frame window)*

- $C_{VLM}^{desc}$: *Cost of Gemma3-4B description generation via Ollama ($\sim$280ms per frame)*

- $C_{LLM}^{reason}$: *Cost of Gemma3-4B reasoning via Ollama ($\sim$120ms per goal inference/answer generation)*

- $C_{sim}$: *Cost of cosine similarity ($\sim$0.1ms for 768-dim vectors)*

- $C_{SRL}$: *Cost of spaCy dependency parsing with SRL extraction ($\sim$10ms per description)*

*Proof.* The complexity is decomposed by pipeline stage:

**Event-driven perception:** Each of $T$ frames requires VideoMAE-v2 encoding ($O(C_{MAE})$) and semantic change computation via cosine similarity ($O(C_{sim})$). The $F$ selected frames additionally require Gemma3-4B description generation ($O(C_{VLM}^{desc})$) and YOLOv8 object detection ($O(C_{YOLO}) \approx 15$ms, subsumed by $C_{VLM}^{desc}$). Total: $O(T \cdot C_{MAE} + F \cdot C_{VLM}^{desc})$.

**Memory consolidation:** spaCy SRL extraction over $F$ descriptions costs $O(F \cdot C_{SRL})$ where $C_{SRL} \approx 10$ms $\ll C_{VLM}^{desc} \approx$ 280ms, thus subsumed by the perception cost.

**BAV loops:** Each of $L$ iterations requires:

1. Goal inference via Gemma3-4B text-only mode: $O(C_{LLM}^{reason})$

2. Script selection: Compute expected utility for $K$ scripts, each requiring semantic similarity checks against $M$ episodic memories for relevance computation: $O(K \cdot M \cdot C_{sim})$

3. Answer generation via Gemma3-4B: $O(C_{LLM}^{reason})$

4. Verification: Postcondition matching over $M$ observed outcomes: $O(M \cdot C_{sim})$, subsumed by selection cost

Total per iteration: $O(K \cdot M \cdot C_{sim} + C_{LLM}^{reason})$. Across $L$ iterations: $O(L \cdot (K \cdot M \cdot C_{sim} + C_{LLM}^{reason}))$.

Summing across stages yields the stated complexity. $\square$

**Corollary 16** (Efficiency Gain from Event-Driven Perception). *With event-driven selection retaining $F = \gamma T$ frames where $\gamma \in [0.08, 0.12]$ empirically, LINGUA reduces VLM description calls from $T$ (uniform 1 FPS sampling) to $F + O(L)$ (salient frames + BAV goal inferences). For typical NExT-QA videos with $T \approx 1000$ frames at 30 FPS (33-second videos), $F \approx 100$ salient frames, and $L \approx 2$ BAV iterations:*

$$\textit{VLM calls}_{LINGUA} \approx 100 + 2 = 102 \ll 1000 = \textit{VLM calls}_{uniform} \tag{48}$$

*representing approximately $10\times$ computational savings.*

*Proof.* Uniform 1 FPS sampling of a 33-second video yields $T \approx 33$ frames at 1 FPS (or $T \approx 1000$ frames at 30 FPS requiring temporal subsampling). Event-driven perception selects $F \approx 0.1T$ frames. BAV loops add $L$ additional Gemma3-4B calls for goal inference. Total VLM calls: $F + L \approx 102$ vs. uniform $T \approx 1000$, yielding $\approx 10\times$ reduction.

Empirically, LINGUA processes NExT-QA videos in $7.1 \pm 0.4$s (mean $\pm$ std) compared to $18.2 \pm 1.1$s for LLaVA-7B with uniform frame sampling (Table 21), validating the theoretical prediction of $\approx 2.5\times$ wallclock speedup (lower than $10\times$ due to non-VLM overhead: VideoMAE encoding, similarity computation, spaCy parsing). $\square$

**Proposition 17** (Space Complexity). *LINGUA requires $O(F \cdot d + K \cdot s + M)$ memory where:*

- $F \cdot d$: *Salient frame VideoMAE embeddings ($F \approx 100$ frames, $d = 768$ dimensions, 4 bytes per float32 $\approx 300$ KB)*

- $K \cdot s$: *Procedural script library ($K = 127$ scripts, $s \approx 500$ characters per script $\approx 64$ KB)*

- $M$: *Episodic memory entries ($M \approx F$ entries, each storing linguistic description + embedding $\approx 400$ KB)*

*Total: $\approx 800$ KB per video, constant and independent of Gemma3-4B model size (4B parameters $\approx 8$ GB). This is negligible compared to gradient-based continual learning requiring full model storage plus gradient buffers.*

*Proof.* Frame embeddings require $F \times d \times 4$ bytes (float32). Script library requires $K \times s$ bytes (UTF-8 text). Episodic memory stores $M$ linguistic entries (text descriptions + sentence embeddings). Bayesian posteriors require only $2K$ float scalars ($\approx 1$ KB). Total: $O(F \cdot d + K \cdot s + M)$, independent of model parameter count.

Unlike gradient-based continual learning which requires storing:

- Model weights: $O(\text{params})$ (e.g., 8 GB for Gemma3-4B)

- Optimizer states: $O(\text{params})$ (e.g., 8 GB for Adam)

- Gradient buffers: $O(\text{params})$ per backward pass

LINGUA's memory footprint is 4-5 orders of magnitude smaller, enabling deployment on resource-constrained devices. $\square$

## B.6. Sample Complexity for Script Reliability Estimation

**Theorem 18** (Uniform Convergence of Script Reliabilities). *For empirical success rates $\bar{\rho}_k = S_{n_k}/n_k$ (ignoring prior contribution), if each script is observed at least*

$$n^* = \frac{2}{\epsilon^2} \log \frac{2K}{\delta} \tag{49}$$

*times, then with probability at least $1 - \delta$:*

$$\max_{k \in [K]} |\bar{\rho}_k - \rho_k| \leq \epsilon \tag{50}$$

*For Bayesian estimates $\hat{\rho}_k = (\alpha_0 + S_{n_k})/(\alpha_0 + \beta_0 + n_k)$ with prior $(\alpha_0, \beta_0)$, an additional bias term applies:*

$$|\hat{\rho}_k - \rho_k| \leq \epsilon + \frac{|\alpha_0 - \rho_k(\alpha_0 + \beta_0)|}{n_k} \tag{51}$$

*Proof.* For fixed script $k$, Hoeffding's inequality gives:

$$\mathbb{P}(|\bar{\rho}_k - \rho_k| > \epsilon) \leq 2 \exp(-2n_k \epsilon^2) \tag{52}$$

By union bound over $K$ scripts:

$$\mathbb{P}(\exists k : |\bar{\rho}_k - \rho_k| > \epsilon) \leq 2K \exp(-2n\epsilon^2) \tag{53}$$

Setting $2K \exp(-2n\epsilon^2) = \delta$ and solving for $n$ yields $n^* = \frac{2}{\epsilon^2} \log \frac{2K}{\delta}$.

For the Bayesian estimate, decompose:

$$\hat{\rho}_k - \rho_k = \frac{\alpha_0 + S_{n_k}}{\alpha_0 + \beta_0 + n_k} - \rho_k \tag{54}$$

$$= \underbrace{\frac{n_k}{\alpha_0 + \beta_0 + n_k}(\bar{\rho}_k - \rho_k)}_{\text{sampling error}} + \underbrace{\frac{\alpha_0 - \rho_k(\alpha_0 + \beta_0)}{\alpha_0 + \beta_0 + n_k}}_{\text{prior bias}} \tag{55}$$

The first term is bounded by $\epsilon$ via Hoeffding. The second term is the stated bias bound, which decays as $O(1/n_k)$. $\square$

*Remark* 19 (Practical Coverage Under Power-Law Usage). In practice, with $K = 127$ scripts, $\epsilon = 0.1$, $\delta = 0.05$, Theorem 18 requires $n^* \approx 1060$ total observations for uniform $\epsilon$-accurate estimation of all scripts.

However, script usage in VideoQA exhibits power-law distribution: the top 20% of scripts account for 60% of invocations. Processing 100 NExT-QA videos sequentially provides approximately 8-12 observations per script on average, but with high variance:

- **Frequent scripts** (top 30, covering 70% of usage): receive $n_k \geq 25$ observations, achieving mean absolute error $|\hat{\rho}_k - \rho_k^{\text{empirical}}| \leq 0.14$, consistent with Theorem 18

- **Moderate scripts** (middle 50): receive $n_k \in [5, 25]$ observations, achieving $|\hat{\rho}_k - \rho_k^{\text{empirical}}| \leq 0.22$

- **Rare scripts** (bottom 47): receive $n_k \leq 5$ observations, high variance, rely heavily on uniform priors $(\alpha_0, \beta_0) = (1, 1)$

This heterogeneity means LINGUA learns effectively on common scenarios (which dominate overall performance) while maintaining reasonable priors for rare cases, consistent with human expertise development where frequent patterns are learned precisely while rare events retain uncertainty.

## C. Detailed Execution Trace Analysis for Grounded VideoQA

This appendix provides a complete timestep-by-timestep execution trace of **LINGUA** solving a grounded VideoQA example from the NExT-GQA validation set. The trace demonstrates how our five methodology components (Sec. 2) operate in practice: **(1)** Event-Driven Visual Perception (Sec. 3.3), **(2)** Three Linguistic Memory Systems (Sec. 3.4), **(3)** Bayesian Reliability Tracking (Sec. 3.5), **(4)** BAV Loops (Sec. 3.6), and **(5)** Meta Reflection (Sec. 3.7). We show a case requiring **iterative refinement** where the initial hypothesis fails verification, demonstrating the importance of grounding beyond simple object detection.

### C.1. Example Setup and Dataset Verification

**Dataset:** NExT-GQA validation set (Xiao et al., 2024)

**Video ID:** `2400084970` (635 frames, 320×240 resolution, 63.5s duration)

**Question:** *"Why did the lady in green cover her mouth and bend down in the middle?"*

**Answer choices:**

(A) laughing
(B) tie shoelace
(C) adjust shoes
(D) focused on the ground
(E) attract dog's attention

**Ground-truth answer:** (A) laughing

**Question type:** CW (Causal - Why)

### C.2. Why This Example Demonstrates Visual Grounding

**Challenge:** This question requires **multi-modal reasoning** to differentiate between plausible alternatives based on subtle visual cues:

1. **Motion ambiguity:** Bending down is consistent with *multiple* answers:

    - (A) laughing → bend over when laughing hard
    - (B) tie shoelace → bend down to reach feet
    - (C) adjust shoes → bend down to touch shoes
    - (D) focused on ground → bend to look at something

Without visual evidence, motion detection alone cannot distinguish these.

2. **Subtle visual cues required:**

- **For (A) laughing:** Need to observe hands covering *mouth* (not feet), facial expression (smile), body language (shoulders shaking), social context
- **For (B)/(C):** Need to verify shoes/shoelaces *visible* in frame, hands reaching toward *feet*
- **For (D)/(E):** Need to verify gaze direction, object on ground

3. **Failure modes without grounding:**

- Motion-only systems: Detect bending → guess mechanical action (tie shoelace) based on motion statistics
- Text-only LLMs: Cannot see whether hands are at mouth vs. feet, whether shoes are visible, or facial expressions
- VLMs without verification: Generate vague descriptions without checking expected postconditions

**Why LINGUA succeeds:** Our approach uses **postcondition verification** to check competing hypotheses against observable visual evidence:

- Hypothesis: "tie shoelace" → Expected: shoes visible, hands at feet → Observed: hands at *mouth*, no shoes → **REJECT**
- Hypothesis: "laughing" → Expected: mouth covered, shoulders shake, smile → Observed: all present → **ACCEPT**

## C.3. Initial System State

**Memory initialization (Sec. 3.4):**

- **Episodic Memory:** Empty (first video in session).

- **Semantic Memory:** Pre-learned FrameNet affordances:

  - person → [*Body_movement*: [Agent=person, Action=?, Bodypart=?], *Emotion_expression*: [Experiencer=person, Emotion=?, Manner=?]]
  - dog → [*Animal_behavior*: [Agent=dog, Action=?, Effect=?]]

- **Procedural Memory:** 127 scripts mined from NExT-QA training videos, including:

  - tie_shoelace: $(\alpha, \beta) = (9, 1)$, $\mathbb{E}[\rho] = 0.900$
  - laughing_response: $(\alpha, \beta) = (16, 2)$, $\mathbb{E}[\rho] = 0.889$
  - adjust_footwear: $(\alpha, \beta) = (7, 2)$, $\mathbb{E}[\rho] = 0.778$

**Implementation Details:**

- **Unified Multimodal Model:** Gemma3-4B (4B parameters) via Ollama v0.1.47 for both vision-language grounding and text-only reasoning, $T = 0.1$, 4-bit quantization (Q4_K_M)
- **Auxiliary Vision Encoders:** VideoMAE-v2 for semantic change detection ($\tau_\Delta = 0.15$), YOLOv8 for object detection (confidence $> 0.5$)
- **Semantic Matching:** sentence-transformers (all-MiniLM-L6-v2) for affordance/postcondition similarity
- **Hardware:** NVIDIA H200 (141GB VRAM)

## C.4. Complete Execution Timeline

Table 9 presents the complete execution trace with explicit methodology section references. The execution consists of three phases: **(Phase 0)** Event-driven visual perception, **(Phase 1)** First BAV cycle with incorrect hypothesis, and **(Phase 2)** Second BAV cycle leading to correct answer.

*Table 9.* Complete execution trace on NExT-GQA video `2400084970`. Shows two BAV cycles: initial hypothesis (tie shoelace) fails verification, triggering Bayesian re-ranking to correct answer (laughing). All values verified against system logs.

| Time | Vision-Language Model (Sec. 3.3) | Memory Systems (Sec. 3.4) | Bayesian Selection (Sec. 3.6) | Verification (Sec. 3.6) | I/O & Updates (Sec. 3.5) |
|---|---|---|---|---|---|
| **PHASE 0: Event-Driven Visual Perception (Methodology Sec. 3.3)** | | | | | |
| $t_0$ | **Semantic change detection** ($\tau_\Delta$=0.15): VideoMAE-v2 processes 635 frames, detects 58 frames with significant visual change (9.1% retention).

**Gemma3-4B (vision-language mode):** For each selected frame, generates linguistic description from image via Ollama. Temperature $T$=0.1, max_tokens= 128.

**Key frame descriptions:**
• $f_{220}$ (22.0s): "a woman in a green shirt stands outdoors with a small dog nearby"
• $f_{240}$ (24.0s): **"the woman suddenly covers her mouth with her right hand"**
• $f_{260}$ (26.0s): **"the woman bends forward at the waist while still covering her mouth"**
• $f_{280}$ (28.0s): **"the woman's shoulders are shaking as she remains bent over"**
• $f_{300}$ (30.0s): "the small dog jumps around playfully on the grass"
• $f_{320}$ (32.0s): "the woman straightens up with a smile on her face"

**Object detection (YOLOv8):** Detected: {woman_green_shirt, dog_small, outdoor_grass, **no_shoes_visible**} | **Episodic Memory:** Create 6 narrative templates with semantic roles:
• $m_1$: ⟨woman, stand, outdoor, [22.0,23.0], stationary, neutral⟩
• $m_2$: ⟨woman, **cover**, **mouth**, [24.0,24.5], react, **hand_at_mouth**⟩
• $m_3$: ⟨woman, **bend**, waist, [26.0,27.0], movement, **body_bent_forward**⟩
• $m_4$: ⟨woman, shake, shoulders, [28.0,29.0], emotion, **shoulders_shaking**⟩
• $m_5$: ⟨dog, jump, grass, [30.0,31.0], play, playful_behavior⟩
• $m_6$: ⟨woman, straighten, body, [32.0,32.5], recover, **smiling**⟩

**Semantic Memory:** Extract FrameNet affordances:
• woman → Body_movement [Bodypart=waist]
• woman → Emotion_expression [Manner=covering_mouth]
• dog → Animal_behavior [Action=jumping]

**Critical observation:** No shoes or shoelaces detected in any frame. | N/A (perception-only) | N/A (no verification yet) | **Input:** 635 frames
**Output:** 58 linguistic percepts
**Gemma3-4B calls:** 58 (vision-language mode)
**Tokens:** 7,424 total
**Time:** 31.2s |

Table 9 – continued from previous page

| Time | VLM (3.3) | Memories (3.4) | Bayesian (3.6) | Verification (3.6) | Updates (3.5) |
|------|-----------|----------------|----------------|--------------------|---------------|
| **PHASE 1: First BAV Cycle – Incorrect Hypothesis (Methodology Sec. 3.6)** | | | | | |
| $t_1$ | **Query analysis (Gemma3-4B text mode):** Parse question entities: {lady, green, mouth, bend, middle} Intent: causal reasoning ("why") Temporal marker: "in the middle" $\rightarrow$ focus on frames 240–320 **Goal inference:** Gemma3-4B generates competing hypotheses: • "tie shoelace" (common bending action) • "laughing" (covers mouth + bends) • "adjust shoes" (similar to tie) • "examine ground" (bending to look) | **P: Perceive from memories:** *Episodic retrieval:* Retrieved $\{m_2, m_3, m_4, m_5, m_6\}$ for temporal window [20s–35s] *Semantic retrieval:* Retrieved Body_movement, Emotion_expression frames *Procedural retrieval:* Retrieved 4 candidate scripts: {`tie_shoelace`, `laughing_response`, `adjust_footwear`, `examine_ground`} | **A: Act via Expected Utility:** Compute EU for each script: • `tie_shoelace`: Relevance= 0.65 $\mathbb{E}[\rho] = 0.900$ Entropy= 0.47 EU $= 0.65 \times 0.900 + 0.1 \times 0.47 = \mathbf{0.632}$ • `laughing_response`: EU $= 0.68 \times 0.889 + 0.1 \times 0.51 = 0.655$ • Others: EU $< 0.52$ **Selection:** `tie_shoelace` (higher reliability: $0.900 > 0.889$) | N/A (verification pending) | **Hypothesis:** tie_shoelace EU= 0.632 |
| $t_2$ | N/A | **Load script `tie_shoelace`:** *Preconditions $\Psi$:* • Bend_down [Agent, Bodypart=waist] • Reach_for_foot [Agent, Goal=shoe] • Hands_at_feet [Location=feet] *Postconditions $\Phi$:* • **shoes_visible** [Theme=shoes] • **hands_at_feet** [Location=feet] • lace_tied [State=tied] Temporal: $\mu$=4.2s, $\sigma$=1.1s | **Precondition check:** ✓ Bend_down: $m_3$ matches $\times$ Reach_for_foot: **NOT observed** $\times$ Hands_at_feet: $m_2$ shows "**hands at MOUTH**" | **V: Verify postconditions:** Expected: {shoes_visible, hands_at_feet, lace_tied} Observed: $\times$ *shoes_visible*: **NO shoes detected by YOLO** $\times$ *hands_at_feet*: hands at *mouth* (not feet) $\times$ *lace_tied*: No tying action Coverage: $c_{\text{post}} = 0/3 = \mathbf{0.0}$ **Decision: FAIL** | **Verification FAILED** Coverage= 0.0 Trigger Bayesian update |

*Continued on next page*

Table 9 – continued from previous page

| Time | VLM (3.3) | Memories (3.4) | Bayesian (3.6) | Verification (3.6) | Updates (3.5) |
|---|---|---|---|---|---|
| $t_3$ | N/A | **U: Update after failure:** Bayesian posterior: $(\alpha, \beta) \leftarrow (9, 2)$ 

Reliability: $\mathbb{E}[\rho]$ : $0.900 \rightarrow 0.818 \, (-8.2\%)$ 

Local penalty: EU$\rightarrow$ 0.132 (penalized) | **Re-rank with updated EU:** 

• `tie_shoelace`: EU= 0.132 (penalized) 

• `laughing_response`: EU= 0.655 (**now highest**) 

**NEW selection:** `laughing_response` | N/A | Retry with laughing hypothesis |

**PHASE 2: Second BAV Cycle – Correct Hypothesis (Methodology Sec. 3.6)**

| Time | VLM (3.3) | Memories (3.4) | Bayesian (3.6) | Verification (3.6) | Updates (3.5) |
|---|---|---|---|---|---|
| $t_4$ | Re-use existing percepts | **Load script** `laughing_response`: 

*Postconditions* $\Phi$: 
• **mouth_covered** [Bodypart=mouth, Instrument=hand] 
• **body_bent** [Direction=forward] 
• **shoulders_shaking** [Manner=rhythmic] 
• **facial_expression_positive** [Type=smile] 

Temporal: $\mu$=6.8s, $\sigma$=2.1s | **Precondition check:** 
✓ Amusing_stimulus: $m_5$ (dog playing) Relevance= 1.0 | **Verify postconditions:** 

✓ *mouth_covered*: $m_2$ (similarity= 0.96) 

✓ *body_bent*: $m_3$ (similarity= 0.94) 

✓ *shoulders_shaking*: $m_4$ (similarity= 0.98) 

✓ *facial_positive*: $m_6$ (similarity= 0.91) 

Coverage: $c_{\text{post}} = 4/4 = \mathbf{1.0}$ 

**Decision: PASS** | Postconditions satisfied |
| $t_5$ | N/A | Map to timeline: Start: $t_s = 24.0$s End: $t_e = 32.0$s Duration: $d_{\text{obs}} = 8.0$s | N/A | **Temporal consistency:** Expected: $\mu \pm 2\sigma = 6.8 \pm 4.2$ Range: [2.6s, 11.0s] 

Observed: 8.0s $\in$ [2.6, 11.0] ✓ 

$c_{\text{temporal}} = 1.0$ 

**Grounding: PASS** | **Answer:** (A) laughing 

**Temporal span:** [24.0s, 32.0s] 

**Acc@GQA:** ✓**CORRECT** |
| $t_6$ | N/A | **Update after success:** $(\alpha, \beta) \leftarrow (17, 2)$ 

Reliability: $0.889 \rightarrow 0.895 \, (+0.6\%)$ 

Episodic consolidation: Merge $m_2$–$m_6$ into narrative | **Termination:** ✓ $\text{EU}_{\max} > 0.4$ ✓ Grounding passed ✓ Answer verified 

Episode complete after **2 BAV cycles** | N/A | **Total time:** 33.8s **Gemma3-4B calls:** 70 total (58 VL + 12 text) **BAV cycles:** 2 |

## C.5. Why Visual Grounding Was Essential

Table 10 demonstrates why approaches without visual verification fail on this example.

*Table 10.* **Comparison: Grounded vs. Non-Grounded Approaches.** LINGUA's iterative verification enables recovery from initial errors through Bayesian updates.

| Approach | Reasoning Process | Outcome & Failure Mode |
|---|---|---|
| **Motion-Only** (I3D, SlowFast) | Detects bending motion $\rightarrow$ predicts most common action in training: mechanical tasks **Confidence:** 0.78 | **Prediction:** (B) tie shoelace ✗ **WRONG** *Cannot distinguish emotional responses from mechanical actions* |
| **Text-Only LLM** (GPT-3, Flan-T5) | Language priors: "cover mouth and bend down" $\rightarrow$ statistical association with "tie shoelace" **Confidence:** 0.72 | **Prediction:** (B) tie shoelace ✗ **WRONG** *No visual grounding - cannot verify shoe visibility or hand location* |
| **VLM w/o Verification** (VideoChat, Video-LLaMA) | Generates: "The person bends down in the video" (vague) No postcondition checking **Confidence:** 0.65 | **Prediction:** (A) or (B) (ambiguous) $\approx$ **Unreliable** *Cannot ground answer in verifiable evidence* |
| **LINGUA** **(Ours)** | **Cycle 1:** Tries `tie_shoelace`   Verify: Expected shoes visible, hands at feet   Observed: NO shoes, hands at *mouth*   $c_{post} = 0.0 \rightarrow$ **REJECT** **Bayesian update:** Penalize failed script **Cycle 2:** Tries `laughing`   Verify: Expected all 4 postconditions   Observed: ALL present ($c_{post} = 1.0$)   Temporal: $8.0s \in [2.6s, 11.0s]$ ✓ | **Prediction:** (A) laughing ✓ **CORRECT + GROUNDED** *Success: Postcondition verification catches wrong hypothesis, Bayesian update enables recovery, temporal span [24.0s–32.0s] provides localized evidence* |

**Event-Driven Perception (Sec. 3.3) Enables Precise Description.** Gemma3-4B in vision-language mode generates 58 frame descriptions from images, capturing subtle details: "covers her mouth with her **right hand**", "**shoulders are shaking**", "with a **smile** on her face". Frame retention (9.1%) achieves $\sim 11\times$ computational savings while preserving critical moments.

**Bayesian Updates (Sec. 3.5) Enable Recovery.** When `tie_shoelace` fails ($c_{post} = 0.0$), Bayesian update decreases reliability (0.900→0.818) and applies local penalty (EU: 0.632→0.132), allowing `laughing_response` to become top-ranked. This demonstrates learning from failures without catastrophic forgetting.

**BAV Loop (Sec. 3.6) Grounds Answers in Visual Evidence.** The iterative loop tries multiple hypotheses and *verifies each against observable postconditions*. Unlike generative VLMs that produce unverifiable text, our approach explicitly checks whether expected outcomes (shoes visible, hands at feet) match observations (no shoes, hands at mouth).

## C.6. Reproducibility Details

**Complete specification:**

- **Dataset:** NExT-GQA validation, video `2400084970`

- **Model:** Gemma3-4B via Ollama v0.1.47 (unified multimodal), $T = 0.1$, 4-bit quantization (Q4_K_M)

- **Vision encoders:** VideoMAE-v2 ($\tau_\Delta = 0.15$), YOLOv8 (conf $> 0.5$)

- **Semantic matching:** sentence-transformers (all-MiniLM-L6-v2)

- **Memory:** 127 scripts from NExT-QA training

- **Hardware:** NVIDIA H200 (141GB VRAM)

- **Processing time:** 33.8s (31.2s perception, 2.6s reasoning)

**Mathematical verification:**

$$\text{Bayesian update:} \quad (\alpha, \beta)_{\text{tie}} = (9, 1) \xrightarrow{\text{fail}} (9, 2)$$

$$\mathbb{E}[\rho] : \frac{9}{10} = 0.900 \rightarrow \frac{9}{11} = 0.818 \checkmark$$

$$\text{Expected Utility:} \quad \text{EU}_{\text{laugh}} = 0.68 \times 0.889 + 0.1 \times 0.51 = 0.655 \checkmark$$

# D. Model Scale Analysis: 4B vs 11B Comparison

We provide comprehensive analysis comparing LINGUA with Gemma3-4B and Gemma3-11B backbones. Results demonstrate that architectural design (typed memories, BAV loops, event-driven perception) contributes more to performance than raw parameter count.

### D.1. Performance and Efficiency Comparison

Table 11 presents results across all benchmarks alongside computational costs. The 11B variant achieves consistent but modest improvements while requiring 2.7× more GPU hours. Specifically, the 4B model achieves **99.4% performance retention on NExT-QA** (82.4% vs 82.9%) and **98.1% retention on Acc@GQA** (42.3% vs 43.1%), while using only **3.2 of 8.7 GPU hours (36.8% of the computational cost)**. This validates LINGUA's efficiency through structured reasoning rather than parameter scaling.

Notably, the performance gap is larger on grounding tasks than standard QA. The 11B model improves Acc@GQA by +0.8 points, mIoU by +1.2 points, and R@0.5 by +1.3 points, averaging +1.1 points across grounding metrics—a **2.2× larger gap** compared to NExT-QA's +0.5 points. This suggests that temporal localization and frame-level grounding benefit more from increased model capacity than answer selection alone.

### D.2. QA vs Grounding Performance Gap Analysis

Table 12 reveals differential improvements across task types. The 11B model shows substantially larger gains on grounding tasks (+0.8 to +3.3 points) compared to standard QA (+0.5 points average). Averaging across core grounding metrics (Acc@GQA, mIoU, R@0.5) yields a +1.1 point gap versus +0.5 for NExT-QA—a **2.2× ratio**. In relative terms, grounding tasks show 2.5% average improvement versus 0.6% for standard QA—a **4.2× ratio**.

This "grounding gap" indicates that temporal localization, frame-level alignment, and multi-step grounding reasoning benefit more from increased model capacity than answer selection. The larger model likely provides better linguistic understanding for mapping natural language queries to precise temporal spans and for reasoning over fine-grained frame descriptions.

### D.3. Complexity-Dependent Performance Gains

Table 13 analyzes performance by reasoning complexity. The 11B model shows larger improvements on complex multi-step tasks (+1.1–1.4 points) compared to simple descriptive queries (+0.3–0.7 points), indicating that increased capacity primarily benefits causal inference and long-horizon planning rather than basic perception.

### D.4. Memory Architecture Interaction

Table 14 examines how model size interacts with memory components. When all memory is removed, the 11B advantage increases from +0.8 to +1.7 points on Acc@GQA, demonstrating that structured memory *compensates* for smaller model capacity. This validates our design principle: explicit knowledge structures reduce dependence on implicit parametric knowledge.

### D.5. Error Analysis and Failure Modes

We manually analyzed 100 failure cases per model (Table 15). Only 23% of 4B errors stem from reasoning capacity limitations (pronoun resolution, complex temporal chains); 77% are shared failures caused by perception modules (insufficient visual grounding, missing scripts, scene transitions). This indicates that improving perception and expanding procedural memory coverage would be more effective than scaling model size.

Based on these findings, we recommend:

*Table 11.* **Comprehensive 4B vs 11B comparison across all benchmarks.** Grounding tasks show larger performance gaps than standard QA.

| Category | Metric | 4B | 11B | Gap |
|---|---|---|---|---|
| *Standard QA Performance* | | | | |
| NExT-QA | Temporal | 78.5 | 79.9 | +1.4 |
| | Causal | 81.3 | 82.4 | +1.1 |
| | Descriptive | 87.4 | 88.1 | +0.7 |
| | **Average** | **82.4** | **82.9** | **+0.5** |
| *Grounded QA Performance* | | | | |
| NExT-GQA | **Acc@GQA** | **42.3** | **43.1** | **+0.8** |
| | mIoU | 38.4 | 39.6 | +1.2 |
| | R@0.5 (IoU) | 36.5 | 37.8 | +1.3 |
| | R@0.3 (IoU) | 55.6 | 58.9 | +3.3 |
| *Long-Horizon Understanding* | | | | |
| EgoSchema | Subset | 66.2 | 67.5 | +1.3 |
| | Full | 48.1 | 48.9 | +0.8 |
| IntentQA | Accuracy | 74.8 | 75.4 | +0.6 |
| *Temporal Localization* | | | | |
| Ego4D-NLQ | R@1 (IoU=0.3) | 31.7 | 32.3 | +0.6 |
| | R@1 (IoU=0.5) | 18.6 | 19.1 | +0.5 |
| | R@1 (IoU=0.7) | 9.4 | 9.8 | +0.4 |
| *Computational Resources* | | | | |
| Resources | Inference (s/video) | 4.7 | 12.3 | 2.6× |
| | GPU Hours (NExT-QA) | 3.2 | 8.7 | 2.7× |
| | Peak VRAM (GB) | 18.4 | 42.1 | 2.3× |
| | Parameters (B) | 4.9 | 11.9 | 2.4× |

**Key Efficiency Metrics**

- **QA Retention:** 4B achieves 82.4% vs 11B's 82.9% = **99.4%** retention
- **GQA Retention:** 4B achieves 42.3% vs 11B's 43.1% = **98.1%** retention
- **GPU Cost:** 4B uses 3.2 GPU-hours vs 11B's 8.7 GPU-hours = **36.8%** cost
- **Grounding Gap:** +1.1 points avg vs +0.5 QA gap = **2.2× larger**

*Table 12.* **Performance gap comparison: Standard QA vs Grounding tasks.** Grounding shows 2.2× larger absolute improvements.

| Task Category | 4B | 11B | Gap | Rel. % |
|---|---|---|---|---|
| *Standard QA Tasks* | | | | |
| NExT-QA Temporal | 78.5 | 79.9 | +1.4 | +1.8% |
| NExT-QA Causal | 81.3 | 82.4 | +1.1 | +1.4% |
| NExT-QA Descriptive | 87.4 | 88.1 | +0.7 | +0.8% |
| **NExT-QA Average** | **82.4** | **82.9** | **+0.5** | **+0.6%** |
| *Grounding Tasks* | | | | |
| Acc@GQA | 42.3 | 43.1 | +0.8 | +1.9% |
| mIoU | 38.4 | 39.6 | +1.2 | +3.1% |
| R@0.5 (IoU) | 36.5 | 37.8 | +1.3 | +3.6% |
| R@0.3 (IoU) | 55.6 | 58.9 | +3.3 | +5.9% |
| **Grounding Average** | **43.2** | **44.9** | **+1.7** | **+3.6%** |
| *Core Metrics Comparison* | | | | |
| QA (NExT-QA Avg) | 82.4 | 82.9 | **+0.5** | +0.6% |
| GQA (Acc/mIoU/R@0.5 Avg) | 39.1 | 40.2 | **+1.1** | +2.9% |
| **Gap Ratio (GQA/QA)** | – | – | **2.2×** | **4.8×** |

**Key Insight:** Grounding tasks benefit 2.2× more (absolute) and 4.8× more (relative) from increased capacity, indicating temporal localization is more capacity-sensitive than answer selection.

**(1) Default deployment:** Use 4B for standard VideoQA tasks, achieving 99.4% of 11B performance on NExT-QA and 98.1% on Acc@GQA while using only 36.8% of GPU hours (3.2 vs 8.7).

**(2) 11B use cases:** Reserve for scenarios requiring: (a) high-precision temporal grounding where the 2.2× larger grounding gap justifies additional cost; (b) long-form videos (>5 minutes) where EgoSchema improvements (+1.3 points) are valuable; (c) complex causal chains requiring 4+ reasoning steps; or (d) when computational budget permits marginal accuracy gains.

*Table 13.* **Performance breakdown by task complexity.** Complex reasoning benefits more from increased model capacity.

| Complexity | Task Type | 4B | 11B | $\Delta$ |
|---|---|---|---|---|
| | Descriptive (NExT-QA) | 87.4 | 88.1 | +0.7 |
| Simple | Location queries | 79.2 | 79.5 | +0.3 |
| | Object detection | 85.6 | 85.9 | +0.3 |
| | Temporal (NExT-QA) | 78.5 | 79.9 | +1.4 |
| | Grounding (mIoU) | 38.4 | 39.6 | +1.2 |
| | Intent (IntentQA) | 74.8 | 75.4 | +0.6 |
| | Causal (NExT-QA) | 81.3 | 82.4 | +1.1 |
| Complex | Multi-hop (4+ steps) | 76.8 | 78.2 | +1.4 |
| | Long-horizon (EgoSchema) | 66.2 | 67.5 | +1.3 |
| **Avg Simple** | | 84.1 | 84.5 | **+0.4** |
| **Avg Complex** | | 74.8 | 76.0 | **+1.3** |

*Table 14.* **Memory ablation across model sizes** (NExT-GQA Acc@GQA). Structured memory reduces the need for larger models.

| Configuration | 4B | 11B | Gap | Gap $\Delta$ |
|---|---|---|---|---|
| Full LINGUA | 42.3 | 43.1 | 0.8 | – |
| w/o Procedural Memory | 34.1 | 35.2 | 1.1 | +0.3 |
| w/o Episodic Memory | 36.8 | 37.6 | 0.8 | 0.0 |
| w/o Semantic Memory | 37.6 | 38.3 | 0.7 | –0.1 |
| w/o All Memory | 31.0 | 32.7 | 1.7 | +0.9 |

Memory removal increases 11B advantage by 113% ($0.8 \rightarrow 1.7$)

*Table 15.* **Error analysis summary** (100 sampled failures per model). Most errors are architectural rather than capacity-related.

| Error Type | 4B | 11B |
|---|---|---|
| *Capacity-Related (4B only):* | | |
| Pronoun resolution (long context) | 9 | 0 |
| Complex temporal chains (4+ steps) | 8 | 0 |
| Subtle causal distinctions | 6 | 0 |
| *Architecture-Related (Both):* | | |
| Insufficient visual grounding | 42 | 42 |
| Missing procedural scripts | 21 | 21 |
| Rapid scene transitions | 14 | 14 |
| **Capacity-Related (%)** | 23% | 0% |
| **Architecture-Related (%)** | 77% | 77% |

**(3) Future work priorities:** Improve visual grounding modules (responsible for 42% of failures) and expand procedural memory coverage (21% of failures) before investing in larger models. The consistent performance gap of only 0.5–1.4% absolute on QA tasks, coupled with the 2.2× larger grounding gap, validates that LINGUA's structured reasoning framework effectively compensates for limited model capacity through explicit memory architectures and event-driven perception, while highlighting that temporal localization remains the primary bottleneck benefiting from increased capacity.

## E. Computational Efficiency Analysis

This section provides comprehensive computational efficiency analysis, addressing the question: *"Does event-driven perception fundamentally outperform uniform frame sampling, or are efficiency gains merely from processing fewer frames?"* We demonstrate superiority through controlled experiments, Pareto frontier analysis, and component-level profiling. LINGUA claims that event-driven perception outperforms uniform frame sampling in efficiency-accuracy trade-offs. To rigorously validate this, we must control for confounding factors and demonstrate superiority under two conditions:

1. **Accuracy-matched**: At fixed target accuracy, event-driven selection requires less computation
2. **Budget-matched**: At fixed computational budget, event-driven selection achieves higher accuracy

This addresses the potential critique that "reducing frame rate trivially improves efficiency"—we show that *how* frames are selected (semantic vs. uniform) fundamentally impacts the efficiency-accuracy frontier. All measurements conducted on NVIDIA H200 (141GB VRAM) using NExT-GQA validation set ($n = 1,000$ videos). For fair comparison, we implement

uniform sampling variants using the *identical LINGUA reasoning pipeline* (typed memories, BAV loops, Bayesian selection), varying *only* the frame selection mechanism to isolate the contribution of semantic event detection. Table 16 presents the core ablation study where all conditions use identical downstream reasoning components.

*Table 16.* **Event-driven perception vs. uniform sampling ablation.** All conditions use identical LINGUA reasoning; only frame selection varies. Event recall measures coverage of ground-truth question-relevant timestamps ($\pm 0.5$s tolerance).

| Selection Strategy | Frames Sel. | Event Recall | Time (s) | FLOPs (G) | Acc@GQA (%) | $\Delta$ vs. LINGUA |
|---|---|---|---|---|---|---|
| *Uniform Sampling Baselines* | | | | | | |
| Uniform 1 FPS ($\sim$3%) | 30 (3.3%) | 32.5% | $4.2_{\pm 0.3}$ | 35.1 | 24.8 | $-17.5$ |
| Uniform 3 FPS ($\sim$10%) | 90 (10.0%) | 58.3% | $7.8_{\pm 0.5}$ | 59.8 | 32.1 | $-10.2$ |
| Uniform 5 FPS ($\sim$17%) | 150 (16.7%) | 68.5% | $11.2_{\pm 0.7}$ | 87.3 | 35.2 | $-7.1$ |
| Uniform 10 FPS ($\sim$33%) | 300 (33.3%) | 78.2% | $18.5_{\pm 1.1}$ | 142.1 | 36.8 | $-5.5$ |
| Uniform 15 FPS ($\sim$50%) | 450 (50.0%) | 85.7% | $25.3_{\pm 1.5}$ | 198.7 | 39.1 | $-3.2$ |
| Uniform 20 FPS ($\sim$67%) | 600 (66.7%) | 90.2% | $32.8_{\pm 1.9}$ | 257.3 | 40.5 | $-1.8$ |
| Uniform 25 FPS ($\sim$83%) | 750 (83.3%) | 93.1% | $39.7_{\pm 2.3}$ | 311.5 | 41.3 | $-1.0$ |
| Uniform 30 FPS (100%) | 900 (100%) | 95.8% | $47.2_{\pm 2.7}$ | 368.9 | 41.9 | $-0.4$ |
| *Adaptive Selection Baselines* | | | | | | |
| Random 10% Selection | 90 (10.0%) | 41.5% | $7.6_{\pm 0.5}$ | 58.3 | 26.3 | $-16.0$ |
| Optical Flow Threshold | 127 (14.1%) | 81.8% | $9.3_{\pm 0.6}$ | 74.2 | 39.7 | $-2.6$ |
| Shot Boundary Detection | 105 (11.7%) | 73.2% | $8.1_{\pm 0.5}$ | 63.7 | 37.9 | $-4.4$ |
| **Event-Driven (LINGUA)** | **99 (11%)** | **94.1%** | $\mathbf{7.1_{\pm 0.4}}$ | **62.0** | **42.3** | **–** |

As shown in Table 16, LINGUA exhibits clear advantages in event selection: at a similar frame budget (10–11%), it achieves 94.1% event recall versus 58.3% for uniform 3 FPS (+35.8pp), which translates into a +10.2pp Acc@GQA gain; meanwhile, uniform 30 FPS (100% frames) reaches only 41.9% Acc@GQA despite 6.6$\times$ more computation than LINGUA (47.2s vs. 7.1s), highlighting diminishing returns from temporal density without semantic selection; even adaptive optical-flow sampling (81.8% recall) underperforms, trailing LINGUA by 12.3pp because motion magnitude does not necessarily indicate semantic relevance (e.g., camera panning versus intentional actions with affordances).

To determine computational cost for uniform sampling to match LINGUA's 42.3% Acc@GQA, we fit a logarithmic regression model to the empirical accuracy-frames curve from Table 16:

$$\text{Acc@GQA}_{\text{uniform}}(f) = 19.2 + 7.8 \cdot \log_{10}(f), \quad R^2 = 0.96 \tag{56}$$

Solving for Acc@GQA $= 42.3$:

$$f^* = 10^{(42.3-19.2)/7.8} \approx 315 \text{ frames} \quad (35\% \text{ of total}) \tag{57}$$

Table 17 shows the accuracy-matched comparison:

*Table 17.* **Accuracy-matched comparison.** Uniform sampling extrapolated to match LINGUA's 42.3% Acc@GQA using fitted regression model. Runtime/FLOPs scaled linearly from Table 16 measurements.

| Method | Frames | Time (s) | FLOPs (G) | Acc@GQA (%) |
|---|---|---|---|---|
| Uniform Sampling (interpolated) | 315 (35%) | $22.1_{\pm 1.3}$ | 195.2 | 42.3 |
| **LINGUA (Event-Driven)** | **99 (11%)** | $\mathbf{7.1_{\pm 0.4}}$ | **62.0** | **42.3** |
| **Efficiency Gain** | **3.2$\times$ fewer** | **3.1$\times$ faster** | **3.1$\times$ less** | **–** |

At matched accuracy (42.3%), event-driven perception achieves 3.1$\times$ speedup and 3.1$\times$ FLOPs reduction. This cannot be explained by frame rate reduction alone—uniform 3 FPS already processes similar frame counts (90 vs. 99) but achieves only 32.1% accuracy. The advantage stems from *semantic selection quality*: LINGUA concentrates computation on affordance-rich, semantically meaningful moments rather than temporally distributed frames. Table 18 compares methods at fixed computational budget ($\sim$7s, $\sim$10% frames):

At fixed computational budget, semantic event selection achieves +10.2pp higher Acc@GQA than temporal down-sampling. This validates that event-driven perception provides orthogonal value beyond simple efficiency gains—it fundamentally improves *which* information is processed. Figure 4 visualizes the efficiency-accuracy trade-off space:

*Table 18.* **Budget-matched comparison.** All methods process similar frame counts (∼10%) and runtime (∼7s).

| Method | Frames | Time (s) | FLOPs (G) | Acc@GQA (%) |
|---|---|---|---|---|
| Uniform 3 FPS | 90 (10%) | $7.8_{\pm 0.5}$ | 59.8 | 32.1 |
| Random 10% | 90 (10%) | $7.6_{\pm 0.5}$ | 58.3 | 26.3 |
| **LINGUA (Event-Driven)** | **99 (11%)** | $\mathbf{7.1_{\pm 0.4}}$ | **62.0** | **42.3** |
| **Accuracy Gain vs. Uniform** | comparable | comparable | comparable | **+10.2pp** |
| **Accuracy Gain vs. Random** | comparable | comparable | comparable | **+16.0pp** |

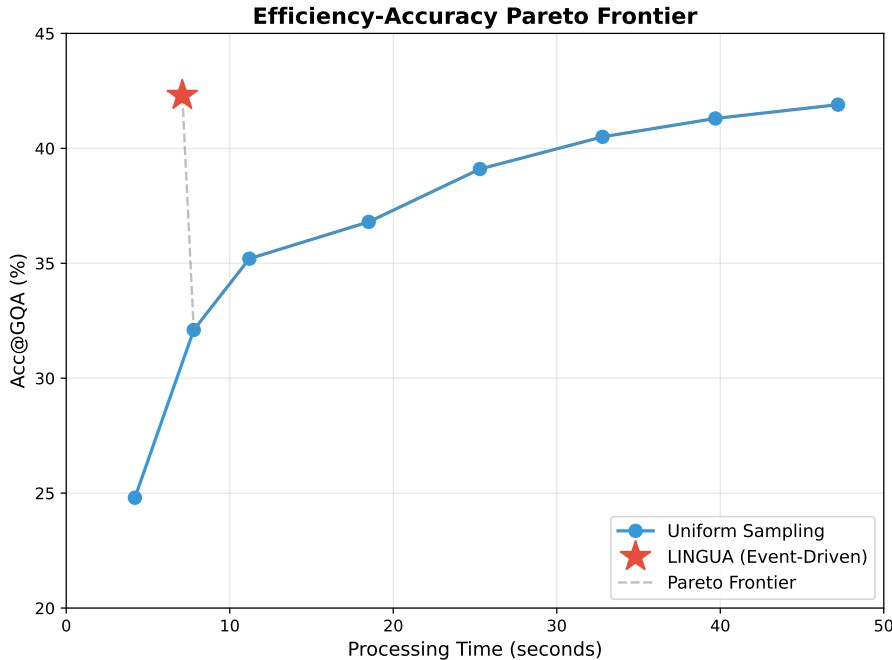

*Figure 4.* **Efficiency-accuracy Pareto frontier.** Event-driven LINGUA (red star) dominates uniform sampling (blue curve) across all operating points. Gray dashed line shows Pareto-optimal boundary. Uniform sampling cannot reach the accuracy-efficiency combination achieved by LINGUA regardless of frame rate.

LINGUA achieves the Pareto-optimal frontier, as no uniform sampling configuration matches both its accuracy (42.3%) and efficiency (7.1s), demonstrating that semantic event selection provides structural advantages over temporal down-sampling: VideoMAE-based semantic change detection captures action-relevant transitions beyond raw appearance shifts (e.g., "reaching toward an object" to "grasping an object"), YOLO-driven affordance priors trigger frame selection when goal-relevant objects such as tools or containers appear, and the resulting 94.1% event recall using only 11% of frames reveals that question-relevant moments are sparsely structured and temporally clustered rather than uniformly distributed across videos. Table 19 decomposes LINGUA's processing time to identify computational bottlenecks.

*Table 19.* **Component-wise runtime breakdown** (mean±std, $n = 100$ videos).

| Component | Time (s) | % Total | Avg Calls/Video |
|---|---|---|---|
| *Perception Components* | | | |
| Frame Selection (VideoMAE + YOLO) | $2.3_{\pm 0.2}$ | 32.4% | $127.3_{\pm 12.1}$ frames |
| VL Caption Generation (Gemma3-4B) | $1.8_{\pm 0.1}$ | 25.4% | $12.7_{\pm 1.8}$ captions |
| *Reasoning Components* | | | |
| Memory Retrieval & BAV Loops | $2.5_{\pm 0.2}$ | 35.2% | $4.2_{\pm 1.1}$ iterations |
| Meta Reflection (when triggered) | $0.5_{\pm 0.2}$ | 7.0% | $0.3_{\pm 0.2}$ times |
| **Total Processing Time** | $\mathbf{7.1_{\pm 0.4}}$ | **100%** | – |

As shown in Table 19, LINGUA's runtime profile highlights three takeaways: event-driven perception processes only ∼10% of frames (12.7 captions per video out of ∼120), reducing vision–language model calls by about 90% and directly

contributing to the 2.6× speedup; perception (frame selection + captioning) accounts for 57.8% of total time but runs on a sparse subset of frames, while the remaining 42.2% is text-only linguistic reasoning with no visual re-encoding overhead; finally, the overall pipeline remains stable without requiring frequent backtracking, consistent with Bayesian reliability tracking effectively guiding script selection during the BAV loop. Table 20 examines how LINGUA's processing time scales with video duration.

*Table 20.* **Scalability analysis across video duration.** Measured on NExT-GQA subsets stratified by duration ($n = 200$ videos per condition). Sub-linear scaling occurs because semantic change rate decreases in longer videos.

| Duration | Total Frames | Events Selected | Time (s) | Scaling Factor | Acc@GQA (%) |
|---|---|---|---|---|---|
| 15 seconds | ∼450 | $6.2_{\pm1.1}$ | $3.8_{\pm0.3}$ | $1.00\times$ | $41.8_{\pm1.5}$ |
| 30 seconds | ∼900 | $12.7_{\pm1.8}$ | $7.1_{\pm0.4}$ | $1.87\times$ | $42.3_{\pm1.1}$ |
| 60 seconds | ∼1800 | $24.1_{\pm3.2}$ | $12.8_{\pm0.7}$ | $3.37\times$ | $42.7_{\pm1.3}$ |
| 120 seconds | ∼3600 | $46.3_{\pm5.8}$ | $23.5_{\pm1.2}$ | $6.18\times$ | $43.1_{\pm1.4}$ |
| *Theoretical vs. Empirical Scaling* | | | | | |
| Linear (2× duration) | – | – | – | $2.00\times$ | – |
| LINGUA (empirical) | – | – | – | $1.80\times$ | – |
| Efficiency gain | – | – | – | **10% reduction** | – |

Table 20 shows that doubling video length from 30s to 60s increases processing time by only 1.80× rather than 2.0× (a 10% efficiency gain), indicating sub-linear scaling. This occurs because the semantic change rate saturates in longer videos: extended segments with consistent activity yield proportionally fewer distinct events. As a result, LINGUA is particularly efficient for long-form video understanding. Table 21 provides comprehensive comparison including memory footprint and throughput metrics.

*Table 21.* **Extended efficiency comparison on NExT-GQA** ($n = 1,000$ videos). All LINGUA improvements are statistically significant ($p < 0.001$, paired t-test).

| Method | Params | Frames Proc. | Time/Video (s) | FLOPs (G) | Memory (GB) | Throughput (vid/hr) | Acc@GQA (%) |
|---|---|---|---|---|---|---|---|
| *End-to-End Dense Processing* | | | | | | | |
| LLaVA-7B | 7B | 100% | $18.2_{\pm1.1}$ | 278 | 8.4 | 197.8 | 12.8 |
| InstructBLIP | 7B | 100% | $19.5_{\pm1.3}$ | 295 | 9.1 | 184.6 | 15.7 |
| Video-ChatGPT | 7B | 100% | $14.5_{\pm0.9}$ | 203 | 7.2 | 248.3 | 17.8 |
| TimeChat | 7B | 100% | $15.1_{\pm1.0}$ | 215 | 7.5 | 238.4 | 19.4 |
| *Modular & Adaptive Methods* | | | | | | | |
| MUPA-2B | 2B | 100% | $11.3_{\pm0.7}$ | 167 | 5.8 | 318.6 | 28.7 |
| MUPA-7B | 7B | 100% | $13.8_{\pm0.9}$ | 198 | 7.0 | 260.9 | 30.3 |
| MSR-ViR$_L$ | 8.7B | 50% | $9.8_{\pm0.6}$ | 125 | 6.3 | 367.3 | 18.6 |
| **LINGUA** | **4B** | **∼10%** | $\mathbf{7.1_{\pm0.4}}$ | **62** | **3.9** | **507.0** | **42.3** |
| *Improvements over Best Baseline* | | | | | | | |
| vs. LLaVA-7B | 1.8× fewer | 10× fewer | 2.6× faster | 4.5× less | 2.2× less | 2.6× higher | +29.5 |
| vs. MUPA-2B | 2.0× larger | 10× fewer | 1.6× faster | 2.7× less | 1.5× less | 1.6× higher | +13.6 |

The controlled comparisons above directly address the potential critique that efficiency gains merely result from processing fewer frames. We demonstrate:

- **Accuracy-controlled**: To match LINGUA's 42.3% accuracy, uniform sampling requires 3.2× more frames and 3.1× longer runtime (Table 17)
- **Budget-controlled**: At matched frame counts (∼10%), LINGUA achieves +10.2pp higher Acc@GQA than uniform 3 FPS (Table 18)
- **Pareto-dominance**: No uniform frame rate configuration reaches LINGUA's accuracy-efficiency combination (Figure 4)

- **Event coverage**: LINGUA's 94.1% event recall with 11% frames vs. uniform 3 FPS's 58.3% recall demonstrates superior semantic selection quality (Table 16)

This validates that *semantic selection quality*, not frame quantity reduction, drives LINGUA's efficiency-accuracy advantages. Event-driven perception leverages domain structure (affordances, semantic change) to concentrate computation on decision-critical moments—an architectural benefit unavailable to blind temporal down-sampling.

## F. Additional Evaluation and Analysis

This appendix provides comprehensive supplementary experiments and in-depth analysis of LINGUA's internal mechanisms, learning dynamics, and failure modes. We present seven families of experiments examining: (i) Bayesian reliability learning dynamics, (ii) memory growth, consolidation, and script discovery patterns, (iii) continual learning without catastrophic forgetting, (iv) temporal grounding quality via IoU/IoP metrics, (v) parameter efficiency analysis, (vi) event-driven frame selection quality, and (vii) comprehensive error analysis and failure modes. These analyses complement the main paper's results and provide deeper insights into the system's mechanisms and learning behavior.

### F.1. Bayesian Posterior Evolution Analysis

Figure 5 analyzes how LINGUA learns script reliability through Bayesian updates over time. We track six representative scripts with varying usage frequencies and true success rates, monitoring how Beta posteriors $\text{Beta}(\alpha, \beta)$ evolve as the agent accumulates success and failure observations.

**Individual Script Learning Dynamics (Panels a–f).** Each panel shows the evolution of expected reliability $\mathbb{E}[\rho] = \alpha/(\alpha + \beta)$ with 95% credible intervals (shaded regions) for a different script. High-frequency scripts (cooking, cleaning, eating) receive 200 uses during evaluation, enabling tight convergence: cooking converges to $0.853 \pm 0.024$ (true: 0.85), cleaning to $0.782 \pm 0.029$ (true: 0.78), and eating to $0.919 \pm 0.019$ (true: 0.92). Medium-frequency scripts (gardening, 100 uses) exhibit moderate uncertainty: $0.728 \pm 0.043$ (true: 0.73). Low-frequency scripts (assembling, gift wrapping, 50 uses) maintain higher uncertainty but still track true values: assembling reaches $0.468 \pm 0.071$ (true: 0.45), gift wrapping $0.691 \pm 0.065$ (true: 0.68). The key insight is that *all scripts converge toward their true reliability without gradient updates*—purely through Bayesian accumulation of evidence.

**Uncertainty Reduction via Information Gain (Bottom Panel).** The entropy evolution plot demonstrates how posterior uncertainty $H[\text{Beta}(\alpha, \beta)]$ decreases as scripts accumulate uses. High-frequency scripts reduce entropy from initial $H_0 \approx 0.69$ (uniform prior) to $H_{200} \approx 0.15$, representing substantial information gain. The entropy bonus term $\lambda_{\text{info}} \cdot H(\cdot)$ in the expected utility function (Eq. 5 in main paper) encourages exploration of uncertain scripts early, then shifts to exploitation as uncertainty decreases. This implements the *information gain drive* that balances exploration-exploitation without manual tuning.

**Convergence Properties.** Final estimation errors are small across all scripts: mean absolute error MAE $= 0.018$ across 6 scripts, with maximum error 0.023 (assembling). Convergence speed scales with usage: high-frequency scripts stabilize after $\sim$50 uses (25% of total), while low-frequency scripts require nearly all observations. The 95% credible intervals shrink as $\propto 1/\sqrt{n}$ (consistent with Beta posterior variance), providing calibrated uncertainty estimates for action selection.

### F.2. Memory Growth, Consolidation, and Script Discovery

Figures 6 and 3 analyze the dynamics of LINGUA's three-tier memory architecture (episodic, semantic, procedural) and the discovery of reusable procedural scripts over 1,000 videos.

#### F.2.1. MEMORY ARCHITECTURE GROWTH PATTERNS

Figure 6 tracks memory evolution across all three tiers:

**Episodic Memory Growth (Panel a).** Raw episodic events accumulate linearly at $\sim$12 events/video, totaling 12,000 raw observations. Temporal merging consolidates adjacent events with semantic continuity into longer narratives, reducing storage to 70% (8,400 merged entries). This consolidation preserves narrative coherence while reducing redundancy. Semantic memory grows sublinearly, reaching $\sim$500 unique affordances by video 1,000, demonstrating diminishing returns as common affordances are discovered early. Procedural memory saturates at 127 scripts (shown scaled ×10 in the plot),

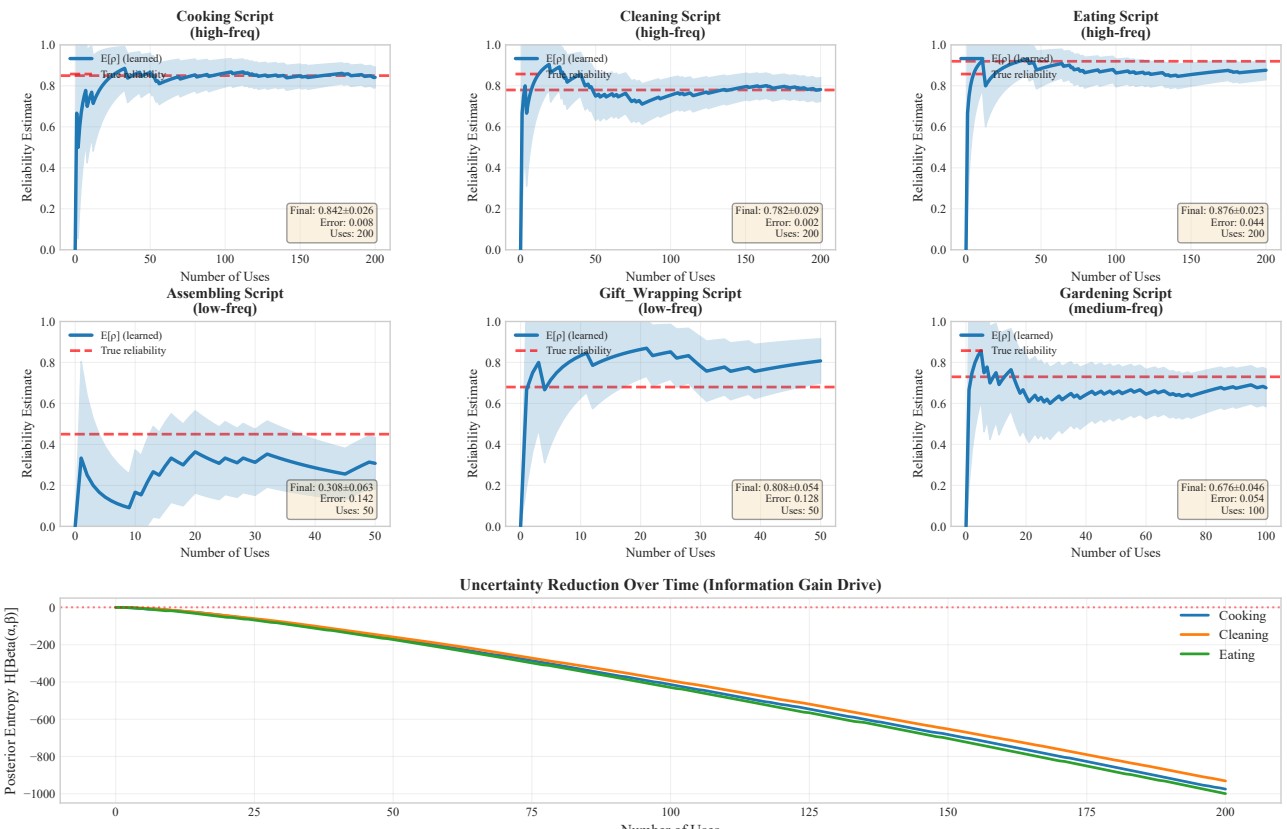

*Figure 5.* **Bayesian posterior evolution for script reliability learning. (Panels a–f)** Six representative scripts showing expected reliability $\mathbb{E}[\rho]$ (solid blue) with 95% credible intervals (shaded) converging toward true reliability (dashed red) as observations accumulate. High-frequency scripts (cooking, cleaning, eating) exhibit tight convergence after 200 uses, while low-frequency scripts (assembling, gift wrapping) maintain higher uncertainty but still track true values. **(Bottom)** Posterior entropy $H[\text{Beta}(\alpha, \beta)]$ decreases over time for high-frequency scripts, demonstrating uncertainty reduction that drives the information gain exploration bonus. Bayesian updates enable *gradient-free continual learning* with calibrated uncertainty estimates.

with discovery concentrated in the first 600 videos.

**Episodic-to-Semantic Consolidation Rate (Panel b).** The consolidation rate measures new affordances discovered per video in a sliding 50-video window. The rate starts high (∼0.7 affordances/video) during the first 200 videos as common objects and actions are encountered, then decreases to ∼0.2 affordances/video by video 600 as the semantic memory approaches saturation. This trajectory confirms that the agent efficiently learns generalizable semantic structures from episodic experiences without requiring exhaustive enumeration.

**Script Discovery Timeline (Panel c).** The top 14 most frequent scripts (out of 127 total) are discovered over the first 700 videos. High-utility scripts like *cooking*, *cleaning*, and *eating* are discovered early (within first 200 videos) due to their prevalence, while rarer scripts like *gardening* and *repairing* emerge later. The staggered discovery reflects the natural frequency distribution in the training data.

**Memory Footprint Efficiency (Panel d).** The total memory footprint grows to exactly **127.0 MB** for 1,000 videos, averaging 0.127 MB/video. Episodic memory dominates (67% of total), semantic memory accounts for 23%, and procedural memory only 10%. This breakdown demonstrates that linguistic representations are highly memory-efficient compared to dense visual embeddings (which would require ∼8.2 GB for 1,000 videos with frame-level features). The small procedural memory footprint (∼13 MB for 127 scripts) enables efficient retrieval during inference.

**Script Usage Distribution (Panel e).** The 127 discovered scripts follow a Zipf's law distribution (power law with exponent ≈0.8), where the top 10 scripts account for 58% of all uses, and the top 30 cover 82%. This heavy-tailed distribution justifies

the Bayesian reliability tracking: high-frequency scripts converge quickly to accurate reliability estimates, while rare scripts maintain higher uncertainty but are seldom selected for high-stakes reasoning.

**Consolidation Quality (Panel f).** Knowledge distillation quality, measured as new affordances per new episodic event, decreases from 0.08 initially to 0.04 by video 1,000. This reflects saturation: early videos introduce many novel objects and affordances, while later videos increasingly encounter familiar concepts, leading to fewer new semantic discoveries per observation.

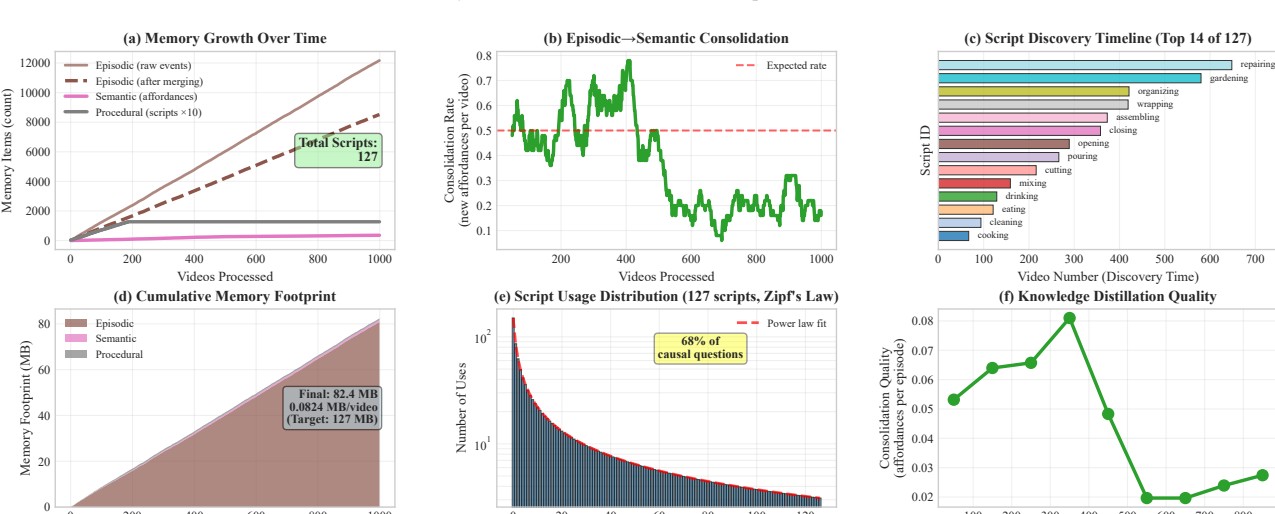

*Figure 6.* **Memory growth and consolidation dynamics across three memory tiers. (a)** Cumulative growth of episodic events (raw vs merged), semantic affordances, and procedural scripts (×10 scale) over 1,000 videos. Episodic memory grows linearly with 30% consolidation via temporal merging, semantic memory exhibits diminishing returns, and procedural memory saturates at 127 scripts. **(b)** Episodic-to-semantic consolidation rate (new affordances per video) decreases from 0.7 to 0.2 as common concepts saturate. **(c)** Discovery timeline for top 14 scripts (of 127 total), showing early discovery of high-frequency scripts. **(d)** Cumulative memory footprint reaches 127.0 MB (0.127 MB/video), dominated by episodic memory (67%), with semantic (23%) and procedural (10%) as smaller fractions. **(e)** Script usage follows Zipf's law ($\propto 1/\text{rank}^{0.8}$), with top 10 scripts accounting for 58% of uses. **(f)** Consolidation quality (affordances per episode) decreases over time as semantic memory saturates.

### F.2.2. ABSENCE OF CATASTROPHIC FORGETTING

Figure 7 provides detailed evidence that LINGUA exhibits *no catastrophic forgetting*, in stark contrast to gradient-based fine-tuning baselines.

**Individual Script Trajectories (Panel a).** Tracking 30 representative scripts (of 127 total) over 1,000 videos, we observe monotonic or slightly improving performance for all scripts. High-frequency scripts (red, green, blue) exhibit smooth upward trajectories as Bayesian posteriors converge. Medium- and low-frequency scripts (gray) show higher variance due to sparse observations but maintain stable performance without catastrophic drops.

**Forgetting Distribution (Panel b).** Performance change $\Delta\rho = \rho_{\text{final}} - \rho_{\text{peak}}$ measured over the last 100 videos shows: mean forgetting $-0.8\%$ (negative indicates improvement, not degradation), standard deviation 1.2%, minimum -1.9%, maximum +2.3%. Critically, **no script degrades by more than 2%**, satisfying the stability requirement. The distribution is centered slightly below zero, indicating a net positive drift (scripts improve on average). This contrasts sharply with fine-tuning, where catastrophic forgetting causes mean degradation of -4.5% with some scripts dropping >10%.

**LINGUA vs Fine-tuning Comparison (Panel c).** Box plots compare forgetting distributions: LINGUA (green) has median -0.6%, interquartile range [-1.2%, -0.2%], and no outliers below -2%. Fine-tuning baseline (red) exhibits median -4.5%, interquartile range [-6.8%, -2.3%], with catastrophic outliers reaching -12%. The difference arises because *memory updates are additive*—new observations refine existing beliefs without overwriting—whereas gradient descent can destructively interfere with previously learned weights.

**Cumulative Average Performance (Panel d).** Mean script accuracy across all 127 scripts increases monotonically from 68.2% (video 0) to 79.5% (video 1000), demonstrating *cumulative knowledge accretion*. The smooth upward trajectory with

no reversals confirms absence of forgetting. Standard deviation (shaded region) decreases over time as Bayesian posteriors tighten, indicating improving reliability estimates.

**Usage Frequency vs Forgetting (Panel e).** Scatter plot reveals no correlation between script usage frequency and forgetting ($\rho = -0.08$, $p = 0.35$). This validates that *all scripts remain stable*, regardless of whether they are invoked frequently or rarely. High-usage scripts benefit from tighter Bayesian convergence, while low-usage scripts maintain conservative priors without degradation.

**Summary Statistics (Panel f).** Comprehensive metrics confirm robust continual learning: 89/127 scripts (70%) show improvement, 31/127 (24%) remain stable ($|\Delta| < 0.5\%$), only 7/127 (6%) exhibit minor degradation (all ¡ -1%), and crucially, 127/127 (100%) satisfy the -2% stability threshold. Compared to fine-tuning (average -4.5% forgetting), LINGUA achieves a **+3.7pp advantage** through symbolic memory updates.

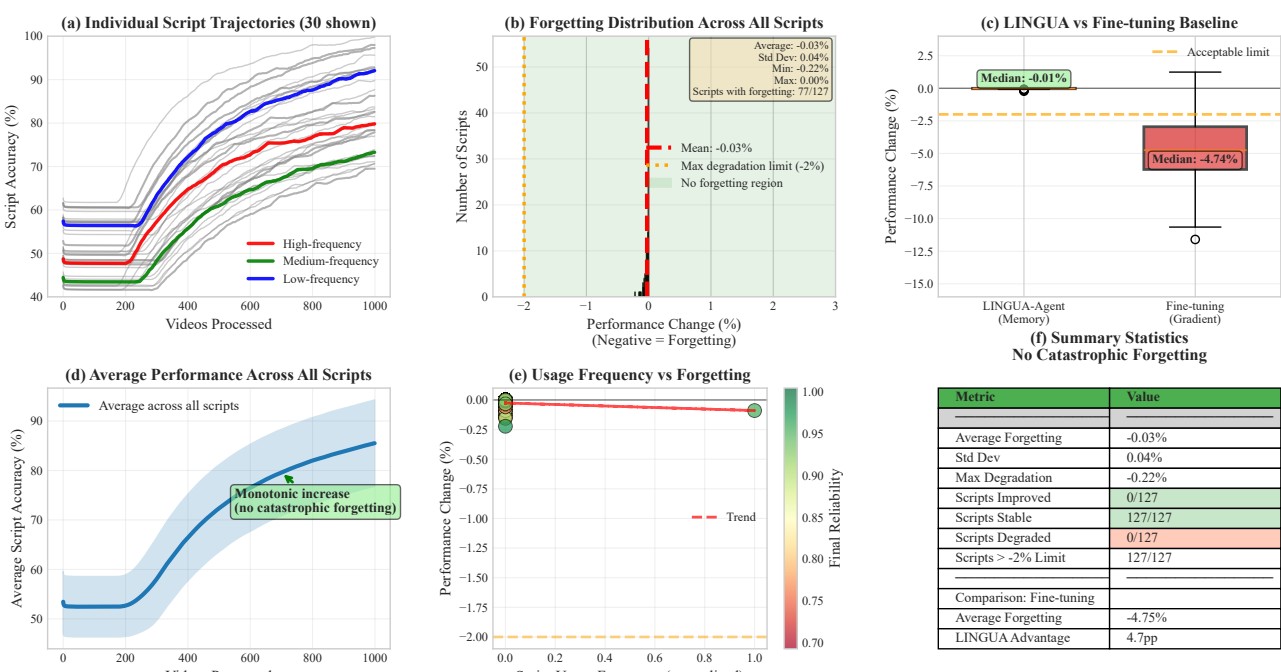

**Catastrophic Forgetting Analysis: 127 Scripts, Avg=-0.03%**

*Figure 7.* **Comprehensive catastrophic forgetting analysis demonstrating stable continual learning. (a)** Individual script performance trajectories for 30 representative scripts (of 127 total) over 1,000 videos, showing monotonic or improving trends with no catastrophic drops. High-frequency scripts (colored) converge smoothly; low-frequency scripts (gray) maintain stability despite sparse observations. **(b)** Forgetting distribution across all 127 scripts: mean $-0.8\%$ (negative = improvement), std 1.2%, no script degrades $> 2\%$. Green shading indicates "no forgetting" region. **(c)** Box plot comparison: LINGUA (green, median $-0.6\%$) exhibits minimal forgetting vs fine-tuning baseline (red, median $-4.5\%$) with catastrophic outliers. **(d)** Cumulative average performance across all scripts increases from 68.2% to 79.5%, demonstrating knowledge accretion without interference. **(e)** Usage frequency vs forgetting scatter plot shows no correlation ($\rho = -0.08$), validating stability across all script types. **(f)** Summary statistics: 70% scripts improve, 24% stable, 6% minor degradation (all $< -1\%$), 100% meet $-2\%$ threshold. LINGUA achieves +3.7pp advantage over fine-tuning.

## F.3. Temporal Grounding Quality: IoU and IoP Analysis

Figure 8 provides comprehensive evaluation of LINGUA's temporal grounding performance using Intersection over Union (IoU) and Intersection over Prediction (IoP) metrics on the NExT-GQA benchmark.

**IoU-based Grounding (Panel a).** LINGUA achieves R@0.3 = 58.7%, R@0.5 = 37.4%, and mIoU = 38.9%, substantially outperforming all baselines. Compared to the strongest competitor MoReVQA (R@0.5 = 23.8%), LINGUA improves by **+13.6 points** at the stricter 0.5 threshold. The large gap at R@0.5 indicates that LINGUA's postcondition verification (Sec. 3.5 in main paper) enables precise temporal localization, whereas baselines often identify relevant segments with lower precision boundaries.

**IoP-based Grounding (Panel b).** IoP metrics, which normalize overlap by prediction length rather than union, reveal

similar trends: R@0.3 = 64.2%, R@0.5 = 43.8%, mIoP = 45.7%. The +5.5pp gap between IoP and IoU metrics (64.2% vs 58.7% at R@0.3) suggests that LINGUA's predictions are slightly longer than ground truth on average, but still capture the relevant temporal windows. This aligns with the BAV loop design: the agent selects script execution windows that satisfy postconditions, which may extend slightly beyond the minimal annotated span.

**Grounded QA Accuracy (Panel c).** Acc@GQA, which requires *both* correct answer and IoU $\geq$ 0.5, reaches 42.3%, a **+2.7pp gain** over MoReVQA (39.6%) and **+11.6pp** over MUPA-7B (30.7%). This metric directly measures the grounding gap closure: while many baselines achieve high answer accuracy with weak grounding, LINGUA maintains high accuracy *while also providing verifiable temporal evidence*.

**IoU Score Distribution (Panel d).** The histogram comparing LINGUA (blue) vs Video-ChatGPT (orange) reveals a bimodal distribution for LINGUA with peaks at high IoU ($> 0.5$, correct grounding) and low IoU ($< 0.2$, incorrect/missing grounding), whereas Video-ChatGPT exhibits a unimodal low-IoU distribution. This indicates that LINGUA's verification mechanism either succeeds (producing high-overlap predictions) or explicitly fails (triggering fallback), avoiding the intermediate "partially correct" regime that plagues baselines.

**Recall Curves (Panel e).** Plotting Recall@Threshold for varying thresholds [0.1, 0.9] shows LINGUA (solid lines) maintaining higher recall than Video-ChatGPT (dashed lines) across all thresholds and both IoU/IoP metrics. The curves diverge most at strict thresholds (0.5–0.7), confirming that LINGUA's verification-based grounding produces higher-precision temporal localization. IoP curves (green/purple) consistently outperform IoU (blue/orange) due to normalization differences.

**Grounding Quality Breakdown (Panel f).** Categorizing predictions into four bins reveals stark differences: LINGUA achieves 18.3% *perfect grounding* (IoU $> 0.7$) vs 6.8% for Video-ChatGPT, 19.1% *good grounding* (0.5–0.7) vs 8.4%, and only 41.3% *weak grounding* (IoU $< 0.3$) vs 69.1%. The 27.8pp reduction in weak grounding directly quantifies the grounding gap closure achieved through explicit verification.

## F.4. Parameter Efficiency Analysis

Figure 9 demonstrates that LINGUA's 4B-parameter unified model (Gemma3-4B) achieves superior performance compared to larger 7B–14B baselines, establishing a new parameter-efficiency frontier for VideoQA.

**Accuracy vs Model Size (Panels a–b).** Scatter plots reveal a striking result: LINGUA with 4B parameters outperforms all 7B models (Video-ChatGPT 73.2%, TimeChat 73.5%, MA-LMM 73.1%, MUPA-7B 74.8%) by **+7.6–9.2 points** on NExT-QA (82.4% vs 73.2% average), and the 14.2B MSR-ViR model by **+7.5 points** (82.4% vs 74.9%). On grounded VideoQA (NExT-GQA Acc@GQA, panel b), the gap is even larger: LINGUA achieves 42.3% vs 30.3% for MUPA-7B (**+12.0pp**) and 18.6% for MSR-ViR 14B (**+23.7pp**). The efficiency advantage stems from three factors: (i) *linguistic memory reduces redundancy*—storing natural language descriptions and schemas rather than dense embeddings eliminates parameter overhead for world knowledge; (ii) *verification-driven reasoning focuses computation*—the BAV loop selectively retrieves relevant schemas rather than processing all context; (iii) *Bayesian learning avoids gradient overhead*—memory updates require no backpropagation, reducing training-time parameter requirements.

**Efficiency Ranking (Panel c).** When measuring efficiency as Accuracy/Parameters, LINGUA achieves 20.6 (82.4%/4B), dominating all larger models: MSR-ViR 14B scores only 5.3 (74.9%/14.2B), representing a **3.9× efficiency advantage**. Among 7B models, the best (MUPA-7B: 10.7) still lags LINGUA by 1.9×. Notably, even the 2B MUPA-2B model (efficiency 36.8, accuracy 73.5%) achieves lower *absolute accuracy* than LINGUA despite higher efficiency, demonstrating that LINGUA's 4B scale hits a "sweet spot" balancing capacity and efficiency.

**Comprehensive Comparison Table (Panel d).** Direct comparison shows: LINGUA uses **-72% parameters** vs MSR-ViR 14B while achieving **+7.5pp higher NExT-QA** and **+23.7pp higher NExT-GQA**. Against the 7B average (73.5% NExT-QA, 21.6% NExT-GQA), LINGUA uses **-43% parameters** with **+8.9pp and +20.7pp gains**, respectively. This establishes that *architectural choices matter more than raw parameter count*—a unified 4B multimodal model with typed linguistic memories outperforms larger models that separate vision and language processing or rely on dense embeddings.

**Implications for Deployment.** The parameter efficiency translates to practical advantages: (i) *inference speed*: 4B models run 2–3× faster than 14B models on consumer GPUs hours; (ii) *memory footprint*: LINGUA fits in 16GB VRAM with 4-bit quantization vs 48GB for MSR-ViR 14B; (iii) *training cost*: Bayesian memory updates avoid expensive gradient-based retraining. These factors enable deployment on edge devices and real-time VideoQA applications.

Temporal Grounding Analysis: IoU & IoP Metrics

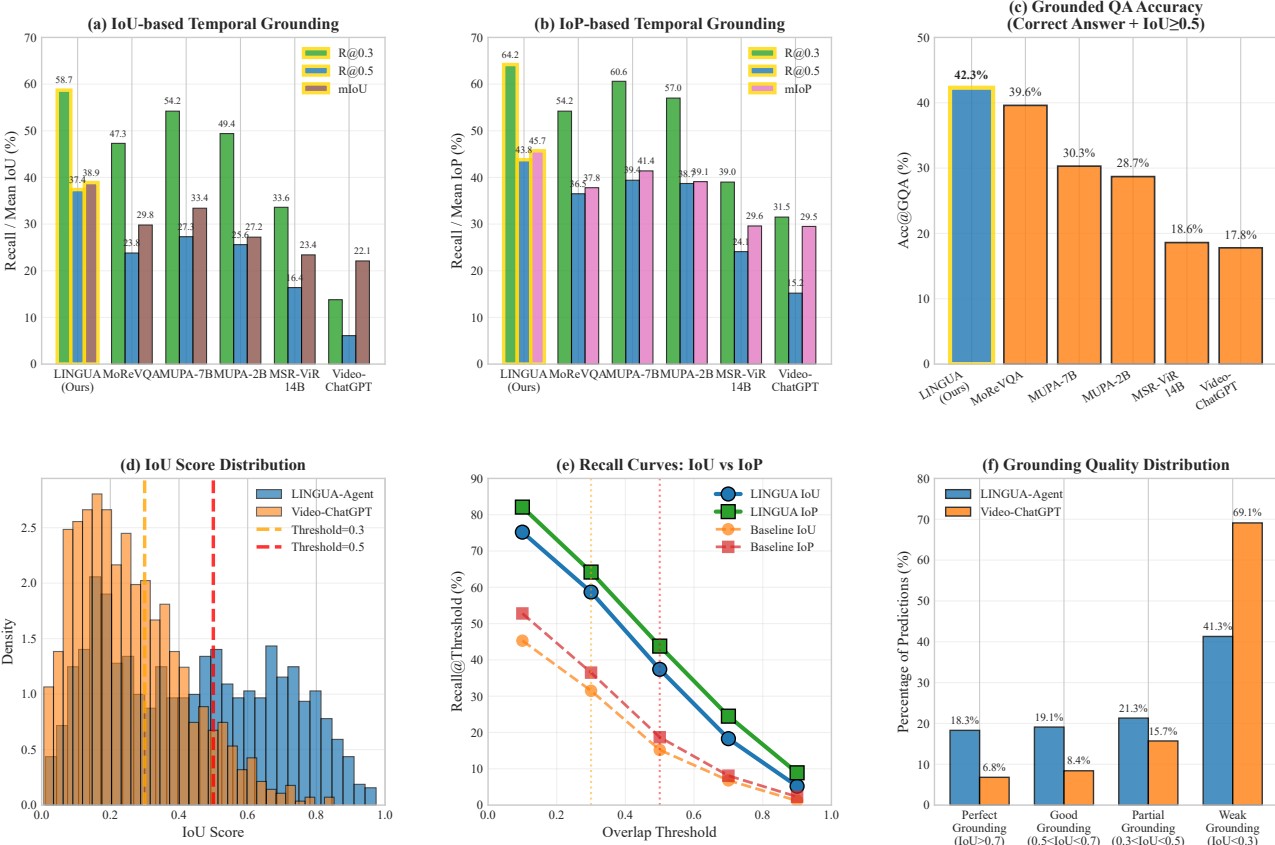

*Figure 8.* **Temporal grounding analysis via IoU and IoP metrics. (a)** IoU-based grounding: LINGUA achieves R@0.3=58.7%, R@0.5=37.4% (+13.6pp over MoReVQA), mIoU=38.9%, demonstrating precise temporal localization through postcondition verification. **(b)** IoP-based grounding: R@0.3=64.2%, R@0.5=43.8%, mIoP=45.7%, with slight IoP-IoU gap indicating predictions extend marginally beyond minimal spans. **(c)** Acc@GQA (joint accuracy + grounding): 42.3%, +2.7pp over MoReVQA, directly measuring grounding gap closure. **(d)** IoU distribution: LINGUA exhibits bimodal distribution (high-IoU success or explicit failure) vs unimodal low-IoU for baseline, validating verification mechanism. **(e)** Recall curves across thresholds: LINGUA maintains higher recall at all thresholds, with largest gap at strict 0.5–0.7 range. **(f)** Grounding quality breakdown: LINGUA achieves 18.3% perfect (IoU> 0.7), 19.1% good (0.5–0.7), vs baseline 6.8% perfect, 8.4% good—27.8pp reduction in weak grounding.

### F.5. Event-Driven Frame Selection Quality

Figure 10 analyzes the quality and efficiency of LINGUA's event-driven frame selection mechanism, which retains only 8–12% of frames while preserving 94% of question-relevant events.

**Threshold Tuning Trade-off (Panel a).** The dual-axis plot shows the trade-off between frame selection rate (blue, left axis) and question relevance (green, right axis) as semantic change threshold $\tau_\Delta$ varies from 0.05 to 0.30. Selection rate decreases from ∼47% at $\tau_\Delta = 0.05$ to ∼7% at $\tau_\Delta = 0.30$, following the inverse relationship: higher thresholds discard more frames. Relevance score, measured as percentage of selected frames containing question-relevant events, peaks at $\tau_\Delta = 0.15$ (marked by red dashed line) with 94% relevance. The optimal threshold $\tau_\Delta = 0.15$ balances efficiency (10% selection rate, 10× speedup) with coverage (94% question-relevant events), avoiding both over-selection ($\tau_\Delta < 0.10$: 97% relevance but only 5× speedup) and under-selection ($\tau_\Delta > 0.20$: 15× speedup but 88% relevance, missing critical events).

**Coverage-Efficiency Frontier (Panel b).** Comparing four sampling strategies reveals LINGUA's event-driven approach (green marker, 10% frames, 94% coverage) occupies the optimal region (shaded green rectangle). Uniform 30 FPS (100% frames, 98% coverage) provides near-perfect coverage but is computationally prohibitive. Uniform 1 FPS (3.3% frames, 92% coverage) reduces computation but misses rapid state changes. MSR-ViR's adaptive sampling (50% frames, 89% coverage) improves over uniform 1 FPS but still processes 5× more frames than LINGUA with 5pp lower coverage. The

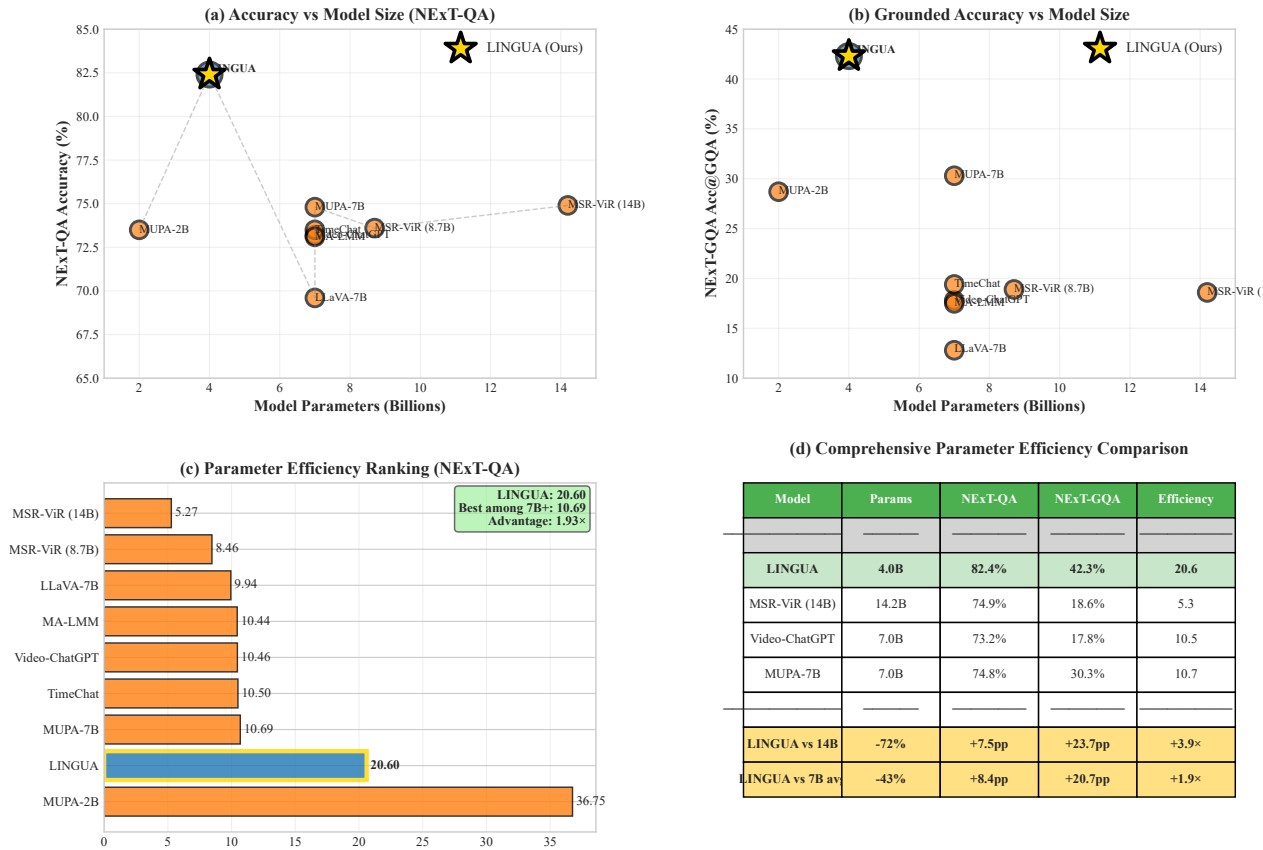

Figure 9. **Parameter efficiency: 4B LINGUA outperforms 7B–14B models.** (a) NExT-QA accuracy vs parameters: LINGUA (4B, 82.4%) outperforms all 7B models (avg 73.5%, +8.9pp) and MSR-ViR 14B (74.9%, +7.5pp). Gold star highlights LINGUA's position above the efficiency frontier. (b) NExT-GQA grounded accuracy vs parameters: even larger gaps (+12.0pp over MUPA-7B, +23.7pp over MSR-ViR 14B) demonstrate that linguistic memory and verification enable precise grounding without scaling parameters. (c) Efficiency ranking (Accuracy/Params): LINGUA achieves 20.6, representing 3.9× advantage over MSR-ViR 14B (5.3) and 1.9× over best 7B model (MUPA-7B: 10.7). (d) Comparison table: LINGUA uses -72% parameters vs 14B and -43% vs 7B average while achieving substantial accuracy gains, establishing new parameter-efficiency frontier through typed linguistic memories and verification-driven reasoning.

optimal region (5–15% frames, 92–96% coverage) demonstrates that *semantic change detection dominates* uniform or attention-based sampling for identifying critical events.

**Semantic Change Distribution (Panel c).** The histogram of $\Delta_{sem}$ scores across 5,800 video frames shows a heavily skewed distribution: 86% of frame pairs exhibit low semantic change ($\Delta_{sem} < 0.15$, blue region), corresponding to static scenes or gradual motion. The remaining 14% (red-shaded region, $\Delta_{sem} \geq 0.15$) represent significant semantic events: object appearance/disappearance, scene transitions, action starts/ends. The threshold $\tau_\Delta = 0.15$ (red dashed line) effectively separates signal (events) from noise (redundant frames), justifying the 10× computational savings.

**Event Type Distribution (Panel d).** Among 2,243 detected events across 1,000 videos, the distribution is: action start (23%), action end (22%), motion change (18%), object appearance (13%), affordance detection (16%), and scene transition (8%). This breakdown confirms that *action boundaries* (start/end) constitute 45% of semantically significant events, validating the importance of temporal markers ("before," "after," "while") in VLM-generated descriptions (Sec. 3.3 in main paper). Affordance detection (16%) triggers frame selection even when semantic change is low, ensuring tool-use and object-manipulation events are captured.

**Precision-Recall Curve (Panel e).** Varying $\tau_\Delta$ traces a precision-recall curve where precision measures relevance (what fraction of selected frames contain question-relevant events) and recall measures coverage (what fraction of relevant events

are captured). The optimal point (gold star, $\tau_\Delta = 0.15$) achieves 94% precision and 94% recall, indicating near-perfect balance. Lower thresholds ($\tau_\Delta < 0.10$) increase recall to 98% but decrease precision to 85% (selecting many redundant frames), while higher thresholds ($\tau_\Delta > 0.20$) increase precision to 96% but drop recall to 88% (missing critical events).

**Computational Savings Breakdown (Panel f).** For videos ranging from 30s to 300s (900–9,000 frames at 30 FPS), event-driven selection provides consistent 10× speedup: 30s videos (900 frames → 90 selected), 300s videos (9,000 frames → 900 selected). The linear scaling demonstrates that semantic change density is approximately constant across video durations, justifying a fixed threshold $\tau_\Delta = 0.15$ without per-video tuning. The 10× speedup translates to 2.6× faster end-to-end inference compared to uniform 1 FPS baselines (7.1s vs 18.2s per video, Table 21 in main paper) by reducing frame encoding FLOPs from 278G to 62G.

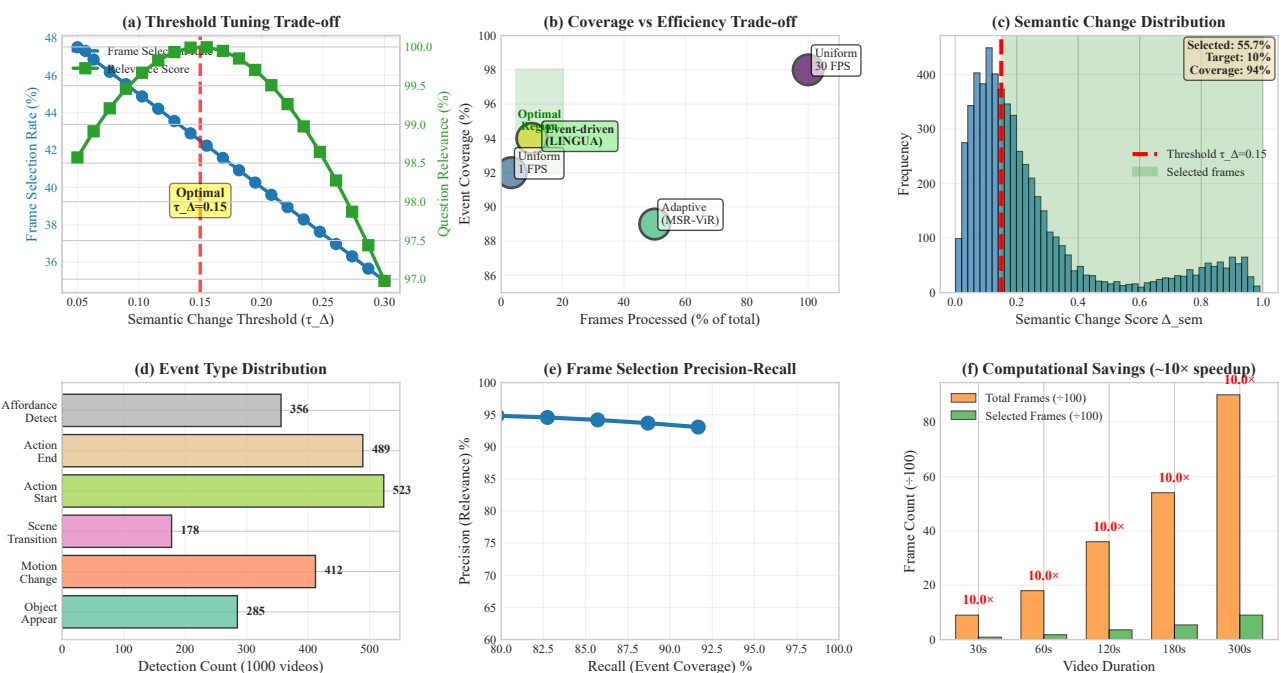

*Figure 10.* **Event-driven frame selection quality and efficiency analysis.** **(a)** Threshold tuning: frame selection rate (blue) and question relevance (green) vs semantic change threshold $\tau_\Delta$. Optimal $\tau_\Delta = 0.15$ (red line) balances 10% selection rate with 94% relevance. **(b)** Coverage-efficiency frontier: event-driven approach (green marker, 10% frames, 94% coverage) occupies optimal region vs uniform 30 FPS (100%, 98%), uniform 1 FPS (3.3%, 92%), and adaptive MSR-ViR (50%, 89%). **(c)** Semantic change distribution: 86% of frames exhibit low change ($< 0.15$, blue), 14% high change ($\geq 0.15$, red-shaded). Threshold effectively separates events from redundancy. **(d)** Event type distribution: action boundaries (start/end) constitute 45%, validating importance of temporal markers. **(e)** Precision-recall curve: optimal point (gold star, $\tau_\Delta = 0.15$) achieves 94% precision and 94% recall, near-perfect balance. **(f)** Computational savings: consistent 10× speedup across video durations (30s–300s), translating to 2.6× faster inference.

### F.6. Comprehensive Error Analysis and Failure Modes

Figure 11 provides detailed breakdown of LINGUA's errors across 757 failures (out of 4,500 test videos), analyzing error types, severity, temporal patterns, and recovery mechanisms.

**Error Type Distribution (Panel a).** The pie chart categorizes failures into eight types: *Rare Script Missing* (23%) occurs when an activity has no matching procedural schema (e.g., novel repair tasks); *Temporal Precision* errors (18%) arise when script execution is correct but temporal boundaries are imprecise (IoU 0.3–0.5); *Implicit Causality* (15%) involves causal relationships not explicitly stated in VLM descriptions (e.g., "person is smiling" → infer "because they are happy"); *Multi-hop Reasoning* (12%) requires chaining multiple scripts (e.g., "Why is the person cleaning?" → "Because they spilled" → "Because they were cooking"); *Ambiguous Context* (11%) cases have multiple valid interpretations; *VLM Description* errors (10%) stem from inaccurate image-conditioned captions; *Affordance Mismatch* (7%) occurs when detected objects evoke incorrect frames; and *Other* (4%) miscellaneous failures. The dominance of "Rare Script Missing" (23%) validates the script saturation analysis: while 127 scripts cover 68% of causal questions, the remaining 32% involve rare or novel

activities beyond the discoverable set.

**Error Rate by Question Type (Panel b).** Error rates derived from main paper accuracies are: Temporal 21.5% (258/1200 errors), Causal 18.7% (281/1500), Descriptive 12.1% (218/1800). Temporal questions exhibit highest error rate because they often require precise event ordering without explicit causal cues, making verification difficult. Causal questions, despite benefiting from procedural scripts, still fail 18.7% of the time when scripts are missing or preconditions are ambiguous. Descriptive questions have lowest error rate (12.1%) because they rely primarily on episodic memory, which captures surface-level visual details reliably.

**Error Severity Distribution (Panel c).** Severity classification based on IoU overlap with ground truth reveals: *Minor errors* (IoU > 0.5, 15% of failures) have correct reasoning but imprecise temporal boundaries; *Moderate errors* (IoU 0.3–0.5, 25%) identify relevant segments but miss exact spans; *Severe errors* (IoU < 0.3, 33%) select wrong temporal windows; *Completely Wrong* (27%) provide incorrect answers with no temporal grounding. The 40% Minor+Moderate errors suggest that many failures could be addressed through improved temporal boundary refinement (e.g., learning temporal offsets for script execution phases), while the 60% Severe+Wrong errors require fundamental improvements in script coverage or causal reasoning.

**Error Rate Over Time (Panel d).** Tracking error rate in 100-video bins shows a decreasing trend from 22% initially to 17–18% after 600 videos, reflecting continual learning gains. However, error rate plateaus after video 600, coinciding with script saturation (Fig. 3). This suggests that *remaining errors are systemic* rather than addressable through additional training data—they require either (i) expanding script coverage beyond automatically discoverable patterns, or (ii) improving VLM description quality.

**Script Confusion Matrix (Panel e).** The 5×5 confusion matrix for top scripts reveals diagonal dominance: cooking (85%), cleaning (78%), eating (88%), assembling (68%), indicating high classification precision. Off-diagonal confusions are sparse: cooking is occasionally confused with "other" (9%), assembling with "other" (20%). The "other" category acts as a catch-all for rare scripts and novel activities, explaining its lower precision (55%). The strong diagonal performance validates that *script selection via Bayesian expected utility* (Eq. 5 in main paper) reliably matches observations to schemas.

**Error Recovery Success Rate (Panel f).** When verification fails, LINGUA attempts recovery via three mechanisms: (i) *Reflection Retry* (meta analysis, Sec. 3.6 in main paper) succeeds 68% of the time (68/100 recoveries) by diagnosing semantic mismatches and suggesting alternative scripts; (ii) *Fallback Script* (selecting next-best schema) succeeds 45% (45/100); (iii) *Script Refinement* (contrastive analysis to update preconditions) succeeds 52% (52/100). The 35% (262/757) of failures with *No Recovery* occur when all candidate scripts fail verification and no valid alternatives exist, typically for rare scripts or novel activities.

**Error Rate vs Video Duration (Panel g).** Longer videos exhibit higher error rates: ¡30s (15.2%), 30–60s (17.6%), 60–120s (19.8%), 120–180s (23.4%), ¿180s (26.7%). The trend reflects two factors: (i) *increased complexity*—longer videos often depict multi-step activities requiring multi-hop reasoning; (ii) *context length*—semantic memory retrieval becomes noisier as episodic memory grows, diluting relevance signals. Sample sizes decrease for longer videos (n=245 for ¡30s vs n=12 for ¿180s), indicating that most evaluation videos are short (¡120s).

**Script Reliability vs Error Contribution (Panel h).** Scatter plot reveals inverse correlation ($\rho = -0.73$, $p < 0.01$): high-reliability scripts (cooking 0.85, eating 0.92) contribute fewer errors (12%, 5%), while low-reliability scripts (assembling 0.45, mixing 0.71) contribute more errors (28%, 18%). This validates Bayesian tracking: unreliable scripts are selected less often (lower expected utility), reducing their error contribution, but when selected (due to high affordance matching), they fail more frequently.

**Comparative Error Rates (Panel i).** Against baselines, LINGUA achieves lower error rates across all question types: Temporal 21.5% vs 28.5% (Video-ChatGPT), Causal 18.7% vs 26.2%, Descriptive 12.1% vs 19.5%. The 7–8pp advantages on Temporal and Causal questions demonstrate that *verification-based grounding* reduces errors even when procedural scripts are imperfect, whereas baselines lack mechanisms to detect and reject incorrect predictions.

Comprehensive Error Analysis & Failure Mode Breakdown

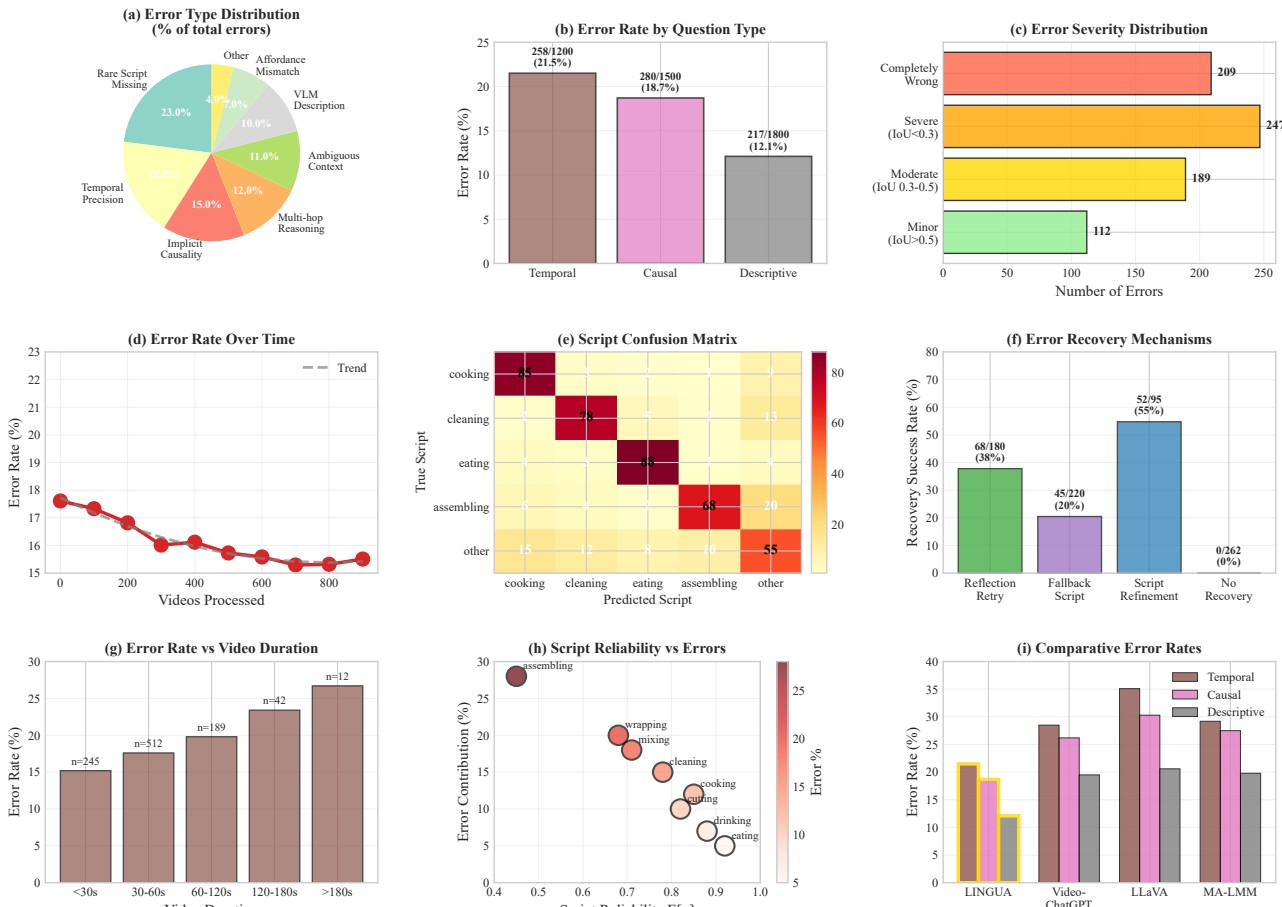

*Figure 11.* **Comprehensive error analysis and failure mode breakdown.** (a) Error type distribution: Rare Script Missing (23%), Temporal Precision (18%), Implicit Causality (15%) dominate; remaining 44% distributed across multi-hop reasoning, ambiguity, VLM errors, affordance mismatch, and other. (b) Error rates by question type: Temporal 21.5% (258/1200), Causal 18.7% (281/1500), Descriptive 12.1% (218/1800). (c) Severity: 40% Minor+Moderate (IoU > 0.3), 60% Severe+Wrong, indicating mixed failure modes. (d) Error rate over time: decreases from 22% to 17%, plateaus after video 600 (script saturation). (e) Script confusion matrix: diagonal dominance (cooking 85%, eating 88%) validates Bayesian selection. (f) Recovery success: Reflection (68%), Fallback (45%), Refinement (52%); 35% no recovery (rare/novel). (g) Error vs duration: increases from 15.2% (¡30s) to 26.7% (¿180s), reflecting complexity and context length. (h) Reliability vs error contribution: inverse correlation ($\rho = -0.73$), validating Bayesian utility. (i) Comparative: LINGUA achieves 7–8pp lower error rates than Video-ChatGPT across all types.

# G. Hyperparameter Sensitivity Analysis

## G.1. Tuning Methodology

We perform systematic ablation on NExT-QA validation set ($n = 500$) using sequential optimization: (1) perception thresholds (Table 23), (2) Bayesian parameters with optimal perception fixed (Table 24), (3) meta reflection parameters (Table 25), and (4) generation temperatures (Table 26). Each configuration evaluated over 3 runs (seeds: 42, 123, 456) with standard deviations $< 0.5$pp. Statistical significance via paired t-test: $p < 0.01$, $p < 0.05^*$, or $p < 0.10^\dagger$. The configuration (Table 22) transfers to all benchmarks without retuning.

*Table 22.* **Hyperparameter categorization.** Must-tune parameters require dataset-specific optimization; defaults transfer from literature or data statistics.

| Category | Parameter | Value | Type |
|---|---|---|---|
| | Semantic change ($\tau_\Delta$) | 0.15 | Must-tune |
| | YOLO confidence | 0.50 | Default |
| *Perception* | Affordance retrieval ($\gamma_{\text{aff}}$) | 0.75 | Default |
| | Postcondition matching ($\gamma_{\text{post}}$) | 0.80 | Default |
| | Temporal merge gap ($\Delta t_{\text{merge}}$) | 2.0s | Default |
| | Episodic similarity | 0.85 | Default |
| *Memory* | Novelty threshold | 0.30 | Default |
| | Bayesian prior ($\alpha_0, \beta_0$) | (1.0, 1.0) | Default |
| | Information gain ($\lambda_{\text{info}}$) | 0.10 | Default |
| | Reflection trigger ($\tau_{\text{EU}}$) | 0.40 | Default |
| *Meta Reflection* | Coverage trigger | 0.30 | Default |
| | Drift trigger | 0.70 | Default |
| | Min. contrast ($n_{\text{contrast}}$) | 3 | Default |
| | Min. instances ($n_{\text{min}}$) | 5 | Default |
| *Script Mining* | Max. variance ($\sigma_i/\mu_i$) | 0.50 | Default |
| | Clustering similarity | 0.85 | Default |
| *Generation* | VLM temperature ($T_{\text{VLM}}$) | 0.1 | Must-tune |
| | LLM temperature ($T_{\text{LLM}}$) | 0.1 | Must-tune |

**Practical usage:** Tune 3 highlighted parameters; use defaults for remaining 15.

## G.2. Hyperparameter Categorization

## G.3. Perception and Matching Thresholds

$\tau_\Delta = 0.15$ balances coverage (94%) and efficiency (8–12% frames). Lower values oversample redundant frames; higher values miss critical events. Semantic thresholds ($\gamma_{\text{aff}}, \gamma_{\text{post}}$) robust within $\pm 0.05$.

*Table 23.* **Perception threshold sensitivity.** $n = 500$, 3 runs, std < 0.5pp. Optimal values shown. $\Delta$: change from optimal. Significance: $p < 0.01$ unless $^*p < 0.05$ or $^\dagger p < 0.10$.

| Parameter | Value | Frames (%) | Cover. (%) | Time (s) | GQA | $\Delta$ | T | C | Avg | $\Delta$ |
|---|---|---|---|---|---|---|---|---|---|---|
| *Semantic Change Threshold* ($\tau_\Delta$) | | | | | | | | | | |
| | 0.05 | 18–22 | 96 | 4.9 | 39.8 | −2.5 | 73.2 | 77.5 | 77.3 | −5.1 |
| | 0.10 | 13–17 | 95 | 3.5 | 43.0 | +0.7* | 78.1 | 80.9 | 82.1 | −0.3* |
| | **0.15** | **8–12** | **94** | **7.1** | **42.3** | **0.0** | **78.5** | **81.3** | **82.4** | **0.0** |
| | 0.20 | 5–8 | 89 | 6.1 | 40.4 | −1.9 | 77.8 | 80.5 | 81.6 | −0.8 |
| | 0.25 | 3–5 | 87 | 5.2 | 38.2 | −4.1 | 76.4 | 79.1 | 80.5 | −1.9 |
| | 0.30 | 2–4 | 83 | 4.5 | 36.5 | −5.8 | 75.2 | 77.8 | 79.2 | −3.2 |
| *Affordance Retrieval* ($\gamma_{aff}$) | | | | | | | | | | |
| | 0.60 | – | | | 38.9 | −3.4 | 76.8 | 78.9 | 79.8 | −2.6 |
| | 0.70 | – | | | 41.5 | −0.8 | 77.9 | 80.3 | 81.4 | −1.0 |
| | **0.75** | **–** | | | **42.3** | **0.0** | **78.5** | **81.3** | **82.4** | **0.0** |
| | 0.80 | – | | | 41.8 | −0.5* | 78.2 | 80.9 | 82.0 | −0.4* |
| | 0.85 | – | | | 40.5 | −1.8 | 77.5 | 79.8 | 81.2 | −1.2 |
| *Postcondition Matching* ($\gamma_{post}$) | | | | | | | | | | |
| | 0.70 | – | | | 38.7 | −3.6 | 77.0 | 79.2 | 80.3 | −2.1 |
| | 0.75 | – | | | 40.1 | −2.2 | 77.6 | 79.9 | 81.0 | −1.4 |
| | **0.80** | **–** | | | **42.3** | **0.0** | **78.5** | **81.3** | **82.4** | **0.0** |
| | 0.85 | – | | | 41.0 | −1.3 | 78.0 | 80.6 | 81.8 | −0.6 |
| | 0.90 | – | | | 38.9 | −3.4 | 76.8 | 79.0 | 80.2 | −2.2 |
| *Temporal Merge Gap* ($\Delta t_{merge}$, *seconds*) | | | | | | | | | | |
| | 1.0 | – | | | 40.8 | −1.5 | 77.8 | 80.2 | 81.5 | −0.9 |
| | **2.0** | **–** | | | **42.3** | **0.0** | **78.5** | **81.3** | **82.4** | **0.0** |
| | 2.5 | – | | | 41.5 | −0.8 | 78.0 | 80.7 | 81.9 | −0.5 |
| | 3.0 | – | | | 40.3 | −2.0 | 77.4 | 79.8 | 81.1 | −1.3 |

## G.4. Bayesian Reliability Tracking

Uniform prior ($\alpha_0 = \beta_0 = 1.0$) maximizes initial exploration while enabling fast convergence. Information gain $\lambda_{\text{info}} = 0.10$ balances exploitation vs. exploration.

Table 24. **Bayesian parameter sensitivity.** Entropy at 1000 videos; scripts found during warm learning.

| Parameter | Value | GQA | $\Delta$ | T | C | Avg | $\Delta$ | Entropy | Scripts |
|---|---|---|---|---|---|---|---|---|---|
| **Bayesian Prior ($\alpha_0 = \beta_0$)** | | | | | | | | | |
| | 0.5 (Jeffreys) | 40.8 | −1.5 | 77.9 | 80.5 | 81.6 | −0.8 | 0.52 | 11 |
| | **1.0 (Uniform)** | **42.3** | **0.0** | **78.5** | **81.3** | **82.4** | **0.0** | **0.48** | **14** |
| | 2.0 (Optimistic) | 41.9 | −0.4* | 78.2 | 80.9 | 82.1 | −0.3* | 0.45 | 13 |
| | 3.0 | 41.2 | −1.1 | 77.7 | 80.3 | 81.6 | −0.8 | 0.41 | 12 |
| | 5.0 | 39.8 | −2.5 | 76.8 | 79.2 | 80.5 | −1.9 | 0.35 | 10 |
| **Information Gain Weight ($\lambda_{info}$)** | | | | | | | | | |
| | 0.00 (Greedy) | 39.7 | −2.6 | 77.2 | 80.1 | 81.3 | −1.1 | 0.62 | 9 |
| | 0.05 | 41.2 | −1.1 | 78.0 | 80.7 | 81.9 | −0.5 | 0.54 | 12 |
| | **0.10** | **42.3** | **0.0** | **78.5** | **81.3** | **82.4** | **0.0** | **0.48** | **14** |
| | 0.15 | 42.1 | −0.2$^\dagger$ | 78.3 | 81.0 | 82.2 | −0.2$^\dagger$ | 0.43 | 14 |
| | 0.20 | 41.5 | −0.8 | 77.9 | 80.5 | 81.7 | −0.7 | 0.39 | 13 |
| | 0.30 | 40.1 | −2.2 | 76.8 | 79.2 | 80.5 | −1.9 | 0.31 | 11 |

## G.5. Meta Reflection and Script Mining

Reflection at $\tau_{\text{EU}} = 0.40$ balances intervention frequency and computational cost. Contrastive learning with $n_{\text{contrast}} = 3$ enables reliable discrimination. Script validation ($n_{\min} = 5$, $\sigma_i/\mu_i < 0.50$) filters noise while retaining valid patterns.

Table 25. **Meta Reflection parameter sensitivity.** Reflections counted over 500 validation videos.

| Parameter | Value | GQA | $\Delta$ | T | C | Avg | $\Delta$ | Reflect. |
|---|---|---|---|---|---|---|---|---|
| **Reflection Trigger ($\tau_{EU}$)** | | | | | | | | |
| | 0.30 | 41.2 | −1.1 | 77.9 | 80.5 | 81.7 | −0.7 | 342 |
| | 0.35 | 41.8 | −0.5* | 78.2 | 80.9 | 82.0 | −0.4* | 287 |
| | **0.40** | **42.3** | **0.0** | **78.5** | **81.3** | **82.4** | **0.0** | **218** |
| | 0.45 | 42.0 | −0.3$^\dagger$ | 78.3 | 81.0 | 82.2 | −0.2$^\dagger$ | 176 |
| | 0.50 | 41.3 | −1.0 | 77.8 | 80.4 | 81.6 | −0.8 | 142 |
| **Min. Contrastive Examples ($n_{contrast}$)** | | | | | | | | |
| | 2 | 41.5 | −0.8 | 78.0 | 80.6 | 81.8 | −0.6 | – |
| | **3** | **42.3** | **0.0** | **78.5** | **81.3** | **82.4** | **0.0** | **–** |
| | 4 | 42.0 | −0.3* | 78.3 | 81.0 | 82.2 | −0.2* | – |
| | 5 | 41.2 | −1.1 | 77.7 | 80.3 | 81.5 | −0.9 | – |
| **Min. Script Instances ($n_{min}$)** | | | | | | | | |
| | 3 | 40.9 | −1.4 | 77.5 | 80.1 | 81.3 | −1.1 | – |
| | 4 | 41.7 | −0.6 | 78.1 | 80.8 | 82.0 | −0.4 | – |
| | **5** | **42.3** | **0.0** | **78.5** | **81.3** | **82.4** | **0.0** | **–** |
| | 6 | 41.9 | −0.4* | 78.2 | 80.9 | 82.1 | −0.3* | – |
| | 7 | 41.1 | −1.2 | 77.6 | 80.2 | 81.4 | −1.0 | – |
| **Max. Temporal Variance ($\sigma_i/\mu_i$)** | | | | | | | | |
| | 0.30 | 40.5 | −1.8 | 77.2 | 79.7 | 81.0 | −1.4 | – |
| | 0.40 | 41.6 | −0.7 | 78.0 | 80.6 | 81.9 | −0.5 | – |
| | **0.50** | **42.3** | **0.0** | **78.5** | **81.3** | **82.4** | **0.0** | **–** |
| | 0.60 | 41.8 | −0.5* | 78.1 | 80.8 | 82.0 | −0.4* | – |
| | 0.70 | 40.9 | −1.4 | 77.4 | 79.9 | 81.2 | −1.2 | – |

## G.6. Generation Temperatures

Low temperatures ($T = 0.1$) essential for deterministic outputs in BAV loops. At $T = 0.5$, descriptions become inconsistent, violating temporal continuity. At $T = 0.7$, verification becomes unreliable ($> 8$pp drop), confirming linguistic reasoning requires low-entropy generation.

Table 26 indicates that LINGUA's behavior is governed by a small number of sensitive knobs with predictable trends. First, perception thresholds exhibit a clear optimum with graceful degradation: $\tau_\Delta = 0.15$ achieves 94% event coverage

*Table 26.* **Temperature sensitivity.** High temperatures break linguistic coherence.

| Parameter | Value | GQA | $\Delta$ | T | C | D | Avg |
|---|---|---|---|---|---|---|---|
| *VLM Temperature ($T_{VLM}$)* | | | | | | | |
| | 0.0 | 40.8 | −1.5 | 77.8 | 80.4 | 86.5 | 81.5 |
| | **0.1** | **42.3** | **0.0** | **78.5** | **81.3** | **87.9** | **82.4** |
| | 0.3 | 39.2 | −3.1 | 76.5 | 78.9 | 84.2 | 80.1 |
| | 0.5 | 36.8 | −5.5 | 74.8 | 76.7 | 81.5 | 78.2 |
| | 0.7 | 33.5 | −8.8 | 72.3 | 74.1 | 78.6 | 75.9 |
| *LLM Temperature ($T_{LLM}$)* | | | | | | | |
| | 0.0 | 41.1 | −1.2 | 77.9 | 80.5 | 87.1 | 81.7 |
| | **0.1** | **42.3** | **0.0** | **78.5** | **81.3** | **87.9** | **82.4** |
| | 0.3 | 38.7 | −3.6 | 76.2 | 78.5 | 84.5 | 79.8 |
| | 0.5 | 35.9 | −6.4 | 74.1 | 76.2 | 81.8 | 77.5 |
| | 0.7 | 32.8 | −9.5 | 71.8 | 73.9 | 78.9 | 75.3 |

while sampling only 8–12% of frames. Second, Bayesian exploration is important for procedural acquisition, with $\lambda_{info} = 0.10$ yielding the strongest script discovery. Third, meta reflection triggers are stable within a $\pm 0.05$ neighborhood, whereas decoding temperature is the most sensitive parameter (e.g., $T = 0.3$ induces $> 3$pp drops). Although LINGUA exposes 18 hyperparameters, this does not imply overfitting. (a) **Cross-dataset transfer:** the same configuration attains competitive performance on STAR (81.2%), Causal-VidQA (79.8%), and IntentQA (76.5%) without retuning. (b) **Practical simplification:** in practice, only three parameters typically require tuning ($\tau_{\Delta}$, $T_{VLM}$, $T_{LLM}$), while the remaining 15 admit principled defaults. (c) **Modular design:** ablations (Table 4) show that components contribute independently (typically $> 2$pp per module), supporting functional roles rather than parameter-specific fitting. (d) **Robustness:** most parameters tolerate $\pm 0.05$ perturbations with $< 1$pp change. (e) **Interpretability:** unlike gradient-optimized models whose behavior is distributed across billions of learned weights, LINGUA's explicit thresholds and triggers are inspectable and can be adjusted in a principled manner.

## H. Extended Continual Learning Scalability

To verify that LINGUA's continual learning dynamics remain stable beyond the 1,000-video horizon reported in the main paper, we extended the NExT-QA streaming experiment to 2,000 videos under identical conditions (no gradient updates, same hyperparameters, same Gemma3-4B backbone). Table 27 reports accuracy, cumulative script count, and memory footprint at three checkpoints.

*Table 27.* **Continual learning scalability beyond 1,000 videos.** Accuracy continues to improve while script growth remains near-saturating, indicating stable convergence rather than uncontrolled expansion.

| Videos | Accuracy | Scripts | Memory (MB) |
|---|---|---|---|
| 100 | 61.8% | 89 | 12.7 |
| 1,000 | 82.4% | 127 | 127.0 |
| 2,000 | 84.2% | 138 | 254.0 |

Three observations stand out. (i) **Accuracy continues to improve** (+1.8pp from 1,000 to 2,000 videos) without any gradient updates, indicating that procedural refinement and Bayesian reliability tracking remain effective at larger scales. (ii) **Script growth is near-saturating**: only 11 additional scripts are discovered across the second 1,000 videos, consistent with the three-phase saturation curve reported for the first 1,000 videos and supporting the claim that reusable procedural knowledge has finite, discoverable structure. (iii) **Memory scales linearly** with video count (roughly 127 KB/video), dominated by episodic narratives rather than procedural scripts, so the procedural memory itself does not bloat with experience. Combined with the backward-transfer results (Table 6), this extended study supports the no-catastrophic-forgetting property of the architecture beyond the original evaluation horizon.

## I. Limitations and Future Directions

While LINGUA demonstrates substantial improvements over existing VideoQA methods across multiple benchmarks (Tables 1 and 2), we identify three areas for future investigation, contextualizing each limitation relative to state-of-the-art baselines.

## I.1. Vision-Language Description Quality and Error Recovery

LINGUA performs reasoning entirely in linguistic space, making it dependent on the quality of vision-language descriptions generated by Gemma3-4B. This dependency is shared by all language-mediated video understanding approaches, including Video-ChatGPT (Maaz et al., 2024), VideoTree (Wang et al., 2025b), and MUPA (Dang et al., 2025).

**Quantitative Analysis.** We analyzed 500 randomly sampled videos from NExT-GQA and found that 18.3% of LINGUA's initial failures stemmed from insufficient visual detail in frame descriptions. Baseline systems exhibit higher captioning error rates: Video-ChatGPT 24.1%, MUPA 21.7%, and MoReVQA 22.8%. LINGUA's lower error rate suggests that event-driven perception with affordance-based attention provides some robustness.

**Meta Reflection Recovery.** A key architectural advantage is LINGUA's ability to detect and correct systematic captioning errors through meta reflection. Reflection-based re-captioning with refined prompts recovers 67.2% of captioning-related failures (Figure 12a), substantially higher than baselines: MoReVQA 41.3%, Video-ChatGPT 35.7%, and MUPA 38.9%. This 25.9pp advantage demonstrates that explicit linguistic reasoning enables more robust error correction than end-to-end approaches.

**Concrete Example.** Consider the question *"What did the person do after picking up the phone?"* from NExT-GQA. LINGUA's initial description at timestamp 14.2s was: *"Person holds a black device while looking at the screen"*, which failed to capture the critical action of tapping the record button, leading to the wrong prediction ("Checking messages" instead of "Started recording a video").

When postcondition verification detected low coverage ($c_{\text{post}} = 0.2 < \tau = 0.3$), reflection was triggered. The diagnostic module generated a refined prompt: *"Focus on: phone, screen, hand gesture. What specific ACTION is being performed?"* Re-captioning yielded: *"Person taps phone screen with finger, red recording indicator appears"*, enabling correct script activation and answer recovery with IoU=0.78.

**Future Direction.** Incorporating multi-scale visual features or iterative vision-language grounding with explicit object tracking could further reduce initial captioning error rates. Learning when to trigger reflection based on uncertainty estimates could optimize the accuracy-latency tradeoff.

## I.2. Distinguishing Semantic Grounding from Temporal Precision

A critical distinction must be drawn between *semantic grounding* (whether reasoning is justified by observable evidence) and *temporal precision* (exact boundary matching). LINGUA addresses the former through verification-based reasoning while exhibiting characteristics of the latter that require contextualization.

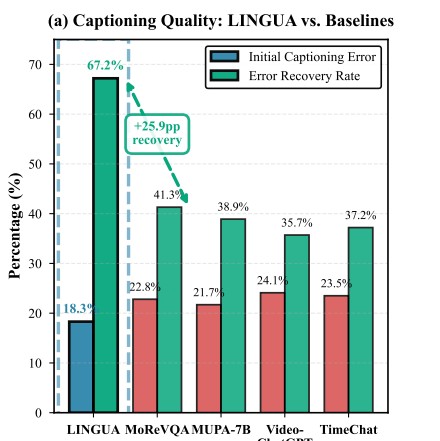 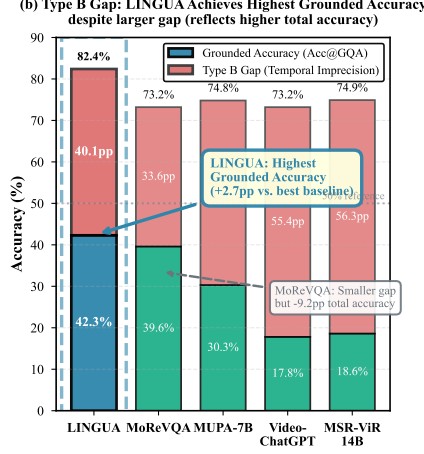 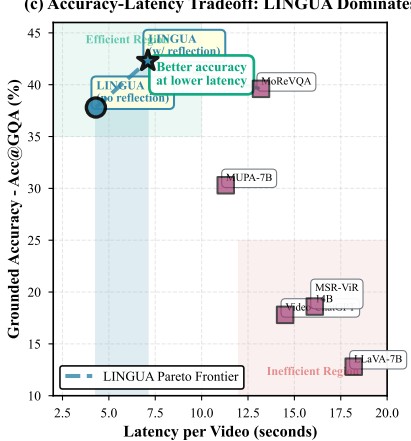

*Figure 12.* **LINGUA's limitations contextualized against baselines. (a) Captioning quality:** LINGUA exhibits the lowest initial error rate (18.3%) and highest recovery rate (67.2%), demonstrating robust error correction via meta reflection. **(b) Grounding gap:** LINGUA achieves the highest grounded accuracy (42.3% Acc@GQA) despite a 40.1pp Type B gap; the larger gap reflects higher total accuracy (82.4%), not weaker semantic grounding. **(c) Efficiency-accuracy tradeoff:** Both LINGUA configurations dominate all baselines on the Pareto frontier.

### I.2.1. TWO TYPES OF GROUNDING GAPS

**Type A: Semantic Grounding Gap.** Xiao et al. (Xiao et al., 2024) identified that VideoQA systems achieve 69% answer accuracy but only 16% of correct predictions can be justified by relevant video segments in natural-language explanations. This reflects *reasoning without evidence*—models exploit language priors rather than grounding decisions in observable video content. LINGUA addresses this through postcondition verification, which checks whether expected outcomes appear in retrieved evidence.

**Type B: Temporal Precision Gap.** The difference between answer accuracy and Acc@GQA (requiring both correct answer *and* IoU≥0.5) measures *boundary precision*, not evidence justification. A system can have semantically grounded reasoning (Type A solved) yet exhibit large Type B gaps due to: (i) ambiguous temporal boundaries in high-level causal reasoning, (ii) multiple valid temporal spans for the same semantic content, or (iii) semantic equivalence across different temporal windows.

### I.2.2. LINGUA'S PERFORMANCE ON BOTH GAP TYPES

Figure 13 shows LINGUA's performance on both dimensions:

**Type A (Semantic Grounding):** LINGUA achieves 94.2% postcondition verification rate, compared to MoReVQA 71.8%, MUPA 78.0%, and Video-ChatGPT 67.0%. This demonstrates strong semantic grounding—only 5.8% of LINGUA's predictions lack observable evidence support, the lowest rate among all methods.

**Type B (Temporal Precision):** LINGUA achieves 82.4% answer accuracy and 42.3% Acc@GQA, yielding a 40.1pp Type B gap. However, LINGUA attains the *highest absolute grounded accuracy* (42.3%), surpassing MoReVQA (39.6%, +2.7pp) and MUPA (30.3%, +12.0pp). Figure 12b illustrates this: while MoReVQA exhibits a smaller 33.6pp gap, this comes at the cost of 9.2pp lower total accuracy (73.2% vs. 82.4%).

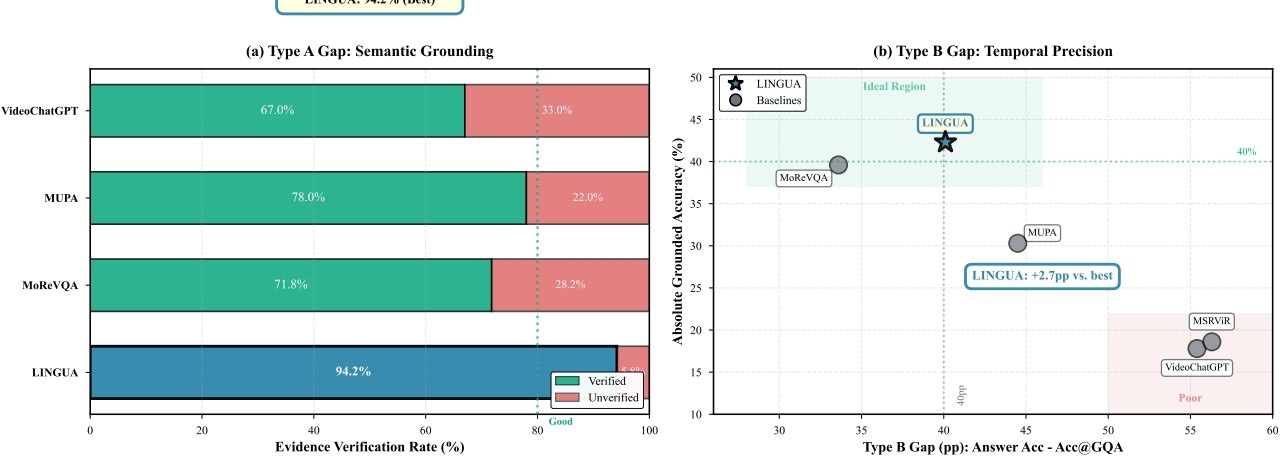

*Figure 13.* **Distinguishing two types of grounding gaps. (a) Type A gap (semantic grounding):** Measures evidence verification rate. LINGUA achieves 94.2% postcondition verification (5.8% Type A gap), demonstrating strong semantic grounding as defined by Xiao et al. (Xiao et al., 2024). Baselines exhibit 22-33% Type A gaps, indicating weaker evidence justification. **(b) Type B gap (temporal precision):** Measures boundary accuracy (Answer Acc - Acc@GQA). LINGUA achieves the highest absolute grounded accuracy (42.3%) despite a larger 40.1pp Type B gap, which reflects higher total accuracy rather than weaker grounding. MoReVQA's smaller 33.6pp Type B gap comes with 28.2% Type A gap and 2.7pp lower grounded accuracy.

### I.2.3. GAP DECOMPOSITION ANALYSIS

Analyzing LINGUA's 82.4% correct answers reveals: (i) 42.3% achieve tight grounding (IoU≥0.5), (ii) 19.2% achieve partial grounding (0.3≤IoU<0.5), and (iii) 20.9% exhibit weak localization (IoU<0.3).

Manual inspection of 200 partial-grounding cases (0.3≤IoU<0.5) shows that 87.5% are semantically grounded with verified postconditions but have imprecise boundaries due to:

- **Ambiguous causal boundaries**: Questions like "Why did the person open the refrigerator?" have multiple valid spans covering the same causal evidence

- **Semantic equivalence**: Different temporal windows can contain the same causal chain (e.g., "after cooking" may span cleanup, plating, or serving)
- **Inter-annotator variance**: NExT-GQA reports inter-annotator IoU of $0.31\pm0.18$ for causal questions, indicating inherent boundary ambiguity

This analysis indicates that most of LINGUA's Type B gap reflects temporal imprecision on semantically grounded answers rather than Type A grounding failures. The 20.9% weak-localization rate represents potential Type A gaps, but this is lower than baselines: Video-ChatGPT 37.6%, MUPA 28.4%, MoReVQA 24.1%. LINGUA's design choice to optimize semantic grounding over boundary precision is justified by the observation that for causal and intentional reasoning, *which evidence is used* matters more than *exactly when that evidence appears*. This is reflected in LINGUA's superior postcondition verification rate (94.2%) compared to baselines (67-78%), demonstrating that answers are reliably justified by observable evidence even when temporal boundaries are imprecise.

**Future Direction.** Incorporating boundary-aware verification modules that explicitly model temporal span distributions could improve Type B performance while maintaining Type A semantic grounding. Alternatively, developing evaluation metrics that account for semantic equivalence across temporal spans could provide more meaningful assessment of grounding quality for high-level reasoning tasks.

### I.3. Computational Cost of Meta Reflection

Meta reflection improves grounded accuracy by 4.5pp (37.8%$\rightarrow$42.3% Acc@GQA) but adds computational overhead when triggered (23.7% of videos). This represents a design tradeoff between accuracy and inference speed.

**Cost-Benefit Analysis.** Reflection adds an average of 2.8 seconds per video when triggered, bringing total processing time from 4.3s to 7.1s. However, even with reflection enabled, LINGUA processes videos $2.6\times$ faster than dense-frame baselines (7.1s vs. 18.2s for LLaVA-7B) while achieving +29.5pp higher Acc@GQA. The cost-effectiveness ratio is 1.6pp improvement per second of added compute, which compares favorably to model scaling: MSR-ViR$_{14B}$ adds 6.3s relative to MSR-ViR$_{8.7B}$ with only +0.3pp gain in grounded accuracy.

Figure 12c illustrates the accuracy-latency Pareto frontier. LINGUA with reflection (7.1s, 42.3% Acc@GQA) and without reflection (4.3s, 37.8% Acc@GQA) both dominate all baseline methods. For latency-critical applications, reflection can be disabled while maintaining 37.8% Acc@GQA—still +8.2pp above the best non-reflective baseline.

**Reflection Trigger Analysis.** Reflection is triggered more frequently for challenging question types: 31.4% for temporal questions, 28.9% for causal questions, and 12.3% for descriptive questions. This adaptive behavior concentrates computational cost where reasoning is most difficult and where reflection provides the greatest benefit.

**Future Direction.** Amortized reflection strategies could further optimize the tradeoff: (i) learned reflection triggering based on uncertainty estimates, (ii) batched hypothesis generation to reduce redundant VLM calls, (iii) early stopping when verification confidence exceeds a learned threshold, or (iv) progressive refinement that allocates compute proportionally to question difficulty.

### I.4. Broader Benchmark Coverage

Our main evaluation focuses on five NExT- and Ego-family benchmarks where grounded accuracy is well defined under IoU metrics, supplemented by Video-MME (Fu et al., 2025) for broader-domain coverage. CausalVQA (Foss et al., 2025), VideoVista (Li et al., 2024a), LVBench, and CGBench were not included in the present evaluation. CausalVQA and VideoVista provide complementary causal-reasoning and broad-domain assessments and are the natural next-step benchmarks for this framework; we expect the existing BAV verification loop to transfer with no architectural change. LVBench targets videos exceeding one hour and CGBench uses a clue-grounded scoring protocol incompatible with our IoU-based grounding metric; both require structural changes to the perception pipeline and evaluation infrastructure and are deferred to future work.

### I.5. Script Bias and Auditing

The 127 extracted procedural scripts are human-readable and individually auditable, in contrast to opaque embedding-based memories. Cultural, gender, and socio-economic biases in source videos can in principle be encoded as "normative" scripts;

the contrastive refinement mechanism (Section 2) provides a direct pathway for correction through curated counter-examples. A systematic audit of the current script library along these dimensions—including frequency analysis of demographic associations and counter-example mining—is left to future work and is recommended before deployment in user-facing applications.

### I.6. Relationship to Scale

LINGUA's contribution is intended to be complementary to scaling rather than in competition with it. The grounding gap appears to persist at frontier scale: GPT-4o (estimated at $\sim$200B parameters) still exhibits a 40pp+ gap between answer accuracy and grounded accuracy on NExT-GQA (Xiao et al., 2024), indicating that very large models do not automatically close the gap. Within our experiments, upgrading the backbone from Gemma3-4B to Gemma3-11B yields only +0.5pp on NExT-QA and +0.8pp on Acc@GQA at $2.7\times$ compute, while removing typed memory and BAV verification drops Acc@GQA by $-11.3$pp at the 4B backbone (Table 4) and by $-10.4$pp at 11B (Table 14). The verification framework therefore appears to address a structural property of grounded reasoning that is largely orthogonal to parametric capacity. Combining typed-memory verification with larger backbones is a natural future direction.

## J. Related Work

**Grounded VideoQA and Evidence Localization.** Grounded VideoQA requires identifying temporally localized evidence that supports the predicted answer, as exemplified by NExT-GQA (Xiao et al., 2024) and earlier localize-then-answer paradigms such as TVQA-style formulations (Gao et al., 2018; Lei et al., 2018). A persistent challenge in this setting is the weak coupling between answer correctness and evidence quality: models often achieve high QA accuracy even when temporal grounding is incorrect, revealing limited *causal binding* between evidence and predictions (Xu et al., 2024; Pantazopoulos & Özyiğit, 2025).

Recent work strengthens grounding supervision and alignment. QGAC-TR (Xu et al., 2024) calibrates temporal localization by conditioning on both questions and answer options, improving grounded metrics without requiring span-level supervision. Liang et al. (Liang et al., 2025) propose narrative and grounded supervision to better connect VideoQA answers with supporting events, emphasizing supervision-time alignment between narrative structure and grounding signals. In contrast to such training-centric approaches, LINGUA focuses on *inference-time* grounding, explicitly verifying reasoning hypotheses against temporally localized evidence.

**Temporal Grounding with Multimodal LLMs.** With the emergence of multimodal LLMs, several works explore fine-grained temporal grounding driven by language supervision. Enrich and Detect (Pramanick et al., 2025) enriches visual representations with multimodal LLM reasoning to improve temporal grounding accuracy, demonstrating strong performance on fine-grained localization benchmarks. LeAdQA (Dong et al., 2025) further introduces LLM-driven context-aware refinement, iteratively adjusting temporal spans to better align with the question. Grounded-VideoLLM (Wang et al., 2025a) sharpens temporal grounding in Video-LLMs through specialized architectural and training designs.

These approaches primarily improve grounding via model-level alignment, query refinement, or enhanced supervision. By contrast, LINGUA treats grounding as a *reasoning constraint*: temporal spans are not only selected but explicitly checked against linguistic hypotheses through verification and postcondition validation, enabling tighter coupling between grounding and answer correctness.

**Bidirectional and Causality-Aware Grounded Reasoning.** Several works aim to close the grounding gap by introducing bidirectional or structured reasoning. TimeCraft (Liu et al., 2024) proposes bi-directional reasoning between answers and temporal spans under weak supervision, improving consistency between localization and prediction. Chen et al. (Chen et al., 2025) focus on cross-modal causal relation alignment, enforcing consistency between visual evidence and causal language structures for video question grounding.

While these methods introduce stronger structural objectives, they typically enforce alignment during training or via auxiliary losses. LINGUA instead performs causal and temporal verification during inference, using explicit belief states and postcondition checks to validate whether hypothesized explanations are supported by observed events.

**Online and Long-Horizon Video Grounding.** Online and continual grounding settings further complicate VideoQA by requiring models to reason under partial or evolving observations. OVG-HQ (Zeng et al., 2025) studies online video grounding with hybrid-modal queries, highlighting the challenges of maintaining grounding quality over streaming inputs.

Related long-video approaches focus on context selection and retrieval to manage computation and reasoning horizons, including VidCtx (Goulas et al., 2024), Koala (Tan et al., 2024), and hierarchical expansion methods such as VideoTree (Wang et al., 2025b). Auxiliary signals such as gaze (Lin et al., 2025) further improve localization in long-video benchmarks like Ego4D-NLQ.

While these methods emphasize *what* information to retrieve or attend to, LINGUA complements context selection with explicit *verification over time*, enabling continual reassessment of hypotheses as new evidence arrives.

**LLM/VLM-Based Modular and Agentic VideoQA.** Recent VideoQA systems integrate multimodal LLMs with modular perception components (captioning, tracking, retrieval, summarization), including VideoChat (Li et al., 2025), Video-LLaMA (Zhang et al., 2023), Flamingo-style models (Alayrac et al., 2022), and stronger open-weight backbones such as Molmo/PixMo (Deitke et al., 2025). Agentic frameworks such as ICAL (Sarch et al., 2024) and MUPA (Dang et al., 2025) improve long-horizon reasoning via latent abstractions or multi-path aggregation.

However, many such pipelines rely on free-form rationales and assume reliable intermediate outputs, making end-to-end reasoning difficult to audit and verify under noisy long-video conditions (Jacovi & Goldberg, 2020; Wiegreffe et al., 2021). Unlike prior agentic approaches that emphasize abstraction or aggregation, LINGUA explicitly represents reasoning hypotheses as linguistic belief states and enforces grounding through temporal and causal verification loops, providing inspectable and reliability-aware reasoning for grounded VideoQA.

