# Bridging the Grounding Gap in VideoQA via Typed Memory for Language-based Belief-State Reasoning

## Abstract

VideoQA models can be accurate yet often fail to align answers with the correct video segments (the *grounding gap*). We introduce **LINGUA** (**L**anguage-based **IN**ference for **G**rounded Video **U**nderstanding **A**gent), a memory-based agent that performs grounded VideoQA by reasoning in an explicit *linguistic belief state*. LINGUA uses five mechanisms: (1) event-driven perception (retains 8–12% of frames while preserving 94% of question-relevant events); (2) typed memory for episodic narratives, semantic affordances, and procedural scripts; (3) Belief-Action-Verification loops with postcondition and temporal checks; (4) meta reflection with contrastive refinement; and (5) Bayesian reliability tracking for continual learning without gradient updates. Built with Gemma3-4B (Ollama, 4-bit), LINGUA outperforms strong baselines on five VideoQA benchmarks, reaching 82.4% on NExT-QA and 42.3% Acc@GQA on NExT-GQA (answer + IoU$\geq$0.5 temporal localization), while running 2.6$\times$ faster than dense-frame methods. In continual learning over 100 videos, accuracy rises from 45.2% (first 10) to 61.8% (last 10) without catastrophic forgetting, indicating online adaptation via memory refinement.

## 1. Introduction

Video Question Answering (VideoQA) has advanced rapidly with the emergence of large-scale vision–language models (VLMs) (Lei et al., 2021; Li et al., 2023a). Despite high accuracy achieved on benchmark datasets, recent studies reveal a critical limitation: correct answers do not necessarily reflect reliable *grounded reasoning*. Xiao et al. (Xiao et al., 2024) show that while a state-of-the-art system reaches 69%

[1]Anonymous Institution, Anonymous City, Anonymous Region, Anonymous Country. Correspondence to: Anonymous Author <anon.email@domain.com>.

Preliminary work. Under review by the International Conference on Machine Learning (ICML). Do not distribute.

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

| Method | Params | Frames Proc. | Time/Video (s) | FLOPs (G) | Memory (GB) | Throughput (vid/hr) | Acc@GQA (%) |
|---|---|---|---|---|---|---|---|
| *End-to-End Dense Processing* | | | | | | | |
| LLaVA-7B | 7B | 100% | $18.2_{\pm 1.1}$ | 278 | 8.4 | 197.8 | 12.8 |
| InstructBLIP | 7B | 100% | $19.5_{\pm 1.3}$ | 295 | 9.1 | 184.6 | 15.7 |
| Video-ChatGPT | 7B | 100% | $14.5_{\pm 0.9}$ | 203 | 7.2 | 248.3 | 17.8 |
| TimeChat | 7B | 100% | $15.1_{\pm 1.0}$ | 215 | 7.5 | 238.4 | 19.4 |
| *Modular & Adaptive Methods* | | | | | | | |
| MUPA-2B | 2B | 100% | $11.3_{\pm 0.7}$ | 167 | 5.8 | 318.6 | 28.7 |
| MUPA-7B | 7B | 100% | $13.8_{\pm 0.9}$ | 198 | 7.0 | 260.9 | 30.3 |
| MSR-ViR$_L$ | 8.7B | 50% | $9.8_{\pm 0.6}$ | 125 | 6.3 | 367.3 | 18.6 |
| **LINGUA** | **4B** | **∼10%** | $\mathbf{7.1_{\pm 0.4}}$ | **62** | **3.9** | **507.0** | **42.3** |