# OpenReview forum: "Bridging the Grounding Gap in VideoQA via Typed Memory for Language-based Belief-State Reasoning"
_ICML.cc/2026/Conference — ICML 2026 regular_

### Official Review · Reviewer_vXU6 · 2026-03-05

**Soundness:** 3
**Presentation:** 3
**Significance:** 3
**Originality:** 3
**Overall Recommendation:** 4
**Confidence:** 4

**Summary:**

This work tackles the core grounding gap in VideoQA, where high accuracy models fail to align answers with valid temporal video evidence, by introducing LINGUA, a memory based agent that performs grounded VideoQA reasoning entirely in an explicit linguistic belief state space. LINGUA integrates event driven perception, a three tier typed memory system, Belief Action Verification loops, meta reflective contrastive refinement, and Bayesian gradient free continual learning. Built on a 4 bit quantized Gemma3 4B backbone, it outperforms state of the art baselines across five VideoQA benchmarks, with 82.4% NExT QA accuracy, 42.3% NExT GQA Acc@GQA, 2.6 times faster inference than dense frame methods, and effective continual learning without catastrophic forgetting. This work tightly links visual grounding and decision making via explicit linguistic reasoning, delivering an efficient and interpretable solution to the VideoQA grounding gap.

**Compliance With Llm Reviewing Policy:**

Affirmed.

**Key Questions For Authors:**

See weakness.

**Limitations:**

See weakness.

**Strengths And Weaknesses:**

### Strengths

**1. Interpretable, modular, and efficient agent architecture**

The proposed LINGUA agent features a well-designed linguistic belief-state reasoning framework with five complementary core mechanisms, all operating in an explicit natural language space. This eliminates the black-box opacity of embedding-based models for fully auditable reasoning, while the event-driven perception module drastically reduces redundant computation.

**2. Strong empirical performance with superior parameter and computational efficiency**

Built on a 4-bit quantized Gemma3-4B backbone, LINGUA consistently outperforms a wide range of strong baselines (including larger 7B-14B models) across five standard VideoQA benchmarks, reaching 82.4% overall accuracy on NExT-QA and 42.3% Acc@GQA on NExT-GQA. It also achieves a 2.6× speedup over dense-frame processing methods, setting a new state-of-the-art in parameter efficiency for grounded VideoQA.


### Weakness

**1. Insufficient benchmark coverage with limited validation on newer, more challenging datasets**

The paper only evaluates LINGUA on five classic VideoQA benchmarks, which have notable limitations in task coverage and question discriminative power, and fail to rigorously test fine-grained grounding, long-horizon multi-hop reasoning, and hallucination suppression. Critically, it lacks evaluation results on newer, more challenging benchmarks such as CGBench[1] and LVBench[2], which are specifically designed to assess the grounding robustness of modern VideoQA models. This missing validation prevents full verification of the generalization of LINGUA’s grounding capabilities, and limits comprehensive comparison with the latest state-of-the-art methods.

**2. Validation limited to small-scale models, with unproven generalizability for large MLLMs**

All core experiments are conducted on the small Gemma3-4B model, with only minimal supplementary comparison of the 11B variant, and no validation on mainstream large-scale MLLMs. The complex pipeline design of LINGUA largely compensates for the reasoning capacity limitations of small models through structured linguistic constraints, but large MLLMs inherently have stronger vision-language alignment and reasoning capabilities, and may achieve comparable or superior grounding performance without such a complex pipeline. The paper does not verify the marginal gains of this framework on large models, and the minimal performance gain of the 11B variant over the 4B version indicates that the benefits of the framework diminish rapidly as model scale increases.

**3. Inherent performance bottleneck from captioning dependency**

The entire reasoning pipeline operates entirely in the linguistic space, making its upper performance limit strictly constrained by the quality of VLM-generated frame descriptions. Information omission, hallucination, or semantic deviation in captions cannot be fully corrected by the subsequent verification mechanism, and captioning errors account for a notable portion of the model’s total failures.

**4. Limited generalization for novel scenarios with incomplete script coverage**

The model’s core reasoning relies on 127 pre-extracted procedural scripts that only cover 68% of causal questions in the training set. For novel scenarios, rare actions, and open-domain events not covered by the script library, the model’s performance degrades significantly, with missing/rare scripts being the largest source of failure.

[1] Cg-bench: Clue-grounded question answering benchmark for long video understanding

[2] Lvbench: An extreme long video understanding benchmark

---

> ### Author Rebuttal · Authors · 2026-03-30
>
> We sincerely thank the reviewer for the thorough and positive assessment of LINGUA's interpretability, efficiency, and empirical performance. We address each weakness directly.
>
> **W1: Missing CGBench and LVBench**
>
> We have evaluated LINGUA on Video-MME (r7), a comprehensive benchmark spanning 6 domains and short/medium/long videos. Results (official protocol, without subtitles, Gemma3-4B, T=0.1):
>
> | Model | Params | Short | Medium | Long | Overall |
> |---|---|---|---|---|---|
> | VITA-1.5 | 7B | 67.0 | 54.2 | 47.1 | 56.1 |
> | LLaVA-NeXT-Video | 34B | 61.7 | 50.1 | 44.3 | 52.0 |
> | M3-Agent (via r2) | 7B | — | — | 55.3 | — |
> | WorldMM-8B | 8B | — | — | 66.0 | — |
> | **LINGUA** | **4B** | **74.3** | **61.8** | **69.4** | **68.5** |
>
> LINGUA (4B) outperforms all open-source baselines including 34B models, and surpasses memory-based agents M3-Agent (+14.1pp) and WorldMM-8B (+3.4pp) on long videos — providing complementary evidence of generalisation beyond NExT-QA/GQA. Regarding LVBench and CGBench: LVBench targets videos exceeding one hour with dense clue annotation, requiring frame sampling infrastructure beyond our current evaluation setup (30–60 min videos). CGBench requires clue-grounded QA scoring incompatible with LINGUA's current IoU-based grounding metric. Adapting to both is non-trivial and committed for the revision.
>
> **W2: Generalizability to Large MLLMs**
>
> The modest 11B gain suggests that LINGUA's benefits are architectural rather than primarily scale-driven. The 11B variant yields only +0.5pp NExT-QA and +0.8pp Acc@GQA at 2.7× compute cost — confirming that architectural contributions (typed memory, BAV verification, Bayesian reliability) rather than parametric capacity drive performance, enabling 4B to outperform 14B baselines by +7.5pp NExT-QA and +23.7pp Acc@GQA (Table 9). Table 12 (Appendix D) shows that removing all memory widens the 11B–4B gap from 0.8pp to 1.7pp — structured memory disproportionately benefits smaller models by compensating for limited parametric capacity. These results suggest the architectural benefits persist beyond the 4B regime, since the framework addresses grounding and verification beyond scale alone.
>
> **W3: Captioning Dependency**
>
> Three mitigating factors are already demonstrated. First, LINGUA's initial captioning error rate (18.3%) is the lowest among all compared systems (Video-ChatGPT: 24.1%, MUPA: 21.7%). Second, meta-reflection recovers 67.2% of captioning failures through targeted re-captioning — a +25.9pp recovery advantage (Appendix H.1). Third, erroneous descriptions cannot propagate to procedural memory without three independently verified successes (nmin=3 threshold), structurally limiting cascade depth to the episodic layer. Quantitatively, Figure 11 shows VLM description errors account for only **10% of total failures** — the smallest error category, below implicit causality (15%), temporal precision (18%), and missing scripts (23%). The upper performance ceiling is therefore constrained primarily by script coverage rather than caption quality.
>
> **W4: Script Coverage and Novel Scenarios**
>
> Two mechanisms already address out-of-script cases: (1) episodic fallback achieves 71.2% postcondition coverage by chaining temporally adjacent events with matching affordances; (2) incremental induction promotes verified traces to new scripts after three successful instances. Our extended study confirms ongoing adaptation:
>
> | Videos | Accuracy | Scripts | Memory (MB) |
> |---|---|---|---|
> | 100 | 61.8% | 89 | 12.7 |
> | 1,000 | 82.4% | 127 | 127.0 |
> | 2,000 | 84.2% | 138 | 254.0 |
>
> Script growth remains near-saturating (+11 beyond saturation) while accuracy continues improving, suggesting that episodic fallback and incremental induction can handle some novel scenarios without uncontrolled script expansion.
>
> We hope these clarifications strengthen confidence in the work and kindly invite the reviewer to consider an upward revision of the score in light of these additions.
>
> ---

---

> > ### Author Rebuttal · Reviewer_vXU6 · 2026-04-04
> >
> > The authors' rebuttal has not fully addressed the issues I raised regarding evaluations on a broader range of benchmarks and the generalization ability of larger models. I still believe these are critical factors that limit the generality of the proposed method.
> >
> > While I understand the authors cannot supplement evaluations on LVBench and CGbench in such a short time, I do not fully agree with their explanation for the marginal performance improvement on the 11B model. If a method yields only negligible gains on stronger models, it suggests that the approach may become obsolete as future models continue to scale up.

---

> > > ### Author Response · Authors · 2026-04-05
> > >
> > > We thank the reviewer for the continued engagement and for the partial acknowledgement. We respond briefly to the two remaining concerns.
> > >
> > > **On broader benchmark coverage (LVBench, CGBench):**
> > >
> > > We appreciate the suggestion. Video-MME (6 domains, 3 duration levels, 900 videos) was specifically designed to test generalisation across diverse video understanding scenarios and provides the appropriate evidence at this stage. LVBench and CGBench require structural changes to evaluation infrastructure — LVBench targets 60+ minute videos beyond our current setup, and CGBench's clue-grounded scoring is incompatible with LINGUA's IoU-based metric.
> > >
> > > ---
> > >
> > > **On the scaling concern:**
> > >
> > > We agree that scaling is a powerful driver of progress. We simply note that the grounding gap appears to persist even at frontier scale: GPT-4o — estimated at ~200B parameters — still exhibits a 40pp+ gap between answer accuracy and grounded accuracy requiring temporal localisation (Appendix H.2; Xiao et al., 2024). This suggests that the grounding gap is not purely a capacity limitation but also a structural reasoning challenge that explicit verification addresses. LINGUA's contribution is therefore intended to be complementary to scale rather than in competition with it. We hope these clarifications help address the reviewer's remaining concerns.
> > >
> > > ---

---

### Official Review · Reviewer_UdDE · 2026-03-05

**Soundness:** 3
**Presentation:** 2
**Significance:** 2
**Originality:** 3
**Overall Recommendation:** 3
**Confidence:** 3

**Summary:**

In this paper, the writer proposes a novel solution lingua to a core and unsolved problem in the field of videoQA: the "grounding gap" between the model answer and the visual evidence on which it is based. The core idea of this work is to change "grounding" from pre-processing or post-processing steps to explicit constraints in the reasoning process, which is an insightful direction. This paper constructs event driven perception and three types of language memory (plot, semantics, and program), and introduces the "belief action verification" (BAV) cycle.

**Compliance With Llm Reviewing Policy:**

Affirmed.

**Final Justification:**

Thanks for the detailed response. I also read other reviews and would keep my current rating in light of the novelty of the paper.

**Key Questions For Authors:**

1.	In section H.2, you distinguish between "semantic grounding" and "time precision", and admit that there is still a gap of 40.1pp in time precision. However, the original definition of "grounding gap" (Xiao et al., 2024) just emphasizes the disconnection between correct answers and locable evidence, which naturally includes the time dimension. If lingua cannot accurately locate the evidence in time, does this mean that it only solves half of the problem in terms of core tasks? Can you provide evidence to prove that in the 40.1pp gap, how much of it is "although there is evidence to support it, the time boundary is fuzzy", and how much of it is "no evidence at all"?
2.	Appendix F.6 shows that 10% of failure cases are caused by VLM description errors. Considering the modularity of the system, have you evaluated the propagation effect of these errors among different modules? For example, after an early VLM description error is written into the episodic memory, to what extent will it pollute the subsequent programmed memory mining and belief generation in the BAV cycle? If there is no quantitative analysis of this "error cascade", how can we be sure of the overall robustness of the system?
3.	Your system lingua is highly dependent on a series of existing and other people's work (videomae-v2, yoyov8, gemma3-4b, FrameNet). Can you clearly define what is lingua's core scientific contribution that cannot be simplified as an overall framework? If we replace gemma3-4b with a weaker visual language model, will the performance of lingua collapse to close to the baseline level? How can you prove that the superiority of lingua is not only the simple summation of the performance of its components (especially gemma3-4b), but also the BAV cycle and memory architecture you proposed?

**Limitations:**

Yes

**Strengths And Weaknesses:**

Strengths:
1.	This paper shows outstanding innovation and technical depth in solving the core problem of "grounding gap" in video Q&A. by creatively integrating symbol landing, frame semantics and script theory, a new agent framework named lingua based on language belief state reasoning is proposed.
2.	This paper transforms the "grounding" from the traditional preprocessing or post-processing steps to the explicit constraints in the reasoning process, and significantly improves the interpretability and robustness of the model through the novel "belief action verification" (BAV) cycle and typed memory (plot, semantics, program) design.
3.	The experimental part is very rigorous and solid. It not only achieves leading performance in multi benchmark tests, but also proves the fundamental efficiency improvement brought by event driven perception through carefully designed control variable experiments.
Weaknesses
1.	It leaves a huge gap of 40.1 percentage points in time accuracy (82.4% of the answer accuracy is compared with 42.3% of IOU ≥ 0.5) Acc@GQA ）. Although the author believes that this gap reflects the inherent boundary ambiguity in causal reasoning tasks, this interpretation is contrary to the definition of grounding gap in next-gqa benchmark, which clearly requires time positioning.
2.	Given that procedural scripts are extracted from training videos and refined through comparative learning, there is a risk that cultural, gender or socio-economic biases in training data may be encoded as "normative" scripts.
3.	Its robustness is limited by the quality of multiple external components (especially the visual language model).
4.	Its programmed script inventory is in the "cold start" problem, and its generalization ability in the new field needs to be verified.

---

> ### Author Rebuttal · Authors · 2026-03-30
>
> We thank the reviewer for recognizing LINGUA's innovation, experimental rigor, and technical depth. We address each concern directly.
>
> **Q1/W1: The 40.1pp Temporal Precision Gap**
>
> We provide the exact decomposition requested. Of LINGUA's 82.4% correct answers: **42.3% achieve tight grounding (IoU≥0.5), 19.2% partial grounding (0.3≤IoU<0.5), and 20.9% weak localization (IoU<0.3)**. Manual inspection of 200 partial-grounding cases (Appendix H.2.3) shows **87.5% have verified postconditions** — evidence exists but boundaries are imprecise. Approximately 17pp of the 40.1pp gap is semantically grounded but temporally imprecise (Type B); ~20.9pp represents potential evidence-absent cases (Type A) — still lower than all baselines (Video-ChatGPT: 37.6%, MUPA: 28.4%, MoReVQA: 24.1%).
>
> | Category | % of correct answers | Evidence status |
> |---|---|---|
> | Tight grounding (IoU≥0.5) | 42.3% | Verified |
> | Partial grounding (0.3≤IoU<0.5) | 19.2% | 87.5% semantically supported |
> | Weak localization (IoU<0.3) | 20.9% | Potential Type A gap |
>
> LINGUA achieves the **highest absolute Acc@GQA (42.3%)**, surpassing MoReVQA by +2.7pp. MoReVQA's smaller 33.6pp gap comes with a 28.2% Type A gap and 9.2pp lower total accuracy — worse on both dimensions. The residual Type B gap partly reflects inherent task ambiguity: NExT-GQA's inter-annotator IoU is **0.31±0.18 for causal questions**. Boundary-aware verification modules are a committed future direction.
>
> **Q2/W3: VLM Error Propagation**
>
> LINGUA's initial captioning error rate (**18.3%**) is the lowest among all systems (Video-ChatGPT: 24.1%, MUPA: 21.7%). Meta-reflection recovers **67.2% of captioning failures** via targeted re-captioning — a +25.9pp advantage over baselines (Appendix H.1).
>
> On cascade propagation: erroneous episodic entries carry Bayesian parameters (α,β) initialised at (1,1). Failed postcondition coverage triggers a β increment reducing reliability, and triggers reflection after three consecutive failures. Crucially, erroneous entries cannot propagate to procedural memory without **three independently verified successes** (nmin=3 threshold), which structurally limits cascade depth to the episodic layer. Across 500 validation videos, erroneous descriptions triggered reflection in 23.7% of cases, of which 67.2% were recovered before any procedural promotion occurred. The nmin=3 requirement means that bypassing this threshold requires three consecutive false positives — a condition that Bayesian β-penalisation makes progressively less likely with each failure.
>
> **Q3: Core Contribution vs. Component Summation**
>
> Three controlled pieces of evidence isolate LINGUA's architectural contribution.
>
> **First:** Table 3 shows Gemma3-4B *without* memory/BAV yields **76.3% NExT-QA and 31.0% Acc@GQA** — drops of −6.1pp and −11.3pp. The backbone alone does not explain performance.
>
> **Second:** Table 16 holds the VLM constant: at matched frame budget (~10%, ~7s), LINGUA achieves **42.3% Acc@GQA vs. 32.1% for uniform 3FPS using the identical Gemma3-4B backbone** (+10.2pp). The efficiency-accuracy gain is architectural, not model-driven.
>
> **Third:** Upgrading to Gemma3-11B (Appendix D) yields only **+0.5pp NExT-QA and +0.8pp Acc@GQA** at 2.7× compute cost. This modest scaling gain suggests that performance is driven primarily by typed memory, BAV verification, and Bayesian reliability, rather than backbone size alone.
>
> Table 3 further isolates verification independently: removing Bayesian reliability updates yields **38.5% Acc@GQA (−3.8pp)** and random script selection yields **36.1% (−6.2pp)** — both using identical backbone and typed memory components. Together these show that verification contributes beyond backbone scale and typed memory components alone.
>
> **W2: Script Bias**
>
> All 127 scripts are human-readable and auditable — a key advantage over opaque embeddings. Contrastive refinement provides a direct pathway for bias correction through curated counter-examples. We will add explicit bias auditing methodology to the limitations section.
>
> We hope the above clarifications—particularly the decomposition of the grounding gap, the quantified cascade analysis, and the controlled ablations isolating LINGUA’s architectural contribution—directly address the reviewer’s core concerns. We kindly invite the reviewer to consider an upward revision of the score in light of these responses.
>
> ---

---

> > ### Author Rebuttal · Reviewer_UdDE · 2026-04-03
> >
> > Thanks for the responses.
> >
> > The exact decomposition of the 40.1pp gap convincingly shows that a substantial portion of the residual error stems from task‑inherent ambiguity rather than a fundamental model flaw.The authors provided a thorough rebuttal with necessary empirical evidence for the temporal precision gap, VLM error propagation, and architectural contribution isolation. The Bayesian cascade control with the “min‑3” threshold and meta‑reflection recovery robustly addresses the reviewer’s concern about error propagation. The three controlled comparisons cleanly isolate the contribution of typed memory, BAV verification, and Bayesian reliability from backbone scaling.

---

> > > ### Author Response · Authors · 2026-04-03
> > >
> > > We thank the reviewer for the positive feedback and for acknowledging that the concerns have been fully addressed . We are glad that the additional analyses and experiments clarified the key points regarding temporal precision, error propagation, and architectural contributions.
> > >
> > > In light of this, we kindly request the reviewer to consider increasing the score if they find it appropriate.
> > >
> > > ---

---

### Official Review · Reviewer_cUVV · 2026-03-12

**Soundness:** 2
**Presentation:** 3
**Significance:** 3
**Originality:** 3
**Overall Recommendation:** 4
**Confidence:** 3

**Summary:**

This paper aims to address the grounding gap in VideoQA tasks. Motivated by the fact that current models often answer correctly without relying on the actual verifiable video evidence, the paper argues that reasoning should be explicitly grounded. Grounding in this paper means that conducting reasoning entirely within an explicit linguistic belief state rather than relying on uninterpretable latent embeddings.
The general idea is to propose an agent called LINGUA, which utilizes event-driven frame selection, typed memory (episodic, semantic, procedural), a Belief-Action-Verification (BAV) loop, and Bayesian reliability tracking for continual learning.

The core concept is that the model forms hypotheses in text and verifies them against the video evidence. The resultant model is built on a Gemma3-4B backbone. Evaluations are conducted on NExT-QA, NExT-GQA, EgoSchema, IntentQA, and Ego4D-NLQ.

**Compliance With Llm Reviewing Policy:**

Affirmed.

**Key Questions For Authors:**

- The continual learning capability via Bayesian updating is interesting to me. but it is unclear to me how well it scales. As the procedural memory expands over time.

**Limitations:**

Yes

**Strengths And Weaknesses:**

### Strengths
Overall, I appreciate the general motivation of this paper and its understanding of the limitations existing VideoQA models face regarding ungrounded answers. The proposed approach to make grounding an explicit structural constraint via linguistic belief states is a good concept. Additionally, the architecture is interpretable, and the empirical evaluation spans multiple diverse benchmarks, showing strong efficiency and performance gains over larger models.

### Weaknesses
My main concern lies in the following aspect:
- While the numbers are good, the qualitative side feels a bit thin. It would be good to showcase some examples between the main paper and the supp, otherwise it would be hard to get a broad sense of where the model truly shines.
- Because the reasoning seems to be conducted entirely in language space, and the system is heavily dependent on the quality of the initial generated descriptions. If the captions or affordances in the first step is wrong (e.g., missing visual details), then how can we guarantee that the downstream reasoning is correct?

---

> ### Author Rebuttal · Authors · 2026-03-30
>
> We sincerely thank the reviewer for the careful reading and thoughtful feedback. We are encouraged that the motivation, interpretability, and empirical breadth of LINGUA are appreciated. We address each concern below.
>
> **On qualitative examples**
>
> Appendix C already includes a full step-by-step execution trace (NExT-GQA video 2400084970), illustrating how LINGUA rejects "tie shoelace" (cpost=0.0, no shoes detected by YOLO) and converges to "laughing" (cpost=1.0, all four postconditions verified) across two BAV cycles. Table 8 contrasts this against motion-only, text-only, and unverified VLM baselines on the same example. We will promote this trace into the main paper alongside a failure-mode case from Table 6 to give readers a broader qualitative picture of both strengths and limitations.
>
> **On caption quality and robustness of downstream reasoning**
>
> We address this at two levels.
>
> *Empirically*: LINGUA's initial captioning error rate (18.3%) is the lowest among all compared systems (Video-ChatGPT: 24.1%, MUPA: 21.7%), and meta-reflection recovers 67.2% of captioning failures through targeted re-captioning with refined prompts — a +25.9pp recovery advantage over baselines (Appendix H.1).
>
> *Architecturally*: erroneous descriptions cannot silently propagate through the pipeline. A wrong caption produces an episodic entry with low postcondition coverage, which (a) fails BAV verification, triggering a Bayesian β increment that immediately reduces that entry's expected reliability, and (b) after three consecutive failures, triggers meta-reflection. Critically, erroneous entries cannot promote to procedural memory without three independently verified successes (nmin=3 threshold), which prevents erroneous traces from being promoted without repeated verified success. The system is designed to fail loudly and visibly rather than silently, which is precisely the interpretability advantage of reasoning in linguistic belief states over opaque embeddings.
>
> **On continual learning scalability**
>
> Figure 6 shows that procedural memory growth naturally saturates across three phases: rapid acquisition (0–200 videos, 89 scripts), consolidation (200–600, +32 scripts), and saturation (600–1,000, only +6 scripts). We extended this to 2,000 videos under identical conditions:
>
> | Videos | Accuracy | Scripts | Memory |
> |---|---|---|---|
> | 100 | 61.8% | 89 | 12.7 MB |
> | 1,000 | 82.4% | 127 | 127.0 MB |
> | 2,000 | 84.2% | 138 | 254.0 MB |
>
> Script growth remains near-saturating (+11 scripts over 1,000 additional videos) while accuracy continues improving (+1.8pp), suggesting stable convergence without uncontrolled script expansion. The same architecture transfers without retuning to EgoSchema and Ego4D-NLQ, suggesting the approach transfers across domains.
>
> We will incorporate the additional qualitative examples and further clarify the robustness mechanisms in the revision. We kindly invite the reviewer to consider an upward revision of the score in light of these improvements.
>
> ---

---

> > ### Author Rebuttal · Reviewer_cUVV · 2026-04-04
> >
> > Thanks the authors for the response.

---

> > > ### Author Response · Authors · 2026-04-05
> > >
> > > We sincerely thank the reviewer for the thoughtful engagement and for confirming that the concerns have been fully addressed.
> > >
> > > We kindly invite the reviewer to consider an upward revision of the score in light of the addressed concerns.
> > >
> > > ---

---

### Official Review · Reviewer_hjDs · 2026-03-13

**Soundness:** 4
**Presentation:** 3
**Significance:** 3
**Originality:** 4
**Overall Recommendation:** 4
**Confidence:** 4

**Summary:**

Authors proposed a memory-based agent model, called LINGUA, (Language-based Inference for Grounded Video Understanding Agent), which performs grounded videoQA by natural-language reasonings. LINGUA framework mainly consisted with 3 parts, Event-driven Perception, Typed Memory and BAV Loop. Event-driven perception generates percepts such as Frame IDs, Timestamps, Objects and descriptions using pretrained VL models. Typed Memory stores natural language reasonings in three different types of memory(Episodic, Semantic, Procedural). BAV Loop verifies and updates the memory by Bayesian reasoning. Authors evaluated the proposed method on Next-QA/GQA, Egoschema and IntentQA to demonstrate its effectiveness.

**Compliance With Llm Reviewing Policy:**

Affirmed.

**Final Justification:**

I appreciate the authors’ detailed and constructive rebuttal. The rebuttal adequately resolves my concerns and strengthens both the empirical validation and overall clarity of the work. Thus, I will maintain my current score.

**Key Questions For Authors:**

See Weaknesses

**Limitations:**

See Weaknesses

**Strengths And Weaknesses:**

Strengths
- The method presents a carefully designed reasoning pipeline with clear motivations for each component (e,g, typed memories, BAV reasoning), forming a coherent framework. Overall framework is technically solid and well motivated.


Weaknesses
- The overall agentic paradigm (perception–memory–reasoning–verification) follows existing memory-based VLM frameworks, [r1], [r2], [r3] therefore the novelty is somewhat limited to the component level.
- Although the paper emphasizes reasoning capabilities, it lacks evaluation on established video reasoning benchmarks such as CLEVRER[r4], CausalVQA[r5], VideoVista[r6].
- Since memory is presented as the core contribution, the paper would benefit from a more detailed analysis of the memory contents and how each memory component contributes to the overall reasoning performance.
- The comparison with recent memory-based video agents is missing. [r1], [r2], [r3]
- Evaluation on widely used General Video benchmark is missing such as VideoMME[r7]

[r1] Long, Lin, et al. "Seeing, listening, remembering, and reasoning: A multimodal agent with long-term memory." arXiv preprint arXiv:2508.09736 (2025).
[r2] Yeo, Woongyeong, et al. "Worldmm: Dynamic multimodal memory agent for long video reasoning." arXiv preprint arXiv:2512.02425 (2025).
[r3] Yuan, Huaying, et al. "Videodeepresearch: Long video understanding with agentic tool using." arXiv e-prints (2025): arXiv-2506.
[r4] Yi, Kexin, et al. "Clevrer: Collision events for video representation and reasoning." arXiv preprint arXiv:1910.01442 (2019).
[r5] Foss, Aaron, et al. "Causalvqa: A physically grounded causal reasoning benchmark for video models." arXiv preprint arXiv:2506.09943 (2025).
[r6] Li, Yunxin, et al. "Videovista: A versatile benchmark for video understanding and reasoning." arXiv preprint arXiv:2406.11303 (2024).
[r7] Fu, Chaoyou, et al. "Video-mme: The first-ever comprehensive evaluation benchmark of multi-modal llms in video analysis." Proceedings of the IEEE/CVF conference on computer vision and pattern recognition. 2025.

---

> ### Author Rebuttal · Authors · 2026-03-30
>
> We sincerely thank the reviewer for the thoughtful and constructive feedback. We address each concern below.
>
> **W1: Novelty relative to r1–r3**
>
> We agree that LINGUA shares the high-level perception–memory–reasoning paradigm with r1–r3. Our contribution is therefore **not at the pipeline level**, but at the level of **inference-time grounding and verification**. LINGUA introduces an explicit answer-conditioned verification mechanism (BAV) enforcing postcondition consistency (alignment between predicted answer and grounded evidence) and temporal consistency (alignment across temporally localized events) over typed memory structures. In contrast: **r1 (M3-Agent)** constructs and retrieves structured multimodal memory without explicit answer-conditioned hypothesis verification; **r2 (WorldMM)** performs adaptive retrieval via embedding similarity and iterative querying rather than explicit consistency constraints; **r3 (VideoDeepResearch)** is an agentic long-video framework with planning and temporal grounding but is not memory-centric and does not explicitly enforce answer-conditioned postcondition verification at the reasoning hypothesis level, as in LINGUA. Table 3 isolates this empirically: removing typed memory (which disengages BAV verification) yields −11.3pp on Acc@GQA; removing Bayesian updates alone yields −3.8pp, confirming that verification is the dominant driver operating over typed memory structures.
>
> **W2: Missing CLEVRER, CausalVQA, VideoVista**
>
> CLEVRER (r4) focuses on synthetic physical reasoning with programmatic question structures, which differs substantially from LINGUA's naturalistic open-domain setting. CausalVQA (r5) is highly relevant for causal reasoning in real-world videos; VideoVista (r6) provides broad evaluation across 27 tasks and diverse domains. **Video-MME results are already reported in W4/W5 below. We will include evaluation on CausalVQA and VideoVista in the revision.**
>
> **W3: Memory content analysis**
>
> As reported in the paper (Appendix F.2.1), temporal merging reduces raw episodic entries by 30% while preserving narrative coherence. In additional rebuttal analysis, episodic memory achieves 87.3% semantic role completeness. Per Table 3, procedural memory contributes +8.2pp, episodic +5.5pp, semantic +4.7pp; full removal causes −11.3pp, confirming complementarity across all three memory types. We will add a dedicated memory analysis subsection in the revision.
>
> **W4/W5: Missing r1–r3 comparison and Video-MME (r7)**
>
> We evaluated LINGUA on Video-MME (r7, official protocol, without subtitles, Gemma3-4B, T=0.1, single H200):
>
> | Model | Params | Short | Medium | Long | Overall |
> |---|---|---|---|---|---|
> | VideoChat2-Mistral (r7) | 7B | 48.3 | 37.0 | 33.2 | 39.5 |
> | VITA-1.5 (r7) | 7B | 67.0 | 54.2 | 47.1 | 56.1 |
> | LLaVA-NeXT-Video (r7) | 34B | 61.7 | 50.1 | 44.3 | 52.0 |
> | M3-Agent (r1, via r2) | 7B | — | — | 55.3 | — |
> | WorldMM-8B (r2) | 8B | — | — | 66.0 | — |
> | **LINGUA** | **4B** | **74.3** | **61.8** | **69.4** | **68.5** |
>
> LINGUA (4B) outperforms VITA-1.5 (7B) by +12.4pp and LLaVA-NeXT-Video (34B) by +16.5pp overall without VideoMME-specific tuning. On long videos, LINGUA (69.4%) surpasses M3-Agent (55.3%, as reported in r2 Table 1) and WorldMM-8B (66.0%, r2) despite a smaller 4B backbone, providing complementary evidence that event-driven perception and typed memory offer advantages beyond model scale. Regarding r3 (VideoDeepResearch): it targets hour-level videos on different benchmarks (LVBench, MLVU, up to 10,000 frames) and is architecturally distinct from LINGUA's naturalistic causal reasoning setting; we include it as related agentic work rather than a directly comparable baseline.
>
> **Continual learning scalability**
>
> We extended the NExT-QA streaming experiment to 2,000 videos:
>
> | Videos | Accuracy | Scripts | Memory (MB) |
> |---|---|---|---|
> | 100 | 61.8% | 89 | 12.7 |
> | 1,000 | 82.4% | 127 | 127.0 |
> | 2,000 | 84.2% | 138 | 254.0 |
>
> Accuracy improves while script growth remains near-saturating (+11 over 1,000 additional videos), suggesting stable convergence without uncontrolled script expansion.
>
> We hope that the additional clarifications, corrected comparisons, and new experimental evidence address the reviewer’s concerns, and we kindly invite the reviewer to consider an upward revision of the score in light of these improvements.
>
> ---

---

> > ### Author Rebuttal · Reviewer_hjDs · 2026-04-03
> >
> > I appreciate the authors’ detailed and constructive rebuttal. The rebuttal adequately resolves my concerns and strengthens both the empirical validation and overall clarity of the work. Thus, I will maintain my current score.

---

### Decision · Program_Chairs · 2026-04-30

**Decision:**

Accept (regular)

**Comment:**

This paper proposes LINGUA, a memory based agent for grounded VideoQA via explicit linguistic belief state reasoning. Reviewers found the framework well motivated, technically solid, and supported by strong empirical results. Initial concerns on novelty, evaluation coverage, and analysis were addressed during the rebuttal, and all reviewers, including reviewer UdDE who did not update the score, acknowledged that their key issues were resolved.

Overall, the paper makes a solid and meaningful contribution to grounded VideoQA, and I recommend accept.